# On The Expressive Power of GNN Derivatives

**Yam Eitan**
Technion – Israel Institute of Technology

**Moshe Eliasof**
Ben-Gurion University of the Negev
University of Cambridge

**Yoav Gelberg**
University of Oxford

**Fabrizio Frasca**
Technion – Israel Institute of Technology

**Guy Bar-Shalom**
Technion – Israel Institute of Technology

**Haggai Maron**
Technion – Israel Institute of Technology
NVIDIA Research

## Abstract

Despite significant advances in Graph Neural Networks (GNNs), their limited expressivity remains a fundamental challenge. Research on GNN expressivity has produced many expressive architectures, leading to architecture hierarchies with models of increasing expressive power. Separately, derivatives of GNNs with respect to node features have been widely studied in the context of the oversquashing and over-smoothing phenomena, GNN explainability, and more. To date, these derivatives remain unexplored as a means to enhance GNN expressivity. In this paper, we show that these derivatives provide a natural way to enhance the expressivity of GNNs. We introduce High-Order Derivative GNN (HOD-GNN), a novel method that enhances the expressivity of Message Passing Neural Networks (MPNNs) by leveraging high-order node derivatives of the base model. These derivatives generate expressive structure-aware node embeddings processed by a second GNN in an end-to-end trainable architecture. Theoretically, we show that the resulting architecture family's expressive power aligns with the WL hierarchy. We also draw deep connections between HOD-GNN, Subgraph GNNs, and popular structural encoding schemes. For computational efficiency, we develop a message-passing algorithm for computing high-order derivatives of MPNNs that exploits graph sparsity and parallelism. Evaluations on multiple graph learning benchmarks demonstrate HOD-GNN 's excellent performance on popular graph learning tasks.

## 1 Introduction

Graph Neural Networks (GNNs) have become foundational tools in geometric deep learning, with widespread applications in domains such as life sciences (Wong et al., 2024), social sciences (Monti et al., 2019), optimization (Cappart et al., 2023), and more. Despite their empirical success, many GNNs suffer from a fundamental limitation: their expressive power is inherently bounded. In particular, the widely used family of Message Passing Neural Networks (MPNNs) is at most as expressive as the Weisfeiler–Lehman (1-WL) graph isomorphism test Morris et al. (2019); Xu et al. (2018), limiting their ability to distinguish between even simple non-isomorphic graphs and capture intricate structural patterns Chen et al. (2020). To address this shortcoming, a growing body of work has proposed more expressive GNN architectures, typically organized into expressivity hierarchies that balance computational cost with representational power Maron et al. (2019); Morris et al. (2019; 2021).

Concurrently to advances in GNN expressivity, the derivatives of the final node representations $\boldsymbol{h}_v^{(T)}$ [1] and the graph-level output $\boldsymbol{h}^{\text{out}}$ with respect to the initial features $\boldsymbol{X}_v$ have played a key role in several research directions. For over-squashing analysis (Di Giovanni et al., 2023a;b), both first-order derivatives $\frac{\partial \boldsymbol{h}_v^{(T)}}{\partial \boldsymbol{X}_u}$ and mixed partial derivatives $\frac{\partial^2 \boldsymbol{h}^{\text{out}}}{\partial \boldsymbol{X}_v \partial \boldsymbol{X}_u}$ quantify inter-node influence and communication capacity. In over-smoothing studies like Arroyo et al. (2025), derivatives $\frac{\partial \boldsymbol{h}^{\text{out}}}{\partial \boldsymbol{X}_v}$ are used to analyze vanishing gradients, connecting over-smoothing to diminished gradient flow. GNN gradient-based explainability methods (Baldassarre & Azizpour, 2019; Pope et al., 2019) also use these derivatives to identify influential nodes and features. The prevalence of these derivatives across diverse contexts in GNN research suggests they encode valuable information that may be informative for graph learning tasks.

**Our approach.** In this work, we reveal a surprising connection between these two lines of research. We show that incorporating derivatives of a base MPNN with respect to initial node features as additional inputs to a downstream MPNN enhances the expressivity of the base components. One intuitive way to understand this connection is through the mechanism by which GNNs with marking(Papp & Wattenhofer, 2022; Pellizzoni et al., 2024) improve expressivity: they choose a node from the input graph and add to it a unique identifier before processing it through an MPNN. While these identifiers are often implemented through an explicit, often discrete, perturbations to the node features, our approach instead computes derivatives of the MPNN output, capturing the effect of *infinitesimal* perturbations. Thus, giving the model access to derivative information both leverages quantities which frequently arise in theoretical analyses and may thus encode valuable structural information, and yields expressivity gains equivalent to GNNs with marking. See Section 3 for details.

We introduce High-Order Derivative GNN (HOD-GNN), a novel expressive GNN family that leverages the derivatives of a base MPNN to improve its expressive power. We first introduce 1-HOD-GNN, which consists of three components: a base MPNN, a derivative encoder network, and a downstream GNN. 1-HOD-GNN computes high-order derivatives of the base MPNN with respect to the features of a *single* node at a time, i.e., $\frac{\partial^\alpha \boldsymbol{h}_v^{(T)}}{\partial \boldsymbol{X}_u^\alpha}$ and $\frac{\partial^\alpha \boldsymbol{h}^{\text{out}}}{\partial \boldsymbol{X}_u^\alpha}$. These derivatives are then encoded into new derivative-aware node features via the encoder network, which are then passed to the downstream GNN. Theoretically, We show that 1-HOD-GNN models are more expressive than standard GNNs, can compute popular structural encodings, and are tightly related to Subgraph GNNs Cotta et al. (2021); Bevilacqua et al. (2021). Empirically, we demonstrate several desirable properties of our model: it achieves strong performance across a range of standard graph benchmarks, scales to larger graphs that remain out of reach for other expressive GNNs, and can accurately count graph substructures, providing direct empirical evidence of its expressive power.

We then extend 1-HOD-GNN to $k$-HOD-GNN, which supports mixed derivatives with respect to $k$ distinct node features (i.e. $\frac{\partial^{\alpha_1 + \cdots + \alpha_k} \boldsymbol{h}_v^{(T)}}{\partial \boldsymbol{X}_{u_1}^{\alpha_1}, \ldots, \boldsymbol{X}_{u_k}^{\alpha_k}}$, $\frac{\partial^{\alpha_1 + \cdots + \alpha_k} \boldsymbol{h}^{\text{out}}}{\partial \boldsymbol{X}_{u_1}^{\alpha_1}, \ldots, \boldsymbol{X}_{u_k}^{\alpha_k}}$). Like 1-HOD-GNN, the $k$-HOD-GNN forward pass begins by computing higher-order mixed derivatives of a base MPNN, which form a $k$-indexed derivative tensor. This tensor is then used to construct new node features using a higher-order encoder network (as in Maron et al. (2018); Morris et al. (2019)), which are subsequently passed to a downstream GNN for final prediction. We theoretically analyze $k$-HOD-GNN, showing that it can distinguish between graphs that are indistinguishable to the $k$-WL test[2], resulting in a model that is more expressive than any of its individual components alone. Furthermore, we leverage results from Zhang et al. (2024b) to analyze $k$-HOD-GNN's ability to compute homomorphism counts, demonstrating its capacity to capture fine-grained structure.

Efficiently computing high-order node derivatives is a core component of HOD-GNN. To this end, we develop a novel algorithm for computing these derivatives via an analytic, message-passing-like procedure. This approach yields two key benefits. First, being fully analytic, it enables differentiation through the derivative computation itself, allowing

---

[1] $\boldsymbol{h}_v^{(t)}$ is the representation of the node $v$ after the $t$-th GNN layer.

[2] We refer here to the folklore WL test rather than the oblivious variant; see Morris et al. (2023) for a detailed discussion of the differences.

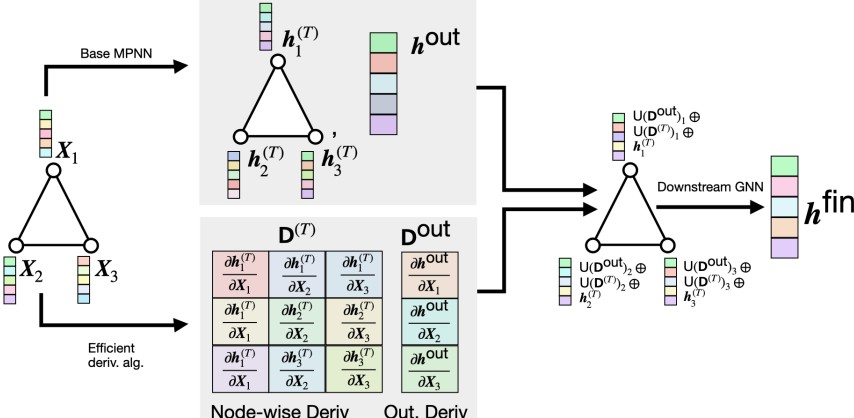

Figure 1: The HOD-GNN pipeline. Given an input graph, we compute the outputs and derivatives of a base MPNN. The derivatives are processed by two encoders (denoted U) to produce features that are concatenated with the base MPNN outputs and passed to a downstream GNN for final prediction.

HOD-GNN to be trained end-to-end. Second, the message-passing-like structure exploits the sparsity of graph data, improving scalability (see Section 4 for a detailed complexity analysis). Combined with the empirical observation that HOD-GNN remains effective even when using base MPNNs with small hidden dimensions, our method scales to benchmarks containing larger graphs that are often out of reach for other expressive GNN architectures.

**Our contributions.** (1) We introduce k-HOD-GNN, a novel expressive GNN family that integrates derivative-based embeddings; (2) We provide a theoretical analysis of its expressivity and computational properties; (3) We propose an algorithm for efficient derivative computation on graphs; (4) We demonstrate consistently high empirical performance across seven standard graph classification and regression benchmarks. Additionally, we show that HOD-GNN scales to benchmarks containing larger graphs that are typically out of reach for many expressive architectures on standard hardware.

## 2 Preliminaries and Previous Work

**Notation.** The size of a set $\mathcal{S}$ is denoted by $|\mathcal{S}|$. $\oplus$ denotes concatenation. We denote graphs by $\mathcal{G} = (\boldsymbol{A}, \boldsymbol{X})$, where $\boldsymbol{A} \in \mathbb{R}^{n \times n}$ is the adjacency matrix and $\boldsymbol{X} \in \mathbb{R}^{n \times d}$ is the node feature matrix, with $n$ nodes and $d$-dimensional features per node. The node set of a graph is denoted by $V(\mathcal{G})$.

**MPNNs and GNN expressivity.** MPNNs(Gilmer et al., 2017) are a widely used class of GNNs that update node representations through iterative aggregation of local neighborhood information. At each layer $t$, the representation $\boldsymbol{h}_v^{(t)}$ of node $v$ is updated via:

$$\boldsymbol{h}_v^{(t)} = \mathsf{MLP}^{(t)} \left( \boldsymbol{h}_v^{(t-1)}, \mathrm{AGG}^{(t)} \left( \left\{ \boldsymbol{h}_u^{(t-1)} : u \in \mathcal{N}(v) \right\} \right) \right), \tag{1}$$

where $\mathcal{N}(v)$ denotes the neighbors of node $v$ in the graph and $\mathrm{AGG}^{(t)}$ are permutation-preserving aggregation function. After $T$ message-passing layers, a graph-level representation is typically obtained by applying a global pooling operation over all node embeddings:

$$\boldsymbol{h}^{\mathrm{out}} = \mathrm{AGG}^{\mathrm{out}} \left( \left\{ \boldsymbol{h}_v^{(T)} \mid v \in V(\mathcal{G}) \right\} \right), \tag{2}$$

MPNNs have inherent expressivity limitations (Morris et al., 2019; Xu et al., 2018; Weisfeiler & Leman, 1968), as they cannot distinguish graphs that are indistinguishable by the 1-WL test. To address this, a wide range of more expressive GNN architectures have been proposed (Morris et al., 2021; Maron et al., 2018; Puny et al., 2023; Cotta et al., 2021; Rieck et al., 2019; Sato et al., 2021; Dwivedi et al., 2023). (See Appendix A for details or (Sato, 2020; Morris et al., 2021; Jegelka, 2022; Li & Leskovec, 2022; Zhang et al., 2024a) for comprehensive surveys.)

**Subgraph GNNs.** Subgraph GNNs (Zhang & Li, 2021; Cotta et al., 2021; Bevilacqua et al., 2021; Frasca et al., 2022; Zhang et al., 2023b;a; Bar-Shalom et al., 2024b) are expressive

GNNs that operate over a set of subgraphs $\mathcal{B}_{\mathcal{G}} = \{\mathcal{S}_{\boldsymbol{v}} \mid \boldsymbol{v} \in V^k(\mathcal{G})\}$, where $V^k(\mathcal{G})$ denotes the set of all $k$-tuples of nodes in the input graph $\mathcal{G}$, and each subgraph $\mathcal{S}_{\boldsymbol{v}}$ corresponds to one such tuple. In this work, we focus on the widely adopted node-marking DS-GNNs (See e.g.

Cotta et al. (2021); Bevilacqua et al. (2021); Papp & Wattenhofer (2022)) and their higher-order generalization, $k$-OSAN (Qian et al., 2022), though we note that many other variants of Subgraph GNNs exist. For precise defintions of DS-GNN and k-OSAN, see Appendix E.1.

**Derivatives of MPNNs.** Derivatives frequently appear in the analysis of GNNs. In the study of *oversquashing*—the failure of information to propagate through graph structures (Alon & Yahav, 2020; Topping et al., 2021; Di Giovanni et al., 2023a;b)—derivatives play a key role (For a comprehensive overview, see Akansha (2023)). Node derivatives are also used in GNN explainability (Ying et al., 2019; Luo et al., 2020; Baldassarre & Azizpour, 2019; Pope et al., 2019). Gradient-based approaches such as Sensitivity Analysis, Guided Backpropagation (Baldassarre & Azizpour, 2019), and Grad-CAM (Pope et al., 2019) rely on derivative magnitudes. Finally, several standalone works make use of node-based derivatives. E.g., Arroyo et al. (2025) use node derivatives to draw a connection between vanishing gradients, and over-smoothing, and Keren Taraday et al. (2024) propose aggregation functions, designed to induce non-zero mixed node derivatives. See Appendix A for further discussion.

## 3 Method

We begin this section with a discussion motivating the use of MPNN derivatives and their contribution to improving expressivity. We then introduce $k$-HOD-GNN, an expressive GNN architecture that enhances representational power by leveraging derivatives of a base MPNN. We first present the full details of the 1-HOD-GNN model, followed by an overview of its higher-order generalization. A comprehensive treatment of the general $k$-order case is provided in Appendix C.1. We emphasize that in $k$-HOD-GNN, the parameter $k$ refers to the number of distinct nodes with respect to which derivatives are taken, not the total derivative order. For instance, 1-HOD-GNN uses derivatives of the form $\frac{\partial^\alpha \boldsymbol{h}_v}{\partial^\alpha \boldsymbol{X}_u}$, but not mixed derivatives such as $\frac{\partial^{\alpha_1+\alpha_2} \boldsymbol{h}_v}{\partial^{\alpha_1} \boldsymbol{X}_{u_1}, \partial^{\alpha_2} \boldsymbol{X}_{u_2}}$, which involve multiple nodes.

### 3.1 Motivation

Beyond being a widely used and informative quantity in GNN analysis, MPNN derivatives can enhance expressivity. To build intuition to why that is, we begin with a simple example, showing that first-order derivatives allow us to count triangles, a task that standard MPNNs cannot perform. Consider the model $\mathcal{M}(\boldsymbol{A}, \boldsymbol{X}) = \boldsymbol{A}^3 \boldsymbol{X}$, which can be implemented by a three-layer GCN with identity activation. For any node $v$, the derivative of its final feature vector $\boldsymbol{h}_v^{(T)}$ with respect to its own input feature vector $\boldsymbol{X}_v$ is exactly $\boldsymbol{A}_{v,v}^3$. aggregating these derivatives, we can compute $\sum_v \boldsymbol{A}_{v,v}^3/6$, which is exactly the number of triangles in the graph.

To illustrate how higher-order derivatives further enhance expressivity, we recall that GNNs with marking (Papp & Wattenhofer, 2022) improve expressive power over standard MPNNs by selecting a node [3] $v$ in the input graph $\mathcal{G} = (\boldsymbol{A}, \boldsymbol{X})$ and attaching a unique identifier to it, yielding the modified input $\boldsymbol{X} + \epsilon \boldsymbol{e}_v$ for some $\epsilon \in \mathbb{R}$. The output is then $\boldsymbol{h}^* = \mathcal{M}(\boldsymbol{A}, \boldsymbol{X} + \epsilon \boldsymbol{e}_v)$. If $\mathcal{M}$ employs an analytic activation function $\sigma$ (See definition E.1 in the appendix), then $\mathcal{M}$ itself is analytic. Consequently, its output can be approximated by the Taylor expansion:

$$\mathcal{M}(\boldsymbol{A}, \boldsymbol{X} + \epsilon \boldsymbol{e}_v) \approx \sum_{i=0}^m \frac{\partial^i \mathcal{M}(\boldsymbol{A}, \boldsymbol{X} + x\boldsymbol{e}_v)}{\partial x^i}\Big|_{x=0} \cdot \epsilon^i = \sum_{i=0}^m \frac{\partial^i \boldsymbol{h}^{\text{out}}}{\partial \boldsymbol{e}_v^i} \cdot \epsilon^i, \tag{3}$$

where $\boldsymbol{h}^{\text{out}} = \mathcal{M}(\boldsymbol{A}, \boldsymbol{X})$ is the output of $\mathcal{M}$ without marking. This shows that by leveraging the higher-order derivatives of an MPNN, one can approximate the output of a GNN with marking to arbitrary precision. As a result, derivatives strictly extend the expressive power of MPNNs. An expanded intuitive discussion of these expressivity gains, along with the natural connection between HOD-GNN and subgraph GNNs, is provided in Appendix B.

---

[3] While GNNs with marking can select multiple nodes, for clarity we focus on the single-node case. The general case is discussed in Appendix B.

## 3.2 The 1-HOD-GNN architecture

**Overview.** A 1-HOD-GNN model, denoted $\Phi$, consists of two GNNs: a base MPNN $\mathcal{M}$ and a downstream GNN $\mathcal{T}$ as well as two derivative encoder networks $\text{U}^{\text{node}}$ and $\text{U}^{\text{out}}$. Given an input graph $\mathcal{G}$, the computation of $\Phi(\mathcal{G})$ proceeds in four steps (see Figure 1): (1) Compute the final node representations and output of $\mathcal{M}$; (2) Compute derivative tensors of the output with respect to the input node features (defined below); (3) Use $\text{U}^{\text{node}}$ and $\text{U}^{\text{out}}$ to extract new derivative informed node features from the derivative tensors. (4) Apply $\mathcal{T}$ to the input graph enriched with derivative-informed features. Importantly, we develop an efficient algorithm for step (2) that enables backpropagation through the derivative computation itself, making all four above steps differentiable (see Section 3.2.1). Consequently, the entire HOD-GNN model can be trained end-to-end, a strategy we adopt in all our experiments.

**Steps 1 & 2.** In the first two stages, we compute the final node representations $\boldsymbol{h}^{(T)}$ and the output vector $\boldsymbol{h}^{\text{out}}$ using the base MPNN $\mathcal{M}$, along with their corresponding derivative tensors, defined below:

**Definition 3.1.** Given a graph $\mathcal{G} = (\boldsymbol{A}, \boldsymbol{X})$ with $n$ nodes, an MPNN $\mathcal{M}$ and an intermediate node feature representation matrix $\boldsymbol{h} \in \mathbb{R}^{n \times d'}$, the derivative tensor of $\mathbf{D}(\boldsymbol{h}) \in \mathbb{R}^{n \times n \times d' \times d \times m}$ is defined by:

$$\mathbf{D}(\boldsymbol{h})[v, u, i, j, \alpha] = \frac{\partial^\alpha \boldsymbol{h}_{v,i}}{\partial \boldsymbol{X}_{u,j}^\alpha}, \tag{4}$$

where $v, u \in V(\mathcal{G})$ are nodes, $i \in [d'], j \in [d]$ specify the feature dimensions of the node feature vectors $\boldsymbol{h}_v, \boldsymbol{X}_u$ respectively, and $\alpha \in [m]$ where $m \in \mathbb{N}$ is a hyperparameter specifying the maximum order of derivatives to be considered. Similarly, given a graph-level prediction vector $\boldsymbol{h}^{\text{out}} \in \mathbb{R}^{d'}$ the derivative tensor $\mathbf{D}(\boldsymbol{h}^{\text{out}}) \in \mathbb{R}^{n \times d' \times d \times m}$ is defined by:

$$\mathbf{D}(\boldsymbol{h}^{\text{out}})[u, i, j, \alpha] = \frac{\partial^\alpha \boldsymbol{h}_i^{\text{out}}}{\partial \boldsymbol{X}_{u,j}^\alpha}. \tag{5}$$

In 1-HOD-GNN, we compute the output derivative tensor $\mathbf{D}^{\text{out}} = \mathbf{D}(\boldsymbol{h}^{\text{out}})$, which captures how the output of the base MPNN $\mathcal{M}$ responds to perturbations in the input node features. In parallel, we compute the node-wise derivative tensor $\mathbf{D}^{(T)}$, where $\mathbf{D}^{(t)} = \mathbf{D}(\boldsymbol{h}^{(t)})$ for $t = 1, \ldots, T$. These tensors characterize how each node's representation at layer $t$ changes in response to variations in the input features. Derivative tensors are computed using Algorithm 1, described in Section 3.2.1 and elaborated on in Appendix D. The algorithm leverages the sparsity of the input graph to enable efficient computation of high-order derivatives. Crucially, Algorithm 1 is fully differentiable with respect to the weights of $\mathcal{M}$, enabling end-to-end training of $\Phi$.

**Step 3.** In the third stage of our method, we extract new node features from the derivative tensors $\mathbf{D}^{\text{out}}$ and $\mathbf{D}^{(T)}$ using the encoder networks $\text{U}^{\text{node}}$ and $\text{U}^{\text{out}}$. First, as $\mathbf{D}^{\text{out}} \in \mathbb{R}^{n \times d' \times d \times m}$ is a tensor indexed by a single node, it can be directly interpreted as a node feature matrix by flattening the remaining dimensions. We thus define the encoder network $\text{U}^{\text{out}}$ to be a DeepSets (Zaheer et al., 2017) update [4]:

$$\text{U}^{\text{out}}(\mathbf{D}^{\text{out}})_v = \text{MLP}(\mathbf{D}^{\text{out}}[v, \ldots]). \tag{6}$$

Secondly, since $\mathbf{D}^{(T)}$ is indexed by pairs of nodes in $\mathcal{G}$, it shares the structure of the adjacency matrix $\boldsymbol{A}$, which is also pairwise-indexed. We can thus define the encoder network $\text{U}^{\text{node}} : \mathbb{R}^{n^2 \times d' \times d \times m} \to \mathbb{R}^{n \times \bar{d}}$ to be any GNN architecture which maps adjacency matrices with edge features to node feature matrices.

To enhance sensitivity to global interactions, we select $\text{U}^{\text{node}}$ to be a 2-Invariant Graph Network (IGN)(Maron et al., 2018), optionally using a sparsity-preserving simplified variant formally defined in Appendix C.1.

---

[4]Empirically, we observe that setting $\text{U}^{\text{out}} = 0$ produces similar results; we nevertheless keep the module in our formulation for completeness.

We construct the derivative-informed node features $\boldsymbol{h}^{\mathrm{der}}$ by combining information from the base MPNN $\mathcal{M}$, the pooled intermediate derivatives, and the output derivatives:

$$\boldsymbol{h}_v^{\mathrm{der}} = \boldsymbol{h}_v^{(T)} \oplus \mathrm{U}^{\mathrm{out}}(\mathbf{D}^{\mathrm{out}}) \oplus \mathrm{U}^{\mathrm{node}}(\mathbf{D}^{(\mathrm{T})}). \tag{7}$$

**Step 4.** In the final stage, we replace the original node features of $\mathcal{G}$ with the derivative-informed features $\boldsymbol{h}^{\mathrm{der}}$, and apply a downstream GNN $\mathcal{T}$ to produce a graph-level prediction. For the remainder of this work, we assume $\mathcal{T}$ is an MPNN, though our approach is compatible with any GNN architecture.

### 3.2.1 Efficient derivative tensor computation

We now describe an efficient algorithm for computing the derivative tensors in a 1-HOD-GNN model $\Phi$ with base MPNN $\mathcal{M}$. For clarity, we focus on the case where $\mathcal{M}$ is a GIN (Xu et al., 2018), in which case the message-passing and readout functions are given by:

$$\boldsymbol{h}_v^{(t)} = \mathsf{MLP}^{(t)}\big((1+\epsilon)\boldsymbol{h}_v^{(t-1)} + \sum_{u \in \mathcal{N}(v)} \boldsymbol{h}_u^{(t-1)}\big), \quad \boldsymbol{h}^{\mathrm{out}} = \mathsf{MLP}\big(\sum_{u \in V(\mathcal{G})} \boldsymbol{h}_u^{(T)}\big). \tag{8}$$

An extension of this algorithm to general MPNNs and higher-order mixed derivatives is provided in Appendix D. For convenience, we decompose the node update in Equation 8 into two parts: an aggregation step and a DeepSets-based update, given respectively by:

$$\overbrace{\tilde{\boldsymbol{h}}_v^{(t-1)} = (1+\epsilon)\boldsymbol{h}_v^{(t-1)} + \sum_{u \in \mathcal{N}(v)} \boldsymbol{h}_u^{(t-1)}}^{\text{agg. update}}, \quad \overbrace{\boldsymbol{h}_v^{(t)} = \mathsf{MLP}(\tilde{\boldsymbol{h}}_v^{(t-1)})}^{\text{DeepSets update}}. \tag{9}$$

The algorithm is based on the following two observations: First since $\tilde{\boldsymbol{h}}_v^{(t-1)}$ is a linear combination of $\boldsymbol{h}_v^{(t-1)}$ and its neighboring node features $\{\boldsymbol{h}_u^{(t-1)} \mid u \in \mathcal{N}(v)\}$, the derivatives of $\tilde{\boldsymbol{h}}_v^{(t-1)}$ are likewise linear combinations of the derivatives of $\boldsymbol{h}_v^{(t-1)}$ and $\{\boldsymbol{h}_u^{(t-1)} \mid u \in \mathcal{N}(v)\}$. More explicitly:

$$\mathbf{D}(\tilde{\boldsymbol{h}}^{(t-1)})[v, \dots] = (1+\epsilon)\mathbf{D}^{(t-1)}[v, \dots] + \sum_{u \in \mathcal{N}(v)} \mathbf{D}^{(t-1)}[u, \dots]. \tag{10}$$

This computation mirrors the GIN aggregation update in Equation 8, leveraging the sparsity of the graph. Second, since the DeepSets update applies an MLP independently to each node feature $\tilde{\boldsymbol{h}}_v^{(t-1)}$, we can apply Faà di Bruno's formula (see e.g. (Hardy, 2006)) to compute the derivatives of each $\boldsymbol{h}_v^{(t)}$ based on the derivatives of $\tilde{\boldsymbol{h}}_v^{(t-1)}$. This results in a "DeepSets-like" derivative update, allowing us to compute $\mathbf{D}^{(t)}$ directly from $\mathbf{D}(\tilde{\boldsymbol{h}}^{(t-1)})$. Iteratively applying these two steps yields the final node-wise derivative tensor $\mathbf{D}^{(T)}$ through a differentiable, message-passing-like procedure. A similar approach allows efficient computation of $\mathbf{D}^{\mathrm{out}}$ from $\mathbf{D}^{(T)}$. See Appendix D for full details of the algorithm.

**Computational Complexity.** An important property of the above algorithm is that, for sparse graphs or relatively shallow base MPNNs, it is computationally efficient. To see this, first notice that since $\boldsymbol{h}^{(0)} = \boldsymbol{X}$, the tensor $\mathbf{D}^{(0)}$ is extremely sparse, satisfying:

$$\mathbf{D}^{(0)}[v, u, i, j, \alpha] = \begin{cases} 1 & \text{if } v = u, \ i = j, \ \alpha = 1, \\ 0 & \text{otherwise.} \end{cases} \tag{11}$$

Thus, it can be stored efficiently using sparse matrices. At each layer $t$, the derivative aggregation step (Equation 10) increases the number of non-zero entries only in proportion to the number of node pairs that exchange messages for the first time. Thus, for small values of $t$ or for sparse graphs $\mathcal{G}$, the derivative tensor $\mathbf{D}^{(t)}$ remains sparse. Moreover, the algorithm's message-passing-like structure ensures runtime efficiency as well. For a full complexity analysis of our algorithm, see Section 4.

### 3.3 k-HOD-GNN via mixed derivatives

We now generalize 1-HOD-GNN, which operates on single-node derivatives (i.e., derivatives of the form $\frac{\partial^\alpha \boldsymbol{h}_v^{(T)}}{\partial \boldsymbol{X}_u^\alpha}$ or $\frac{\partial^\alpha \boldsymbol{h}^{\mathrm{out}}}{\partial \boldsymbol{X}_u^\alpha}$), to $k$-HOD-GNN, which extracts information from mixed

partial derivatives across $k$ nodes (i.e., $\frac{\partial^{\alpha_1 + \cdots + \alpha_k} \boldsymbol{h}_v^{(T)}}{\partial \boldsymbol{X}_{u_1}^{\alpha_1} \cdots \partial \boldsymbol{X}_{u_k}^{\alpha_k}}$, or $\frac{\partial^{\alpha_1 + \cdots + \alpha_k} \boldsymbol{h}^{\text{out}}}{\partial \boldsymbol{X}_{u_1}^{\alpha_1} \cdots \partial \boldsymbol{X}_{u_k}^{\alpha_k}}$). $k$-HOD-GNN offers increased expressive power at the cost of greater computational complexity. We begin by formally defining the $k$-indexed derivative tensors. For simplicity, we assume the input node features are 1-dimensional, handling the more general case in Appendix C.1.

**Definition 3.2.** Given a graph $\mathcal{G} = (\boldsymbol{A}, \boldsymbol{X})$ with $n$ nodes, and an MPNN $\mathcal{M}$ and an intermediate node feature matrix $\boldsymbol{h} \in \mathbb{R}^{n \times d'}$, the $k$-indexed derivative tensor of $\mathbf{D}_k(\boldsymbol{h}) \in \mathbb{R}^{n \times n^k \times d' \times m^k}$ is defined by:

$$\mathbf{D}_k(\boldsymbol{h})[v, \boldsymbol{u}, i, \boldsymbol{\alpha}] = \frac{\partial^{\alpha_1 + \cdots + \alpha_k} \boldsymbol{h}_{v,i}}{\partial \boldsymbol{X}_{u_1}^{\alpha_1} \cdots \partial \boldsymbol{X}_{u_k}^{\alpha_k}}. \tag{12}$$

where $v \in V(\mathcal{G})$, $\boldsymbol{u} = (u_1, \ldots, u_k) \in V^k(\mathcal{G})$, $i \in [d']$ and $\boldsymbol{\alpha} = (\alpha_1, \ldots, \alpha_k) \in [m]^k$. $\mathbf{D}_k$ is defined similarly for graph-level prediction vectors.

The $k$-indexed derivative tensors capture how the output and node representations of $\mathcal{M}$ change under joint perturbations to the features of $k$ nodes, thereby encoding rich higher-order structural interactions within the graph. We compute these tensors using an extension of the derivative computation process described in Section 3.2.1 (See Appendix D for more details).

As in the 1-HOD-GNN case, a $k$-HOD-GNN model $\Phi$ consists of a base MPNN $\mathcal{M}$ and a downstream network $\mathcal{T}$. Given an input graph $\mathcal{G}$, the computation of $\Phi(\mathcal{G})$ proceeds in the same four stages established earlier. First, we compute the output and final node representations of $\mathcal{M}$ along with the $k$-indexed derivative tensors $\mathbf{D}_k^{(T)}$ and $\mathbf{D}_k^{\text{out}}$. We then use $\boldsymbol{h}^{(T)}$, $\mathbf{D}_k^{(T)}$ and $\mathbf{D}_k^{\text{out}}$ to extract new derivative informed node features. These node features are computed through:

$$\boldsymbol{h}_v^{\text{der}} = \mathsf{MLP}\left(\boldsymbol{h}_v^{(T)} \oplus \mathrm{U}^{\text{node}}(\mathbf{D}_k^{(T)})_v \oplus \mathrm{U}^{\text{out}}(\mathbf{D}_k^{\text{out}})_v\right). \tag{13}$$

where $\mathrm{U}^{\text{node}}$ and $\mathrm{U}^{\text{out}}$ are learned $(k+1)$-IGN and $k$-IGN encoders respectively. Finally, we substitute the original node features of graph $\mathcal{G}$ with $\boldsymbol{h}^{\text{der}}$, and input the resulting graph into $\mathcal{T}$ to generate the final graph-level prediction. For more details, see Appendix C.1.

## 4 THEORETICAL ANALYSIS

In this section, we analyze the expressive power and computational complexity of $k$-HOD-GNN. Formal statements and complete proofs of all results in this section are provided in Appendix E.2.

**Expressive power.** We begin by formally relating $k$-HOD-GNN to both $k$-OSAN subgraph GNNs as well as $(k+2)$-IGNs, revealing new insights into HOD-GNNs' expressive power, and their position in the WL hierarchy.

**Theorem 4.1** (informal). *Any $k$-OSAN model can be approximated by a $k$-HOD-GNN model using an analytic activation function, to any precision. Additionally, any $k$-HOD-GNN model can be approximated by a $(k+2)$-IGN model.*

**Corollary 4.2.** *There exist non-isomorphic graphs that are indistinguishable by the folklore $k$-WL ($k$-FWL) test but are distinguishable by $k$-HOD-GNN. Additionally, any pair of graphs that is indistinguishable by the $(k+1)$-FWL test is also indistinguishable by $k$-HOD-GNN.*

The proof of Theorem 4.1 relies on the analyticity of the activation functions used by our base MPNN and the use of higher-order derivatives. However, in what follows, we show that even when restricted to first-order derivatives and using the commonly employed ReLU activation, HOD-GNN remains strictly more expressive than a widely used technique for enhancing GNN expressivity: incorporating Random Walk Structural Encodings (RWSEs)(Dwivedi et al., 2021) into a base MPNN.

**Theorem 4.3** (Informal). *Even when limited to first-order derivatives and ReLU activations, 1-HOD-GNN is strictly more expressive than MPNNs enhanced with random walk structural encodings.*

The first part of Theorem 4.3 is constructive: it shows that a simple initialization of the base MPNN's weights yields derivatives equal to RWSEs. In our experiments, we use a slightly mod-

ified version of this initialization (see Appendix G), allowing HOD-GNN to serve as a learnable extension of RWSE. Further analysis of the expressive power of HOD-GNN when using edge-feature derivatives, or when only using output-level derivatives are presented in Appendix F.

**Space and time complexity.** To conclude this section, we analyze the computational complexity of $k$-HOD-GNN and compare it to other expressive architectures, namely, $(k+1)$-IGN and $k$-OSAN. We show that $k$-HOD-GNN achieves better complexity when using relatively shallow base MPNNs, while maintaining comparable complexity with deeper ones. The primary source of computational overhead in $k$-HOD-GNN lies in the derivative tensor computation, and the encoder network forward pass. We now analyze each of these components.

First, while $k$-HOD-GNN computes derivative tensors $\mathbf{D}_k^{(t)}$ with $O(n^{k+1})$ potential entries, these tensors are sparse for relatively shallow base MPNNs. Moreover, each $\mathbf{D}_k^{(t)}$ can be efficiently computed from $\mathbf{D}_k^{(t-1)}$. This is formalized in the following proposition:

**Proposition 4.4.** *In a $k$-HOD-GNN model applied to a graph with $n$ nodes and maximum degree $d$ The number of non-zero entries in $\mathbf{D}^{(t)}$ is at most $O\left(n \cdot \min\{n^k, d^{k \cdot t}\}\right)$. Additionally, each $\mathbf{D}^{(t)}$ can be computed from $\mathbf{D}^{(t-1)}$ in time $O(d \cdot n \cdot \min\{n^k, d^{k \cdot (t-1)}\})$.*

Focusing next on the encoder networks, we show that they can be designed to exploit derivative sparsity for improved efficiency, while retaining the full expressivity of the $k$-OSAN architecture:

**Proposition 4.5.** *In a $k$-HOD-GNN model, the encoder functions $U^{node}$ and $U^{out}$ can be chosen such that the model retains the expressive power of $k$-OSAN, while the computation of $U^{node}(\mathbf{D}^{(T)})$ and $U^{out}(\mathbf{D}^{out})$ has both time and space complexity $O\left(n \cdot \min\{n^k, d^{k \cdot T}\}\right)$.*

Propositions 4.4 and 4.5 suggest that a $k$-HOD-GNN model with a base MPNN of depth $T$ has space complexity $O\left(n \cdot \min\{n^k, d^{k \cdot T}\}\right)$ and time complexity $O\left(d \cdot n \cdot \min\{n^k, d^{k \cdot (T-1)}\}\right)$. In comparison, $k$-OSAN has space complexity $O(n^{k+1})$ and time complexity $O(d \cdot n^{k+1})$, while $(k+1)$-IGN incurs both time and space complexity of $O(n^{k+1})$. Assuming the input graph is sparse (i.e., $d \ll n$), $k$-HOD-GNN is more efficient then $k$-OSAN and $(k+1)$-IGN when the base MPNN is shallow ($d^T < n$), while all three models have comparable complexity when the base MPNN is deep ($d^T > n$).

## 5 EXPERIMENTS

Our experimental study is designed to validate the theoretical arguments from the previous section and to address the following guiding questions: **(Q1)** How does HOD-GNN perform on real-world datasets when compared against strong GNN baselines? **(Q2)** Can HOD-GNN scale to larger graphs that are beyond the reach of Subgraph GNNs, and how does it perform in this regime? **(Q3)** How does the expressive power of HOD-GNN compare with natural and widely used GNN baselines? We evaluate HOD-GNN across eight benchmarks, with additional experimental details provided in Appendix G.

**Baselines.** We compare HOD-GNN against strong representatives from three natural families of GNNs. First, motivated by its connection to positional/structural encodings (PSEs, Section 4), we consider **encoding-augmented MPNNs**, including Laplacian PEs (Dwivedi et al., 2023), RWSEs (Dwivedi et al., 2021), SignNet (Lim et al., 2022), random node identifiers (Abboud et al., 2020; Sato et al., 2021), as well as recent methods such as GPSE (Cantürk et al., 2024) and MOSE (Bao et al., 2024). Second, since HOD-GNN is theoretically related to **Subgraph GNNs**, we compare with representative models like GNN-AK (Zhao et al., 2022), SUN (Frasca et al., 2022), and Subgraphormer (Bar-Shalom et al., 2023). Because such models often struggle to scale, we also include sampling-based variants such as Policy-Learn (Bevilacqua et al., 2024), HyMN (Southern et al., 2025), and Subgraphormer with random sampling. Finally, we benchmark against widely used and modern **general-purpose GNNs**, including GIN (Xu et al., 2018), GCN (Kipf & Welling, 2016), GatedGCN (Bresson & Laurent, 2017), GPS (Rampášek et al., 2022), and GraphViT (He et al., 2023). Across experiments we include representatives from each family, while the

Table 1: Performance on OGB and ZINC datasets (4 seeds). **First** and **second** best scores are highlighted. Scores sharing a color are not statistically distinguishable based on Welch's t-test with a relaxed threshold of $p < 0.2$. "–" denotes results not previously reported, and "x" indicates that digits beyond this point were not provided.

| Method ↓ / Dataset → | ZINC-12K (MAE ↓) | MOLTOX21 (ROC-AUC ↑) | MOLBACE (ROC-AUC ↑) | MOLHIV (ROC-AUC ↑) |
|---|---|---|---|---|
| **Common Baselines** | | | | |
| GCN (Kipf & Welling, 2016) | 0.321±0.009 | 75.29±0.69 | 79.15±1.44 | 76.06±0.97 |
| GIN (Xu et al., 2018) | 0.163±0.004 | 74.91±0.51 | 72.97±4.00 | 75.58±1.40 |
| PNA (Corso et al., 2020) | 0.761±0.002 | 73.30±1.1x | – | 79.05±1.32 |
| GPS (Rampášek et al., 2022) | 0.070±0.004 | 75.70±0.40 | – | 78.80±1.01 |
| GraphViT (He et al., 2023) | 0.085±0.005 | **78.51±0.77** | – | 77.92±1.49 |
| **Subgraph GNNs** | | | | |
| Reconstr. GNN (Cotta et al., 2021) | – | 75.15±1.40 | – | 76.32±1.40 |
| GNN-AK+ (Zhao et al., 2022) | 0.091±0.011 | – | – | **79.61±1.19** |
| SUN (EGO+) (Frasca et al., 2022) | 0.084±0.002 | – | – | **80.03±0.55** |
| Full (Bevilacqua et al., 2024) | 0.087±0.003 | 76.25±1.12 | 78.41±1.94 | 76.54±1.37 |
| OSAN (Qian et al., 2022) | 0.177±0.016 | – | 72.30±6.60 | – |
| Random (Bevilacqua et al., 2024) | 0.102±0.003 | 76.62±0.63 | 78.14±2.36 | 77.30±2.56 |
| Policy-Learn (Bevilacqua et al., 2024) | 0.097±0.005 | **77.36±0.60** | 78.39±2.28 | 78.49±1.01 |
| Subgraphormer (Bar-Shalom et al., 2024a) | **0.063±0.001** | – | **84.35±0.65** | 79.58±0.35 |
| HyMN (Southern et al., 2025) | 0.080±0.003 | **77.82±0.59** | **81.16±1.21** | **81.01±1.17** |
| **PSEs** | | | | |
| GIN + Laplacian PE (Dwivedi et al., 2023) | 0.162±0.014 | 76.60±0.3x | 80.40±1.5x | 75.60±1.1x |
| GIN + RWSE (Dwivedi et al., 2021) | 0.128±0.005 | 76.30±0.5x | 79.60±2.8x | 78.10±1.5x |
| SignNet (Lim et al., 2022) | 0.102±0.002 | – | – | – |
| RNI (Abboud et al., 2020) | 0.136±0.0070 | – | 61.94±2.51 | 77.74±0.98 |
| GSN (Bouritsas et al., 2022) | 0.101±0.010 | 76.08±0.79 | 77.40±2.92 | **80.39±0.90** |
| ENGNN (Wang & Zhang, 2025) | 0.114±0.005 | – | – | 78.51±0.86 |
| GPSE (Cantürk et al., 2024) | **0.065±0.003** | **77.40±0.8x** | **80.80±3.1x** | 78.15±1.33 |
| MOSE (Bao et al., 2024) | **0.062±0.002** | – | – | – |
| **Ours** | | | | |
| HOD-GNN | **0.0666±0.0035** | **77.99±0.71** | **82.10±1.45** | **80.86±0.52** |

specific choice of baselines in each task reflects relevance and standard practice in prior work. Throughout the paper, we report results directly from prior work and include any relevant baseline, even if values for some of the benchmarks were not reported. This allows for a broad and fair comparison rather than excluding useful baselines. Missing entries are marked by "–".

**OGB and ZINC.** To evaluate HOD-GNN's real-world performance **(Q1)**, we benchmark it on standard graph property prediction datasets: ZINC (Irwin et al., 2012) for regression, and three molecular classification tasks from the OGB suite (Hu et al., 2020b)—`molhiv`, `molbace`, and `moltox21`. These benchmarks provide standardized splits and are the de facto choice for assessing GNN performance. As shown in Table 1, HOD-GNN delivers excellent results across all tasks, standing out as the only model that consistently ranks within the top two tiers.

**Peptides.** Section 4 established that HOD-GNN has improved computational complexity compared to Subgraph GNNs. To demonstrate its scalability in practice **(Q2)** and to further assess its performance on real-world data **(Q1)**, we evaluate HOD-GNN on the `Peptides` datasets from the LRGB benchmark (Dwivedi et al., 2022), where the goal is to predict global structural and functional properties of peptides represented as graphs. As stated in prior work (Southern et al., 2025; Bar-Shalom et al., 2023), full-bag Subgraph GNNs cannot process these graphs directly using standard hardware, requiring the use of subsampling strategies that can reduce expressivity and introduce optimization challenges due to randomness. In contrast, HOD-GNN handles these graphs directly without subsampling. As shown in Table 2, HOD-GNN surpasses all sampling-based Subgraph GNNs and is the only model that consistently ranks within the top two tiers, underscoring both its scalability and effectiveness on challenging real-world molecular tasks.

**Synthetic experiments.** To evaluate the realized expressiveness of HOD-GNN **(Q3)** and empirically support Theorems 4.1 and 4.3, we conduct two synthetic studies. First, following the protocol of Huang et al. (2022), we assess the ability of 1-HOD-GNN to learn to count small substructures, a standard proxy for practical GNN expressivity (Bouritsas et al., 2022; Arvind et al., 2020). Theorems 4.1 and 4.3 predict that: (i) with analytic activations, 1-HOD-GNN matches the power of certain Subgraph GNNs, and (ii) with ReLU, it is strictly stronger than MPNNs with RWSEs. Table 6 (Appendix H) confirms both predictions. Additional details are provided in Appendix H.1.

We additionally test 1- and 2-HOD-GNN on the regular graph pairs in the BREC dataset (Wang & Zhang, 2024), which include 50 pairs separable by 3-WL but not 2-WL and 90 pairs indistinguishable even by 3-WL. Table 7 shows that 2-HOD-GNN separates 34/90 of the 3-WL-indistinguishable pairs, placing it among the strongest models and empirically validating the theoretical advantage predicted by Theorem 4.1. Additionally, 1-HOD-GNN performs similarly to DS-GNN, consistent with Theorem 4.1. Additional details are provided in Appendix H.2.

**Additional ablations and empirical insights.** Appendix H includes further ablations and analysis. We first evaluate how the hyperparameter $m$ from Definition 3.1, which sets the maximum derivative order used in HOD-GNN, affects expressive power. Results show that increasing $m$ consistently strengthens expressivity, until the performance gains eventually plateau. This suggests that small values such as $m \in \{2, 3, 4\}$ are already effective in practice. We also analyze the stability of HOD-GNN, showing that for derivative orders $m = 1, \ldots, 4$, the training curves remain stable and the norms of the derivative tensors stay well-behaved relative to the final node-feature norms produced by the base MPNN. Finally, Inspired by recent work linking overly expressive GNN expressivity to poor generalization (Franks et al., 2024; Maskey et al., 2025; Carrasco et al., 2025), we evaluate HOD-GNN 's train–test gaps on OGB datasets. HOD-GNN consistently exhibits smaller gaps than both less expressive (e.g., GIN, GCN) and more expressive (e.g., DSS-GNN (ED)) baselines, indicating strong generalization without overfitting. See Appendix H.5 for details.

Table 2: Performance on PEPTIDES (4 seeds). **First** and second best scores are highlighted. Same color scores are not statistically distinguishable based on Welch's t-test with a relaxed threshold of $p < 0.2$.

| Model | Peptides-func AP ↑ | Peptides-struct MAE ↓ |
|---|---|---|
| **Common Baselines** | | |
| GCN | 59.30±0.23 | 0.3496±0.0013 |
| GINE | 54.98±0.79 | 0.3547±0.0045 |
| GCNII | 55.43±0.78 | 0.3471±0.0010 |
| GatedGCN | 58.64±0.77 | 0.3420±0.0013 |
| DIGL+MPNN+LapPE | 68.30±0.26 | 0.2616±0.0018 |
| MixHop-GCN+LapPE | 68.43±0.49 | 0.2614±0.0023 |
| DRew-GCN+LapPE | 71.50±0.44 | 0.2536±0.0015 |
| SAN+LapPE | 63.84±1.21 | 0.2683±0.0043 |
| GraphGPS+LapPE | 65.35±0.41 | 0.2500±0.0005 |
| Exphormer | 65.27±0.43 | 0.2481±0.0007 |
| GraphViT | 69.19±0.85 | 0.2474±0.0016 |
| **Subgraph GNNs** | | |
| Policy-Learn | 64.59±0.18 | **0.2475±0.0011** |
| Subgraphormer 30% | 64.15±0.52 | 0.2494±0.0020 |
| HyMN | 68.57±0.55 | **0.2464±0.0013** |
| **PSEs** | | |
| GCN + Laplacian PE | 62.18±0.55 | 0.2492±0.0019 |
| GCN + RWSE | 60.67±0.69 | 0.2574±0.0020 |
| SignNet | 63.14±0.59 | – |
| GPSE + GCN | 63.16±0.85 | **0.2487±0.0011** |
| GPSE + GPS | 66.88±1.51 | **0.2464±0.0025** |
| MOSE | 63.5x±1.1x | 0.318±0.010 |
| **Ours** | | |
| HOD-GNN | **69.68±0.56** | **0.2450±0.0011** |

**Summary.** Across all experiments, we find consistent evidence supporting the guiding questions outlined above. **(A1)** Across the ZINC, OGB, and Peptides datasets, HOD-GNN is the only architecture that consistently ranks within the top two tiers. **(A2)**, On the challenging `Peptides` datasets, HOD-GNN scales to larger graphs that full-bag Subgraph GNNs cannot handle, while maintaining strong predictive performance. **(A3)** On several synthetic experiments, 1-HOD-GNN matches the expressivity of Subgraph GNNs while surpassing encoding-augmented MPNNs, showing that HOD-GNN is **as expressive yet more scalable** than subgraph GNNs. Moreover, 2-HOD-GNN exhibits even stronger expressive power, demonstrating the benefits of increasing derivative order. Together, these findings establish HOD-GNN as a scalable, expressive, and broadly effective GNN framework across synthetic, molecular, and large-scale real-world tasks.

## 6 CONCLUSION

We introduce HOD-GNN, a GNN that enhances the expressivity of a base MPNN by leveraging its high-order derivatives. We provide a theoretical analysis of HOD-GNN's expressive power, connecting it to the $k$-OSAN framework and RWSE encodings, and show that it can offer better scalability than comparable expressive models. Empirically, HOD-GNN has strong performance across a range of benchmarks, matching or surpassing encoding based methods, Subgraph GNNs, and other common baselines.

**Limitations and Future Work.** Although HOD-GNN has favorable theoretical complexity (Section 4), its practical efficiency depends on sparse matrix operations, and more optimized implementations could significantly improve scalability. Additionally, the connection between MPNN derivatives and oversquashing or oversmoothing, alongside HOD-GNN's strong performance with deep base MPNNs and small hidden dimensions, suggests a deeper link not fully addressed in this work. Exploring this connection is a promising direction for future research.

## Acknowledgements

HM is supported by the Israel Science Foundation through a personal grant (ISF 264/23) and an equipment grant (ISF 532/23), and by the Career Advancement Chairs in Artificial Intelligence – Schmidt Futures. ME acknowledges support from the Israeli Ministry of Innovation, Science and Technology. YG is supported by the UKRI Engineering and Physical Sciences Research Council (EPSRC) CDT in Autonomous and Intelligent Machines and Systems (grant reference EP/S024050/1). F.F. conducted this work supported by an Aly Kaufman Post-Doctoral Fellowship. G.B. is supported by the Jacobs Qualcomm PhD Fellowship.

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

# A   Previous Work

**Expressive Power and Hierarchies in GNNs.** The expressive power of GNNs is often measured by their ability to distinguish non-isomorphic graphs. Foundational results (Morris et al., 2019; Xu et al., 2018) show that standard MPNNs are bounded by the 1-Weisfeiler-Lehman (1-WL) test (Weisfeiler & Leman, 1968), motivating the development of more expressive architectures, see (Sato, 2020; Morris et al., 2021; Jegelka, 2022; Li & Leskovec, 2022; Zhang et al., 2024a) for comprehensive surveys. For instance, Morris et al. (2019) and Maron et al. (2018) introduced GNN hierarchies matching the expressivity of the $k$-WL test at a computational cost of $\mathcal{O}(n^k)$ in both time and memory. Other approaches include equivariant polynomial-based models (Maron et al., 2019; Puny et al., 2023), Subgraph GNNs (Zhang & Li, 2021; Cotta et al., 2021; Bevilacqua et al., 2021; Frasca et al., 2022; Zhang et al., 2023b;a; Bar-Shalom et al., 2024b) topologically enhanced GNNs (Rieck et al., 2019; Bodnar et al., 2021; Eitan et al., 2024) and more. A complementary line of work improves expressivity by enriching node features with informative structural descriptors, such as substructure and homomorphism counts (Bouritsas et al., 2022; Bao et al., 2024), random node features (Abboud et al., 2020; Sato et al., 2021; Eliasof et al., 2023), or spectral methods (Dwivedi et al., 2023; Lim et al., 2022).

**Derivatives of GNNs.** Derivatives frequently appear in the analysis of GNNs. A prominent example is the study of *oversquashing*-the failure of information to propagate through graph structures (Alon & Yahav, 2020; Topping et al., 2021; Di Giovanni et al., 2023a;b). A central tool in analyzing oversquashing is the use of derivatives: specifically, the gradients of final node representations with respect to initial features (see Di Giovanni et al., 2023a), and mixed output derivatives with respect to pairs of input nodes (see Di Giovanni et al., 2023b). For a comprehensive overview of oversquashing, see Akansha (2023). Node derivatives also play a key role in GNN explainability (Ying et al., 2019; Luo et al., 2020; Baldassarre & Azizpour, 2019; Pope et al., 2019). Gradient-based approaches such as Sensitivity Analysis, Guided Backpropagation (Baldassarre & Azizpour, 2019), and Grad-CAM (Pope et al., 2019) rely on derivative magnitudes to compute importance scores. Finally, several standalone works make use of node-based derivatives. For instance, Arroyo et al. (2025) use node derivatives to draw a connection between vanishing gradients, and over-smoothing. In a different direction, Keren Taraday et al. (2024) propose new aggregation functions for MPNNs, designed specifically to induce non-zero mixed node derivatives.

**Learning over derivative input.** Beyond GNNs, several recent works have explored learning directly from derivative-based inputs. Xu et al. (2022) propose a framework that processes spatial derivatives of implicit neural representations (INRs) to modify them without explicit decoding. Mitchell et al. (2021) introduce a learned method for fact editing in LLMs using their gradients. Gelberg et al. (2025) present a general architecture for learning over sets of gradients, with applications in meta-optimization, domain adaptation, and curvature estimation.

# B   Motivation

Beyond being a widely used and informative quantity in GNN analysis, MPNN derivatives can enhance expressivity. We now provide an intuition for why this occurs by drawing a connection to Subgraph GNNs.

Consider a DS-GNN model $\Psi$ composed of a base MPNN $\mathcal{M}$ followed by a downstream MPNN $\mathcal{T}$. We assume for simplicity that $\mathcal{M}$ outputs graph-level scalars, and that the activation function $\sigma$ used in $\mathcal{M}$ is *analytic* with infinite convergence radius[5]. That is, for every $x \in \mathbb{R}$, $\sigma$ satisfies:

$$\sigma(x) = \sum_{\alpha=0}^{\infty} \frac{\sigma^{(\alpha)}(0)}{\alpha!} x^{\alpha}. \tag{14}$$

---

[5]Many commonly used functions, including sin, cos, and exp, are analytic with infinite convergence radius. See Appendix E.2 for a discussion of the case where $\sigma$ is not analytic.

For each node $v$ of a given input graph $\mathcal{G}$, we define a function $f_v : \mathbb{R} \to \mathbb{R}$ by:

$$f_v(x) := \mathcal{M}(\boldsymbol{A}, \boldsymbol{X} \oplus x \cdot \boldsymbol{e}_v), \tag{15}$$

which corresponds to the ouptput of $\mathcal{M}$ obtained by scaling the node marking feature at node $v$ by $x$. Observe that $f_v$ is analytic[6] since $\mathcal{M}$ is composed of analytic functions, and that by definition, $f_v(1) = \boldsymbol{h}_v^{\mathrm{sub}}$, i.e., the representation of the graph augmented with a mark for the node $v$. Also observe that by expanding $f_v$ around $x = 0$, we can approximate $\boldsymbol{h}_v^{\mathrm{sub}}$ to any desired precision using a **finite number** of derivatives, up to order $m$:

$$f_v(1) \approx \sum_{\alpha=0}^{m} \frac{f_v^{(\alpha)}(0)}{\alpha!}. \tag{16}$$

Moreover, each derivative $f_v^{(\alpha)}(0)$ corresponds to a partial derivative of the output of $\mathcal{M}$ with respect to the $v$-th coordinate of the augmented input:

$$f_v^{(\alpha)}(0) = \left.\frac{\partial^\alpha \mathcal{M}(\boldsymbol{A}, \boldsymbol{X} \oplus x \cdot \boldsymbol{e}_v)}{\partial^\alpha x}\right|_{x=0} = \frac{\partial^\alpha \boldsymbol{h}^{\mathrm{out}}}{\partial^\alpha \boldsymbol{e}_v}. \tag{17}$$

This suggests that by using an encoder network to extract node features from the first $m$ derivatives of $\mathcal{M}$ with respect to each node feature, we can effectively reconstruct $\boldsymbol{h}_v^{\mathrm{sub}}$. By passing these derivative-based features to the downstream GNN $\mathcal{T}$, we can approximate the behavior of $\Psi$, and therefore be at least as expressive.

## C   HOD-GNN VARIANTS

### C.1   κ-HOD-GNN

In this section, we elaborate on the $k$-HOD-GNN architecture, discussed in Section 3.3. Similar to 1-HOD-GNN, a $k$-HOD-GNN model, denoted $\Phi$, consists of a base MPNN $\mathcal{M}$, two derivative encoders $\mathrm{U}^{\mathrm{node}}, \mathrm{U}^{\mathrm{out}}$, and a downstream network $\mathcal{T}$. Given an input graph $\mathcal{G} = (\boldsymbol{A}, \boldsymbol{X})$, the computation of $\Phi(\mathcal{G})$ proceeds in four stages: (1) compute the output and final node representation of the input graph using $\mathcal{M}$; (2) compute the k-indexed derivative tensors $\mathbf{D}_k^{(T)}, \mathbf{D}_k^{\mathrm{out}}$; (3) extract new derivative informed node features form the derivative tensor; (4) Use these new node features for downstream processing.

**Steps 1 & 2.** In the first two stages, , we compute the final node representations $\boldsymbol{h}^{(T)}$ and the output vector $\boldsymbol{h}^{\mathrm{out}}$ using the base MPNN $\mathcal{M}$, along with their corresponding derivative tensors defined below (the $k$-indexed derivative tensor was defined in Section 3.3 for he case where the input features are 1-dimensional).

**Definition C.1.** Given a graph $\mathcal{G} = (\boldsymbol{A}, \boldsymbol{X}_0)$ with $n$ nodes, $\boldsymbol{X}_0 \in \mathbb{R}^{n \times d}$, an MPNN $\mathcal{M}$ and some intermediate node feature representation matrix $\boldsymbol{h} \in \mathbb{R}^{n \times d'}$, the $k$-indexed derivative tensor of $\mathbf{D}_k(\boldsymbol{h}) \in \mathbb{R}^{n \times n^k \times d' \times m^{d \times k}}$ is defined by:

$$\mathbf{D}_k(\boldsymbol{h})[v, \boldsymbol{u}, i, \boldsymbol{\alpha}] = \partial^{\boldsymbol{\alpha}} \boldsymbol{h}_{v,i}(\boldsymbol{X}_0) = \left.\frac{\partial^{|\boldsymbol{\alpha}|} \boldsymbol{h}_{v,i}}{\prod_{(j_1, j_2) \in [d] \times [k]} \partial \boldsymbol{X}_{u_1, j_1}^{\boldsymbol{\alpha}_{j_1, j_2}}}\right|_{\boldsymbol{X} = \boldsymbol{X}_0}. \tag{18}$$

where $v \in V(\mathcal{G})$, $\boldsymbol{u} = (u_1, \ldots, u_k) \in V^k(\mathcal{G})$ , $i \in [d'], \boldsymbol{\alpha} = (\boldsymbol{\alpha}_{j_1, j_2})_{d,k} \in \{0, \ldots, m-1\}^{d \times k}$ and $|\boldsymbol{\alpha}| = \sum \boldsymbol{\alpha}_{j_1, j_2}$.

Similarly, given a graph-level prediction vector $\boldsymbol{h}^{\mathrm{out}} \in \mathbb{R}^{d'}$ the derivative tensor $\mathbf{D}_k(\boldsymbol{h}^{\mathrm{out}}) \in \mathbb{R}^{n^k \times d' \times d^k \times m^k}$ is defined by:

$$\mathbf{D}_k(\boldsymbol{h})[\boldsymbol{u}, i, \boldsymbol{\alpha}] = \partial^{\boldsymbol{\alpha}} \boldsymbol{h}_i(\boldsymbol{X}_0). \tag{19}$$

The derivative tensors $\mathbf{D}_k^{(T)} = \mathbf{D}(\boldsymbol{h}^{(T)})_k$ and $\mathbf{D}_k^{\mathrm{out}} = \mathbf{D}(\boldsymbol{h}^{\mathrm{out}})_k$ are computed using a message-passing-like procedure detailed in Appendix 1, which enables both efficient computation and allows us to backpropagate through it, supporting end-to-end training of $k$-HOD-GNN.

---

[6]For a definition of multi-dimensional analytic functions see Appendix E.1.

**Step 3.** In the third stage of our method, we extract new node features informed by the derivative tensors $\mathbf{D}_k^{(T)}, \mathbf{D}_k^{\text{out}}$, using the encoder networks $\text{U}^{\text{node}} : \mathbb{R}^{n \times n^k \times d' \times m^{d \times k}} \to \mathbb{R}^{n \times d''}$ and $\text{U}^{\text{out}} : \mathbb{R}^{n^k \times d' \times m^{d \times k}} \to \mathbb{R}^{n \times d''}$. As $\mathbf{D}_k^{(T)}$ is a tensor indexed by $(k+1)$ nodes and $\mathbf{D}_k^{\text{out}}$ is a tensor indexed by $k$ nodes, natural choices for $\text{U}^{\text{node}}$ and $\text{U}^{\text{out}}$ are $(k+1)$-IGN and a $k$-IGN (Maron et al., 2018) designed specifically to process such data.

Furthermore, Proposition 4.4 shows that for sparse graphs or relatively shallow base MPNNs, the derivative tensor $\mathbf{D}_k^{(T)}$ itself becomes sparse, with space complexity $O\left(n \cdot \min\{n^k, d^{k \cdot T}\}\right)$, where $d$ is the maximum degree of the input graph. This motivates the choice of a node encoder $\text{U}^{\text{node}}$ that preserves this sparsity structure and exploits it for improved computational efficiency.

To enable this, we can define the encoder $\text{U}^{\text{node}}$ as a subclass of the general $(k+1)$-IGN architecture, implemented as a DeepSet (Zaheer et al., 2017) operating independently on the derivative entries associated with each node feature $\boldsymbol{h}_v^{(T)}$. Specifically, we define:

$$\text{U}^{\text{ds-node}}(\mathbf{D}_k^{(T)})_v = \text{DeepSet}(\{(\mathbf{D}_k^{(T)}[v, \boldsymbol{u}, i, \boldsymbol{\alpha}], t(v, \boldsymbol{u}, \boldsymbol{\alpha})) \mid \boldsymbol{u} \in V^k(\mathcal{G}), i \in [d'], \boldsymbol{\alpha} \in [m]^{k \times d}\}). \tag{20}$$

Here we assume nodes are given in index form, that is $v \in \{1, \dots, n\}., \boldsymbol{u} \in \{1, \dots, n\}^k$, and $t(v, \boldsymbol{u}, \boldsymbol{\alpha})$ is a function that encodes the derivative pattern associated with the index tuple $(v, \boldsymbol{u}, \boldsymbol{\alpha})$, that is

$$t(v_1, \boldsymbol{u}_1, \boldsymbol{\alpha}_1) = t(v_2, \boldsymbol{u}_2, \boldsymbol{\alpha}_2) \quad \Leftrightarrow \quad \boldsymbol{\alpha}_1 = \boldsymbol{\alpha}_2 \text{ and } \exists \sigma \in S_n \text{ such that } v_1 = \sigma(v_2), \ \boldsymbol{u}_1 = \sigma(\boldsymbol{u}_2), \tag{21}$$

where $S_n$ denotes the symmetric group on $n$ elements. In other words, $t$ maps each derivative index tuple to a *canonical identifier* that is invariant under permutations of the node indices but sensitive to the derivative multi-index.

This design improves both space and time complexity, as the sets over which the DeepSet operates are typically sparse. The proof of Theorem 4.1 provided in appendix E.2 shows that this encoder architecture is enough to be as expressive as $k$-OSAN.

Finally, we proceed the same way as 1-HOD-GNN constructing the derivative-informed node features $\boldsymbol{h}^{\text{der}}$ by combining information from the base MPNN, the pooled intermediate derivatives, and the output derivatives:

$$\boldsymbol{h}_v^{\text{der}} = \boldsymbol{h}_v^{(T)} \oplus \text{U}^{\text{out}}(\mathbf{D}_k^{\text{out}}) \oplus \text{U}^{\text{node}}(\mathbf{D}_k^{(\text{T})}). \tag{22}$$

**Step 4.** This step is identical to that of 1-HOD-GNN described in Section 3.2.

## C.2 EDGE-HOD-GNN

In this section, we extend the 1-HOD-GNN formulation to incorporate edge-feature derivatives rather than node-feature derivatives. We refer to this variant as edge-HOD-GNN. The construction of this architecture closely parallels that of Section 3.2 and is detailed below.

**Steps 1 & 2.** Similarly to steps 1 & 2 in Section 3.2 , in the first two stages, we compute the final node representations $\boldsymbol{h}^{(T)}$ and the output vector $\boldsymbol{h}^{\text{out}}$ using the base MPNN $\mathcal{M}$. The key difference is that we now compute the edge derivative tensors, defined below:

**Definition C.2.** Given a graph $\mathcal{G} = (\boldsymbol{A}, \boldsymbol{X}, \boldsymbol{E})$ with $n$ nodes, $l$ edges, an edge feature matrix $\boldsymbol{E}$, an MPNN $\mathcal{M}$ and an intermediate node feature representation matrix $\boldsymbol{h} \in \mathbb{R}^{n \times d'}$, the edge-derivative tensor $\mathbf{D}(\boldsymbol{h}) \in \mathbb{R}^{n \times l \times d' \times d \times m}$ is defined by:

$$\mathbf{D}_{\text{edge}}(\boldsymbol{h})[v, e, i, j, \alpha] = \frac{\partial^{\alpha} \boldsymbol{h}_{v,i}}{\partial \boldsymbol{E}_{e,j}^{\alpha}}, \tag{23}$$

where $v \in V(\mathcal{G}), e \in E(\mathcal{G}), i \in [d'], j \in [d]$ specify the feature dimensions of the feature vectors $\boldsymbol{h}_v, \boldsymbol{E}_e$ respectively, and $\alpha \in [m]$ where $m \in \mathbb{N}$ is a hyperparameter specifying the maximum order of derivatives to be considered. Similarly, given a graph-level prediction

vector $\boldsymbol{h}^{\text{out}} \in \mathbb{R}^{d'}$ the derivative tensor $\mathbf{D}(\boldsymbol{h}^{\text{out}}) \in \mathbb{R}^{l \times d' \times d \times m}$ is defined by:

$$\mathbf{D}_{\text{edge}}(\boldsymbol{h}^{\text{out}})[e, i, j, \alpha] = \frac{\partial^{\alpha} \boldsymbol{h}_i^{\text{out}}}{\partial \boldsymbol{E}_{e,j}^{\alpha}}. \tag{24}$$

**Step 3.** Similarly to step 3 in Section 3.2, in the third stage of our method, we extract new edge features from the edge derivative tensors $\mathbf{D}_{\text{edge}}^{\text{out}}$ and $\mathbf{D}_{\text{edge}}^{(T)}$ using encoder networks $\text{U}_{\text{edge}}^{\text{node}}$ and $\text{U}_{\text{edge}}^{\text{out}}$. First, as $\mathbf{D}_{\text{edge}}^{\text{out}} \in \mathbb{R}^{l \times d' \times d \times m}$ is a tensor indexed by a single edge, it can be directly interpreted as an edge feature matrix by flattening the remaining dimensions. We thus define the encoder network $\text{U}^{\text{out}}$ to be a DeepSets (Zaheer et al., 2017) update:

$$\text{U}_{\text{edge}}^{\text{out}}(\mathbf{D}_{\text{edge}}^{\text{out}})_e = \text{MLP}(\mathbf{D}_{\text{edge}}^{\text{out}}[e, \dots]). \tag{25}$$

Secondly, since $\mathbf{D}_{\text{edge}}^{(T)}$ is indexed by node-edge pairs, it shares the structure of the incidence matrix of $\mathcal{G}$ denoted by $\boldsymbol{B}$. We can thus define the encoder network $\text{U}_{\text{edge}}^{\text{node}} : \mathbb{R}^{n \times l \times d' \times d \times m} \to \mathbb{R}^{l \times \bar{d}}$ to be any GNN architecture which maps incidence matrices to edge feature matrices. An example of such architecture is Albooyeh et al. (2019).

**Step 4.** In the final stage, we replace the original edge features of $\mathcal{G}$ with the derivative-informed features $\boldsymbol{E}^{\text{der}}$, and apply a downstream GNN $\mathcal{T}$ to produce a graph-level prediction.

## D  DERIVATIVE COMPUTATION

We now extend the derivative computation algorithm presented in Section 3.2.1, to account for $k$-mixed derivatives as well as more general MPNN architectures. Similarly to Section 3.2.1, we first split the node update procedure of an MPNN into two parts: an aggregation step:

$$\tilde{\boldsymbol{h}}_v^{(t-1)} = \boldsymbol{h}_v^{(t-1)} \oplus \text{AGG}^{(t)}\left(\left\{\boldsymbol{h}_u^{(t-1)} : u \in \mathcal{N}(v)\right\}\right), \tag{26}$$

and a DeepSet step:

$$\boldsymbol{h}_v^{(t)} = \text{MLP}(\tilde{\boldsymbol{h}}_v^{(t-1)}). \tag{27}$$

Our algorithm begins by computing the initial derivative tensor $\mathbf{D}^{(0)}$ (we abuse notation and ommit the subscript $k$), and then recursively constructs each $\tilde{\mathbf{D}}^{(t-1)} = \mathbf{D}(\tilde{\boldsymbol{h}}^{(t-1)})$ from $\mathbf{D}^{(t-1)}$ and then $\mathbf{D}^{(t)}$ from $\tilde{\mathbf{D}}^{(t-1)}$. Finally, the output derivative tensor $\mathbf{D}^{\text{out}}$ is obtained from $\mathbf{D}^{(T)}$. See Algorithm 1 for the full procedure.

To analyze the time and memory complexity of each step, we define the derivative sparsity of a node $v$ of an input graph $\mathcal{G}$ at layer $t$ of the base MPNN, as the number of non-zero derivatives corresponding to $\boldsymbol{h}_v^{(t)}$. That is

$$s_{v,t} = \left|\left\{(\boldsymbol{u}, i, \boldsymbol{\alpha}) \,\Big|\, \mathbf{D}^{(t)}[v, \boldsymbol{u}, i, \boldsymbol{\alpha}] \neq 0\right\}\right|. \tag{28}$$

$$s_t = \min_{v \in V(\mathcal{G})} s_{v,t}. \tag{29}$$

When $s_t$ is small, the tensor $\mathbf{D}^{(t)}$ can be stored efficiently in memory using sparse representations, requiring only $O(n \cdot s_t)$ space. The quantities $s_{v,t}$ and $s_t$ are leveraged in Appendix E.2 to derive concrete asymptotic bounds for the complexity of Algorithm 1.

**Computing $\mathbf{D}^{(0)}$.** Since $\boldsymbol{h}^{(0)} = \boldsymbol{X}$, the derivatives are straightforward to compute:

$$\mathbf{D}^{(0)}[v, \boldsymbol{u}, i, \boldsymbol{\alpha}] = \begin{cases} 1 & \text{if } \exists s \text{ s.t. } v = \boldsymbol{u}_s, \boldsymbol{\alpha}_{s,i} = 1, \sum_{s' \neq s, j \in [d]} \boldsymbol{\alpha}_{s',j} = 0 \\ 0 & \text{otherwise.} \end{cases} \tag{30}$$

Although $\mathbf{D}^{(0)}$ is high-dimensional, it is extremely sparse with $s_0 = O(1)$.

**Computing $\tilde{\mathbf{D}}^{(t-1)}$ from $\mathbf{D}^{(t-1)}$.** Abusing notation, and assuming that for every node $v$ in the input graph, both $\boldsymbol{h}_v^{(t-1)}$ and $\boldsymbol{h}_v^{(t-1),\text{agg}} = \text{AGG}^{(t)}\left(\{\boldsymbol{h}_u^{(t-1)} : u \in \mathcal{N}(v)\}\right)$ lie in $\mathbb{R}^{d'}$,

we observe that since $\tilde{\boldsymbol{h}}_{v,0:d'-1}^{(t-1)} = \boldsymbol{h}_v^{(t-1)}$, the derivatives of the first $d'$ coordinates of $\tilde{\boldsymbol{h}}$ are precisely $\mathsf{D}^{(t-1)}$. In contrast, the derivatives with respect to the last $d'$ coordinates depend heavily on the choice of aggregation function AGG.

However, when the aggregation function is linear, i.e., of the form

$$\mathrm{AGG}^{(t)}\left(\{\boldsymbol{h}_u^{(t-1)} : u \in \mathcal{N}(v)\}\right) = \sum_{v' \in \mathcal{N}(v)} b_{v',v}\boldsymbol{h}_{v'}^{(t-1)}, \tag{31}$$

for some coefficients $b_{v',v}$ that depend only on the adjacency matrix $\boldsymbol{A}$ of the input graph (this is the case in most widely used MPNNs), the computation of derivatives simplifies. Since differentiation commutes with linear operations, the computation in Equation 31 carries over, yielding

$$\mathsf{D}(\boldsymbol{h}^{(t-1),\mathrm{agg}})[v,\dots] = \sum_{v' \in \mathcal{N}(v)} b_{v',v}\mathsf{D}^{(t-1)}[v',\dots]. \tag{32}$$

Aggregating the derivatives of neighboring nodes mirrors the structure of message passing, which endows it with several beneficial properties. First, using Equation 32, the tensor $\tilde{\mathsf{D}}^{(t-1)}[v,\dots]$ can be computed in time $O(d \cdot s_{t-1})$ where $d$ is the maximal degree of the input graph[7]: for each node, we aggregate $d$ neighbor derivative vectors, each containing at most $s_{t-1}$ non-zero entries. This means that the total computation time of this step is $O(d \cdot n \cdot s_{t-1})$.

The argument above also implies that the total number of non-zero elements in $\tilde{\mathsf{D}}^{(t-1)}[v,\dots]$ is bounded by $O\left(\min\{d \cdot s_{t-1}, n^k\}\right)$. Thus, the above update leverages the sparsity of the graph to achieve efficiency in both space and time complexity.

**Computing $\mathsf{D}^{(t)}$ from $\tilde{\mathsf{D}}^{(t-1)}$.**

We begin by assuming that the MLP in Equation 27 has depth 1, i.e.,

$$\mathsf{MLP}^{(t)}(\boldsymbol{x}) = \sigma(\boldsymbol{W}^{(t)} \cdot \boldsymbol{x} + \boldsymbol{b}^{(t)}). \tag{33}$$

In the general case where the MLP has depth $l$, the procedure described below is applied recursively $l$ times.

We define the intermediate linear activation as

$$\boldsymbol{h}_v^{(t-1),\mathrm{Lin}} = \boldsymbol{W}^{(t)} \cdot \tilde{\boldsymbol{h}}_v^{(t)} + \boldsymbol{b}^{(t)}, \tag{34}$$

and describe how to compute $\mathsf{D}(\boldsymbol{h}^{(t-1),\mathrm{Lin}})$ from $\tilde{\mathsf{D}}^{(t-1)}$, followed by the computation of $\mathsf{D}^{(t)}$ from $\mathsf{D}(\boldsymbol{h}^{(t-1),\mathrm{Lin}})$.

First, since the update in Equation 34 is affine, we can drop the bias term and get:

$$\mathsf{D}(\boldsymbol{h}^{(t-1),\mathrm{Lin}})[v,\boldsymbol{u},i,\alpha] = \sum_{i'} \boldsymbol{W}_{i,i'}^{(t)} \cdot \tilde{\mathsf{D}}^{(t-1)}[v,\boldsymbol{u},i',\alpha]. \tag{35}$$

Second, notice that

$$\boldsymbol{h}_v^{(t)} = \sigma(\boldsymbol{h}_v^{(t-1),\mathrm{Lin}}). \tag{36}$$

We can use the Faà di Bruno's formula (see e.g. (Hardy, 2006)) which generalizes the chain rule to higer-order derivatives, for our last step. Faà di Bruno's formula states that for a pair of functions $g : \mathbb{R}^n \to \mathbb{R}$, $f : \mathbb{R} \to \mathbb{R}$, $y = g(x_1,\dots,x_n)$ the following holds, regardless of whether the variables $x_1,\dots,x_n$ are all distinct, identical, or grouped into distinguishable categories of indistinguishable variables:

$$\frac{\partial^n}{\partial x_1 \cdots \partial x_n}f(y) = \sum_{\pi \in \Pi} f^{(|\pi|)}(y) \cdot \prod_{B \in \pi} \frac{\partial^{|B|}y}{\prod_{j \in B}\partial x_j} \tag{37}$$

where:

---

[7]We slightly abuse notation by using $d$ to denote both the input feature dimension and other quantities; the meaning should be clear from context.

- $\Pi$ denotes the collection of all partitions of the index set $\{1, \ldots, n\}$,
- The notation $B \in \pi$ indicates that $B$ is one of the subsets (or "blocks") in the partition $\pi$ ,
- For any set $A$ , the notation $|A|$ represents its cardinality. In particular, $|\pi|$ is the number of blocks in the partition, and $|B|$ is the number of elements in the block $B$.

Equations 36 and 37 Imply that we are able to compute $\mathbf{D}^{(t)}[v, \boldsymbol{u}, \ldots]$ based only on $\mathbf{D}(\boldsymbol{h}^{(t-1),\mathrm{Lin}})[v, \boldsymbol{u}, \ldots]$ and the derivatives of $\sigma$ at the point $\boldsymbol{h}_v^{(t-1),\mathrm{Lin}}$. Combining this update with Equation 35 results in a way to compute $\mathbf{D}^{(t)}$ from $\tilde{\mathbf{D}}^{(t-1)}$.

Importantly, the update above computes each entry of $\mathbf{D}^{(t)}[v, \boldsymbol{u}, \ldots]$ using only the corresponding entries of $\tilde{\mathbf{D}}^{(t-1)}[v, \boldsymbol{u}, \ldots]$, i.e., those associated with the same node tuple $(v, \boldsymbol{u})$. Moreover, from Equations 35 and 37, it follows that if $\tilde{\mathbf{D}}^{(t-1)}[v, \boldsymbol{u}, \ldots] = \mathbf{0}$, then $\mathbf{D}^{(t)}[v, \boldsymbol{u}, \ldots] = \mathbf{0}$ as well. This implies that like $\tilde{\mathbf{D}}^{(t-1)}$ the number of non-zero entries in each $\mathbf{D}^{(t)}[v, \ldots]$ is also bounded by $O\left(\min\{d \cdot s_{t-1}, n^k\}\right)$. Consequently, the update—performed only over the non-zero entries of $\tilde{\mathbf{D}}^{(t-1)}$—has a runtime complexity of $O\left(n \cdot \min\{d \cdot s_{t-1}, n^k\}\right)$.

**Computing $\mathbf{D}^{\mathbf{out}}$ from $\mathbf{D}^{(T)}$.**

Recall that

$$\boldsymbol{h}^{\mathrm{out}} = \mathrm{AGG}_{\mathrm{fin}}\left(\left\{\boldsymbol{h}_v^{(T)} \mid v \in V(\mathcal{G})\right\}\right), \tag{38}$$

where $\mathrm{AGG}_{\mathrm{fin}}$ denotes the final aggregation over node embeddings. This operation can be treated analogously to the node update step: For most common MPNNs, it decomposes into a linear aggregation followed by an MLP. Consequently, the derivative tensor $\mathbf{D}^{\mathrm{out}}$ can be computed from $\mathbf{D}^{(T)}$ using the same primitives described above.

---

**Algorithm 1** Efficient Computation of Derivative Tensors

---

**Require:** Graph $\mathcal{G} = (\boldsymbol{A}, \boldsymbol{X})$, base GIN $\mathcal{M}$ with $T$ layers
  1: $\boldsymbol{h}^{(0)} \leftarrow \boldsymbol{X}$                                                                          ▷ node feature init.
  2: $\mathbf{D}^{(0)} \leftarrow \mathbf{D}(\boldsymbol{X})$                                                           ▷ deriv. init through Eq 30.
  3: **for** $t = 1$ to $T$ **do**
  4:     $\tilde{\boldsymbol{h}}_v^{(t-1)} \leftarrow \boldsymbol{h}_v^{(t-1)} \oplus \left(\sum_{v' \in \mathcal{N}(v)} b_{v',v} \boldsymbol{h}_{v'}^{(t-1)}\right)$                    ▷ linear node agg.
  5:     $\tilde{\mathbf{D}}^{(t-1)}[v, \ldots] \leftarrow \mathbf{D}^{(t-1)}[v, \ldots] \oplus \left(\sum_{v' \in \mathcal{N}(v)} b_{v',v} \mathbf{D}^{(t-1)}[v', \ldots]\right)$        ▷ deriv. agg.
  6:     $\boldsymbol{h}_v^{(t)} = \mathsf{MLP}^{(t-1)}(\tilde{\boldsymbol{h}}^{(t-1)})_v$                                            ▷ DeepSet update
  7:     $\mathbf{D}^{(t)}[v, \ldots] = \text{get-der}(\mathsf{MLP}^{(t-1)}, \tilde{\mathbf{D}}^{(t-1)}[v, \ldots], \tilde{\boldsymbol{h}}_v^{(t-1)})$        ▷ deriv. DeepSet update.
  8: **end for**
        $\mathbf{D}^{\mathrm{out}} = \text{get-out-der}(\mathbf{D}^{(T)})$                                            ▷ extract output deriv.
  9: **return** $\mathbf{D}^{(T)}, \mathbf{D}^{\mathrm{out}}$

---

## E  EXTENDED THEORETICAL ANALYSIS

### E.1  DEFINITIONS

Before delving into the proofs, we begin by formally defining several key concepts used throughout the analysis:

**Definition E.1** (Analytic Function). A function $f : \mathbb{R}^n \to \mathbb{R}$ is said to be *analytic* at $\boldsymbol{x}_0 \in \mathbb{R}^n$ if for some $R \in \mathbb{R}^n$ it holds that for all $|X| < \mathrm{R}$:

$$f(\boldsymbol{x}) = \sum_{|\boldsymbol{\alpha}|=0}^{\infty} \frac{1}{\boldsymbol{\alpha}!} \partial^{\boldsymbol{\alpha}} f(\boldsymbol{x}_0)(\boldsymbol{x} - \boldsymbol{x}_0)^{\boldsymbol{\alpha}} \tag{39}$$

where $\boldsymbol{\alpha} \in \mathbb{N}^n$ and we use the following notation:

- $|\boldsymbol{\alpha}| = \alpha_1 + \alpha_2 + \cdots + \alpha_n$,

- $\boldsymbol{\alpha}! = \alpha_1! \cdot \alpha_2! \cdots \alpha_n!$,

- $(\boldsymbol{x})^{\boldsymbol{\alpha}} = (x_1)^{\alpha_1} \cdot (x_2)^{\alpha_2} \cdots (x_n)^{\alpha_n}$

- $\partial^{\boldsymbol{\alpha}} f(\mathbf{a}) = \frac{\partial^{|\boldsymbol{\alpha}|} f}{\partial x_1^{\alpha_1} \partial x_2^{\alpha_2} \cdots \partial x_n^{\alpha_n}}\Big|_{\mathbf{x}=\mathbf{a}}$.

The largest such $R$ is called the radius of convergence. A function $f : \mathbb{R}^n \to \mathbb{R}^m$ is analytic if all functions $f_1, \ldots, f_m$ are analytic.

**Definition E.2** ($k$-OSAN)**.** A $k$-OSAN model $\Psi$ consists of a base MPNN $\mathcal{M}$ that produces updated node features (as opposed to directly outputting a graph-level prediction), followed by a downstream MPNN $\mathcal{T}$ that aggregates these features to produce a final graph-level output. Given an input graph $\mathcal{G} = (\boldsymbol{A}, \boldsymbol{X})$, the output $\Psi(\mathcal{G})$ is computed in four stages:

**Step 1**: Construct a bag of subgraphs $\mathcal{B}_{\mathcal{G}} = \{\mathcal{S}_{\boldsymbol{u}} \mid \boldsymbol{u} \in V^k(\mathcal{G})\}$ each of the form $\mathcal{S}_{\boldsymbol{u}} = (\boldsymbol{A}, \boldsymbol{X} \oplus \boldsymbol{e}^{\boldsymbol{u}})$. Here $\boldsymbol{e}^{\boldsymbol{u}} \in \mathbb{R}^{n \times k}$ is a "node marking"[8] feature matrix assigning a unique identifier to each node $u_1, \ldots u_k$. That is:

$$\boldsymbol{e}^{\boldsymbol{u}}_{v,j} = \begin{cases} 1 & v = u_j \\ 0 & \text{else.} \end{cases} \tag{40}$$

**Step 2**:

Compute the $(k+1)$-node indexed tensor:

$$\mathbf{H}[v, \boldsymbol{u}] = \mathcal{M}(\mathcal{S}_{\boldsymbol{u}})_v \tag{41}$$

**Step 3**: use a set aggregation function to produce new node features:

$$\boldsymbol{h}^{\text{sub}}_v = \text{AGG}(\{\mathbf{H}[v, \boldsymbol{u}] \mid \boldsymbol{u} \in V^k(\mathcal{G})\}). \tag{42}$$

**Step 4**: Compute the final output through:

$$\Psi(\mathcal{G}) = \mathcal{T}(\boldsymbol{A}, \boldsymbol{h}^{\text{sub}}). \tag{43}$$

For $k = 1$, $k$-OSAN are also reffered to as DS-GNNs.

**Definition E.3** (Random Walk Structural Encoding)**.** For a graph $\mathcal{G} = (\boldsymbol{A}, \boldsymbol{X})$, the Random Walk Structural Encoding (RWSE) with $L$ number of steps is defined as

$$\boldsymbol{h}^{\text{rwse}} = \bigoplus_{l=1}^{L} \text{diag}(\tilde{\boldsymbol{A}}^l), \tag{44}$$

where $\tilde{\boldsymbol{A}} = \boldsymbol{A} \cdot \text{Diag}(\deg(u_1)^{-1}, \ldots, \deg(u_n)^{-1})$ is the row-normalized adjacency matrix, $\text{diag}(\cdot)$ denotes the vector of diagonal entries of a matrix, and $\text{Diag}(\boldsymbol{v})$ denotes the diagonal matrix with vector $\boldsymbol{v}$ on its diagonal.

### E.2 Proofs

**Theorem 4.1.** We begin by formally stating and proving Theorem 4.1, splitting it into two Theorem-corollary pairs.

**Theorem E.4** ($k$-HOD-GNN is as expressive as $k$-OSAN)**.** *Let* $\{\mathcal{G}^i = (\boldsymbol{A}^i, \boldsymbol{X}^i) \mid i \in [l]\}$ *be a finite set of graphs, and let* $\Psi$ *be a $k$-OSAN model. for any* $\epsilon > 0$, *there exists a $k$-HOD-GNN model* $\Phi$ *such that for each* $i \in [l]$

$$|\Psi(\mathcal{G}_i) - \Phi(\mathcal{G}_i)| < \epsilon. \tag{45}$$

---

[8]Although alternative methods for initializing node features in subgraphs have been proposed, they offer the same expressive power. We therefore focus on the simple approach used here.

**Corollary E.5.** *There exist non-isomorphic graphs that are indistinguishable by the folklore k-WL test but are distinguishable by k-HOD-GNN. Additionally, k-HOD-GNN is able to compute the homomorphism count of k-apex forest graphs.*

*Proof.* We begin the proof by making a few simplifying assumptions on $\Psi$, which we can do without loss of generality. We begin by assuming that all activation functions used in $\mathcal{M}$, the base MPNN of $\Psi$, are analytic with infinite radius of convergence (e.g., $\exp(x)$, $\sin(x)$). This assumption can be made without loss of generality: Since MLPs with non-polynomial analytic activations are universal approximators, each MLP in $\mathcal{M}$ can be replaced with one using an analytic activation function that approximates the original to arbitrary precision. Furthermore, since the composition of analytic functions with infinite convergence radius remains analytic with infinite convergence radius, it follows that $\mathcal{M}$—as a composition of affine transformations and activation functions—is itself analytic.

Secondly, we assume that the final node representations produced by $\mathcal{M}$ are one-dimensional. This assumption can be made without loss of generality: we can append a final MLP to $\mathcal{M}$ that compresses each node's feature vector to a scalar, and prepend an MLP to the downstream network $\mathcal{T}$ that reconstructs the original feature dimension. This effectively amounts to inserting an autoencoder, where the encoder is a final pointwise update in $\mathcal{M}$ and the decoder is an initial pointwise update in $\mathcal{T}$. Since this architecture can approximate the original architecture to arbitrary precision, we may assume without loss of generality that $\mathcal{M}$ produces 1-dimensional node embeddings.

Additionally, we can assume the input node feature matrices $\boldsymbol{X}^i, i \in [l]$ are also all 1-dimensional. This follows from the same argument as above.

Finally, we consider $k$-HOD-GNN models that only use the node derivative tensor $\mathbf{D}^{(T)}$, which we use to extract node features through an IGN encoder U, and disregard $\mathbf{D}^{\text{out}}$.

We prove the theorem in three steps

1. We show that an intermediate representation of $\text{U}(\mathbf{D}^{(T)})$ can encode the tensor $\mathbf{H}_{v,\boldsymbol{u}}$ that is produced at stage (2) of the forward pass of $\Psi$ (see Definition E.2). [9]

2. we show that $\text{U}(\mathbf{D}^{(T)})$ can approximate the node feature matrix $\boldsymbol{h}^{\text{sub}}$ produced at stage (3) of the forward pass of $\Psi$.

3. We show $\Phi$ can approximate $\Psi$ to finite precision.

**Step 1** For each k-tuple of nodes $\boldsymbol{u} \in V^k(\mathcal{G}^i)$ and matrix $\boldsymbol{Y} \in \mathbb{R}^{k \times k}$, we define the node feature matrix $\text{broad}_{\boldsymbol{u}}(\boldsymbol{Y}) \in \mathbb{R}^{n \times k}$ (here we abuse notation and not include the graph index $i$) by

$$\text{broad}_{\boldsymbol{u}}(\boldsymbol{Y})_{v,i} = \begin{cases} y_{j,i} & v = \boldsymbol{u}_j \\ 0 & \text{else.} \end{cases} \tag{46}$$

Additionally, for each node $v \in \mathcal{G}^i$, we define $f^i_{v,\boldsymbol{u}} : \mathbb{R}^{k \times k} :\to \mathbb{R}$

$$f^i_{v,\boldsymbol{u}}(\boldsymbol{Y}) = \mathcal{M}(\boldsymbol{A}, \boldsymbol{X} \oplus \text{broad}_{\boldsymbol{u}}(\boldsymbol{Y})). \tag{47}$$

Finally, define $\boldsymbol{Y}^{\boldsymbol{u}} \in \mathbb{R}^{k \times k}$ by

$$\boldsymbol{Y}^{\boldsymbol{u}}_{i,j} = \boldsymbol{e}^{\boldsymbol{u}}_{u_i,j}. \tag{48}$$

where $\boldsymbol{e}^{\boldsymbol{u}}$ is the node marking node feature matrix introduced in Definition E.2 (That is, $\text{broad}_{\boldsymbol{u}}(\boldsymbol{Y}^{\boldsymbol{u}}) = \boldsymbol{e}^{\boldsymbol{u}}$ ).

First, it is easy to see that

$$f^i_{v,\boldsymbol{u}}(\boldsymbol{Y}^{\boldsymbol{u}}) = \mathcal{M}(\mathcal{S}^i_{\boldsymbol{u}})_v = \mathbf{H}^i[v, \boldsymbol{u}], \tag{49}$$

where $\mathcal{S}^i_{\boldsymbol{u}}$ and $\mathbf{H}^i[v, \boldsymbol{u}]$ are introduced in Definition E.2.

---

[9]In cases where $\boldsymbol{u}$ has repeated entries $\boldsymbol{u}_{j_1} = \boldsymbol{u}_{j_2}$, we only consider values of $\boldsymbol{Y}$ for which $\boldsymbol{Y}_{j_1,:} = \boldsymbol{Y}_{j_2,:}$.

Second, as $\mathcal{M}$ is analytic with infinite convergence radius, the functions $f_{v,\boldsymbol{u}}^i$ are all analytic with infinite convergence radi, and so

$$f_{v,\boldsymbol{u}}^i(\boldsymbol{Y^u}) = \sum_{|\boldsymbol{\alpha}|=0}^{\infty} \frac{1}{\boldsymbol{\alpha}!} \partial^{\boldsymbol{\alpha}} f^i(\boldsymbol{0})(\boldsymbol{Y^u})^{\boldsymbol{\alpha}}. \tag{50}$$

Here, $\boldsymbol{\alpha} \in \{0,\ldots,m-1\}^{k \times k}$ and $(\boldsymbol{Y^u})^{\boldsymbol{\alpha}} = \prod(\boldsymbol{Y_{j_1,j_2}^u})^{\boldsymbol{\alpha}_{j_1,j_2}}$. Since we are concerned with a finite number of graphs, for any $\epsilon > 0$ we can choose an integer $I$ such that for all graphs $\mathcal{G}^i$ and all $v \in V(\mathcal{G}^i)$, $\boldsymbol{u} \in V^k(\mathcal{G}^i)$ it holds that:

$$|f_{v,\boldsymbol{u}}^i(\boldsymbol{Y^u}) - \sum_{|\boldsymbol{\alpha}|=0}^{I} \frac{1}{\boldsymbol{\alpha}!} \partial^{\boldsymbol{\alpha}} f_{v,\boldsymbol{u}}^i(\boldsymbol{0})(\boldsymbol{Y^u})^{\boldsymbol{\alpha}}| < \epsilon. \tag{51}$$

Defining

$$\tilde{\mathbf{H}}[^iv,\boldsymbol{u}] = \sum_{|\boldsymbol{\alpha}|=0}^{I} \frac{1}{\boldsymbol{\alpha}!} \partial^{\boldsymbol{\alpha}} f_{v,\boldsymbol{u}}^i(\boldsymbol{0})(\boldsymbol{Y^u})^{\boldsymbol{\alpha}}, \tag{52}$$

we get that

$$\tilde{\mathbf{H}}^i \approx \mathbf{H}^i \tag{53}$$

Additionally, from the definition of $f_{v,\boldsymbol{u}}^i$ it holds that the derivatives of $f_{v,\boldsymbol{u}}^i$ at zero correspond to the entries of the $k$-indexed derivative tensor of $\mathcal{G}^i$, denoted by $\mathbf{D}^{(T),i}$[10], that is

$$\partial^{\boldsymbol{\alpha}} f_{v,\boldsymbol{u}}^i(\boldsymbol{0}) = \mathbf{D}^{(T),i}[v,\boldsymbol{u},\boldsymbol{\alpha}], \tag{54}$$

where we only take derivatives with respect to the last $k$ feature dimensions, which correspond to the "marking vectors"

Moreover, let $t$ be a function such that $t(v,\boldsymbol{u},\boldsymbol{\alpha})$ encodes the derivative pattern associated with the index tuple $(v,\boldsymbol{u},\boldsymbol{\alpha})$. That is, $t$ satisfies:

$$t(v_1,\boldsymbol{u}_1,\boldsymbol{\alpha}_1) = t(v_2,\boldsymbol{u}_2,\boldsymbol{\alpha}_2) \quad \Leftrightarrow \quad \boldsymbol{\alpha}_1 = \boldsymbol{\alpha}_2; \text{ and } ; \exists \sigma \in S_n \text{ such that } v_1 = \sigma(v_2),; \boldsymbol{u}_1 = \sigma(\boldsymbol{u}_2), \tag{55}$$

where $S_n$ is the symmetric group on $n$ elements. The value of $\frac{1}{\boldsymbol{\alpha}!}(\boldsymbol{Y^u})^{\boldsymbol{\alpha}}$ is determined entirely by $t(v,\boldsymbol{u},\boldsymbol{\alpha})$, and thus can be recovered from it.

This implies that the DeepSet encoder $\mathrm{U}^{\mathrm{ds\text{-}node}}$ defined in Equation 20 in Appendix C.1, can have an intermediate layer $L$ such that

$$L(\mathbf{D}^{(T),i}) = \tilde{\mathbf{H}}^i. \tag{56}$$

Here $L$ simply multiples each entry $\mathbf{D}^{(T),i}[v,\boldsymbol{u},\boldsymbol{\alpha}]$ by $\frac{1}{\boldsymbol{\alpha}!}(\boldsymbol{Y^u})^{\boldsymbol{\alpha}}$, followed by summing over the $\boldsymbol{\alpha}$ indices. Thus, by Equations 53 and 56, choosing $\mathcal{M}$ as the base MPNN of a $k$-HOD-GNN model allows us to approximate each $\mathbf{H}^i$ to arbitrary precision using its derivatives. Note that since the DeepSet encoder is a restricted instance of a $(k+1)$-IGN encoder, it can achieve the same effect.

**Step 2** Equation 42 shows that the node features $\boldsymbol{h}_v^{\mathrm{sub},i}$ are constructed by applying a set-wise aggregation function over the set $\{\mathbf{H}^i[v,\boldsymbol{u}] \mid \boldsymbol{u} \in V^k(\mathcal{G}^i)\}$. Any continuous set-wise aggregation function can be approximated to arbitrary precision by a DeepSet architecture (Zaheer et al., 2017) (see Segol & Lipman (2019) for proof). Moreover, any DeepSet model applied in parallel over the first index of a $k$-indexed node tensor can be exactly implemented by a $k$-IGN, since each layer in such a model consists of a linear equivariant transformation followed by a pointwise nonlinearity. Thus, we can construct our encoder U to first approximate the mapping $\mathbf{D}^i \to \tilde{\mathbf{H}}^i$ as an intermediate representation, and then

---

[10]here we abuse notation and omit the subscrit $k$ in $\mathbf{D}_k$.

approximate the subsequent mapping $\tilde{\mathsf{H}}^i \to \boldsymbol{h}^{\text{sub},i}$. Since both approximations can be made to arbitrary precision, this completes the proof of the claim in Step 2.

**Step 3** The final forward step—applying a downstream GNN to the updated node features to produce a graph-level representation—is identical in both OSAN and HOD-GNN. Therefore, by choosing the downstream GNN in the $k$-HOD-GNN model to match that of $\Psi$, the proof is complete.

$\square$

Using Theorem E.4 we now prove corollary E.5

*Proof of corollary E.5.* It was shown by Qian et al. (2022) that $k$-OSAN models can distinguish between graphs that are indistinguishable by the $k$-WL test. As Theorem E.4 establishes that our method can approximate any $k$-OSAN model to arbitrary precision, it follows that $k$-HOD-GNN can do the same. Similarly, Zhang et al. (2024b) showed that $k$-OSAN models can compute homomorphism counts of $k$-apex forests—graphs in which the removal of at most $k$ nodes yields a forest. Therefore, by Theorem E.4, Corollary E.5 follows. $\square$

**Theorem E.6** (($k+2$)-IGNs are as expressive as $k$-HOD-GNN)**.** *Let $\Phi$ be a $k$-HOD-GNN model and let $\mathcal{G}, \mathcal{G}'$ be a pair of graphs such that*

$$\Phi(\mathcal{G}) \neq \Phi(\mathcal{G}'). \tag{57}$$

*There exists a $(k+2)$-IGN model $\Psi$ such that:*

$$\Psi(\mathcal{G}) \neq \Psi(\mathcal{G}'). \tag{58}$$

**Corollary E.7.** *$k$-HOD-GNN is unable to distinguish any pair of $(k+1)$-FWL indistinguishable graphs.*

*Proof.* First, recall that the $k$-HOD GNN $HOD - GNN$ is composed of a base MPNN $\mathcal{M}$, a downstream MPNN $\mathcal{T}$, a $(k+1)$-IGN encoder $\mathrm{U}^{\text{node}}$ and a $k$-IGN encoder $\mathrm{U}^{\text{out}}$, all of which are less expressive than the $(k+2)$-IGN architecture. Thus, it is enough to show that $(k+2)$-IGN is able to simulate the efficient derivative algorithm presented in Appendix D, to compute the $k$-order derivative tensors $\mathsf{D}^{(T)}$ and $\mathsf{D}^{(\text{out})}$. We now show how $(k+2)$-IGN is able to simulate each step of this algorithm.

Before we begin, recall that a $(k+2)$-IGN operates on tensors $\mathsf{T} \in \mathbb{R}^{n^{k+2} \times d}$ where $d$ is called "the feature dimension". As the derivative tenors are of the form $\mathsf{D} \in \mathbb{R}^{n^{k+1} \times d' \times m^{d \times k}}$ we slightly change the notation for the tensors $T$ which $(k+2)$-IGN operates on to $\mathsf{T} \in \mathbb{R}^{n^{k+2} \times d_1, \cdots \times d_k}$ allowing multiple feature dimensions. We stress this is only a notation convenience, as we can transform $T$ back to a single feature dimension simply by "flattening" the different feature dimensions. Thus, for the rest of the proof, similarly to definition C.1 we assume $T$ is indexed by $T[\boldsymbol{u}, i, \boldsymbol{\alpha}]$ where $\boldsymbol{u} = (u_1, \dots, u_{k+2}) \in V(\mathcal{G})^{k+2}$, $i \in [d], \boldsymbol{\alpha} = (\boldsymbol{\alpha}_{j_1, j_2})_{d,k} \in \{0, \dots, m-1\}^{d \times k}$ and $|\boldsymbol{\alpha}| = \sum \boldsymbol{\alpha}_{j_1, j_2}$.

Throughout the proof, tensor representations corresponding to $\mathcal{G}, \mathcal{G}'$ will be denoted by $\mathsf{T}, \mathsf{T}'$ respectively.

**Step 1: Computing $\mathsf{D}^{(0)}$.**

For an input graph $\mathcal{G} = (\boldsymbol{A}, \boldsymbol{X})$ Recall that:

Since $\boldsymbol{h}^{(0)} = \boldsymbol{X}$, we have:

$$\mathsf{D}^{(0)}[v, \boldsymbol{u}, i, \boldsymbol{\alpha}] = \begin{cases} 1 & \text{if } \exists s \text{ s.t. } v = \boldsymbol{u}_s, \boldsymbol{\alpha}_{s,i} = 1, \sum_{s' \neq s, j \in [d]} \boldsymbol{\alpha}_{s',j} = 0 \\ 0 & \text{otherwise.} \end{cases} \tag{59}$$

The initial tensor $\mathsf{T}^{(0)}$ used by a $(k+2)$-IGN is such that $T[\boldsymbol{u}, \dots] = T[\boldsymbol{v}, \dots]$ if and only if the map $v_i \to u_i$ is a graph isomorphism on the subgraphs of $\mathcal{G}$ induced by $\boldsymbol{v}$ and $\boldsymbol{u}$ respectively.

In addition, one of the core operations of $(k+2)$-IGNs allows it to apply a "pointwise" linear layer followed by an activation on any entry $\mathbf{T}[\boldsymbol{u}, \dots]$ of the tensor $\mathbf{T}$ simultaneously. That is, using $(k+2)$-IGN layers we can update the tensor $\mathbf{T}$ via $\text{MLP}(\mathbf{T}[\boldsymbol{u}, \dots]) \to \mathbf{T}[\boldsymbol{u}, \dots]$ (See Maron et al. (2019); Frasca et al. (2022) for more details).

Define

$$S_1 = \{\mathbf{T}^{(0)}[\boldsymbol{u}, \dots] | u_1 = u_2\} \cup \{\mathbf{T}'^{(0)}[\boldsymbol{u}, \dots] | u_1 = u_2\}. \tag{60}$$

$$S_2 = \{\mathbf{T}^{(0)}[\boldsymbol{u}, \dots] | u_1 \in \mathcal{N}_G(u_2)\} \cup \{\mathbf{T}'^{(0)}[\boldsymbol{u}, \dots] | u_1 \in \mathcal{N}_{G'}(u_2)\}. \tag{61}$$

$$S_3 = \{\mathbf{T}^{(0)}[\boldsymbol{u}, \dots] | u_1 \notin (\mathcal{N}_G(u_2) \cup \{u_2\})\} \cup \{\mathbf{T}'^{(0)}[\boldsymbol{u}, \dots] | u_1 \notin (\mathcal{N}_{G'}(u_2) \cup \{u_2\})\}. \tag{62}$$

Here $\mathcal{N}(\cdot)$ is the neighborhood of a node.

From the definition of the initial tensor $\mathbf{T}^{(0)}$ above, $S_1, S_2, S_3$ all have pairwise empty intersections. Thus, we can define an MLP such that $\forall \boldsymbol{x} \in S_1$

$$\text{MLP}(\boldsymbol{x})[i, \boldsymbol{\alpha}] = \begin{cases} 1 & \text{if } \exists s \text{ s.t. } \boldsymbol{\alpha}_{s,i} = 1, \sum_{s' \neq s, j \in [d]} \boldsymbol{\alpha}_{s',j} = 0 \\ 0 & \text{otherwise,} \end{cases} \tag{63}$$

$\forall \boldsymbol{x} \in S_2$:

$$\text{MLP}(\boldsymbol{x})[i, \boldsymbol{\alpha}] = 1, \tag{64}$$

and $\forall \boldsymbol{x} \in S_3$:

$$\text{MLP}(\boldsymbol{x})[i, \boldsymbol{\alpha}] = 0. \tag{65}$$

by updating $\mathbf{T}[\boldsymbol{u}, \dots] = \text{MLP}(\mathbf{T}^{(0)}[\boldsymbol{u}, \dots])$ , $\mathbf{T}'[\boldsymbol{u}, \dots] = \text{MLP}(\mathbf{T}'^{(0)}[\boldsymbol{u}, \dots])$ we now get:

$$\mathbf{T}[u_1, \dots, u_{k+1}, u_{k+1}, \dots] = \mathbf{D}^{(0)}[u_2, \dots, u_{k+2}, \dots] \tag{66}$$

and for $u_1 \neq u_2$ :

$$\mathbf{T}[u_1, u_2, \dots, u_{k+2}, \dots] = \boldsymbol{A}_{u_1, u_2}. \tag{67}$$

Thus $\mathbf{T}$ now stores both the information of the tensor $\mathbf{D}^{(0)}$ and the adjacency $\boldsymbol{A}$.

**Step 2: Computing $\tilde{\mathbf{D}}^{(t-1)}$ from $\mathbf{D}^{(t-1)}$.**

For simplicity assume the base MPNN used in $k$-HOD-GNN is a GIN architecture (the proof can be easily generalized to the general case)

Recall from Appendix D that

$$\tilde{\mathbf{D}}^{(t-1)}[u, \dots] = \mathbf{D}^{(t-1)}[u, \dots](1 + \epsilon) \sum_{u' \in \mathcal{N}(u)} \mathbf{D}^{(t-1)}[u', \dots]. \tag{68}$$

This amounts to constructing a "flattened" node feature vector $\mathbf{D}_{\text{flat}} \in \mathbb{R}^{n \times \tilde{d}}$ defined by

$$\mathbf{D}_{\text{flat}}[u] = \mathbf{D}^{(t-1)}[u, \dots].\text{flatten}() \tag{69}$$

and performing standard message passing on it.

We can follow a similar path, defining a "flattened" matrix $\mathbf{T}_{\text{flat}} \in \mathbb{R}^{n^2 \times \tilde{d}}$ defined by

$$\mathbf{T}_{\text{flat}}[u_1, u_2] = \mathbf{T}[u_1, u_2, \dots].\text{flatten}() \tag{70}$$

and then use $(k+2)$-IGN layer to perform 2-IGN updates on $\mathbf{T}_{\text{flat}}$. From equation 67 the tensor $\mathbf{T}$ retains the adjacency information of the input graph. Following arguments

presented in Maron et al. (2019) a 2-IGN layer can simulate message passing. Thus, we are able to compute $\tilde{\mathbf{D}}^{(t-1)}$, completing the step.

**Step 3: Computing $\mathbf{D}^{(t)}$ from $\tilde{\mathbf{D}}^{(t-1)}$.** As shown in Appendix D, $\mathbf{D}^{(t)}$ can be computed from $\tilde{\mathbf{D}}^{(t-1)}$ by a point-wise update. That is, there exists a continuous function $f$ which depends on the choice of the activation and weights of the base MPNN $\mathcal{M}$ such that

$$\mathbf{D}^{(t)}[\boldsymbol{u}, i, \boldsymbol{\alpha}] = f(\tilde{\mathbf{D}}^{(t-1)}[\boldsymbol{u}, i, \boldsymbol{\alpha}]). \tag{71}$$

We can thus choose an MLP which approximates $f$ to finite precision and update our tensors $\mathbf{T}, \mathbf{T}'$ according to $\mathbf{T}[\boldsymbol{u}, \ldots] = \mathrm{MLP}(\mathbf{T}^{[}\boldsymbol{u}, \ldots])$ , $\mathbf{T}'[\boldsymbol{u}, \ldots] = \mathrm{MLP}(\mathbf{T}'[\boldsymbol{u}, \ldots])$. This finishes the current step.

Iteratively updating the tensors $\mathbf{T}$ and $\mathbf{T}'$ according to steps 1-3, we reach final tensors such that

$$\mathbf{T}[u_2, u_2 \ldots,, u_{k+2}, \ldots] = \mathbf{D}^{(T)}[u_2, \ldots, u_{k+2}, \ldots] \tag{72}$$

$$\mathbf{T}'[u_2, u_2, \ldots, u_{k+2}, \ldots] = \mathbf{D}'^{(T)}[u_2, \ldots, u_{k+2}, \ldots] \tag{73}$$

a $(k+2)$-IGN can then apply a projection $P : \mathbb{R}^{n^{k+2} \times d_1, \cdots \times d_k} \to \mathbb{R}^{n^{k+1} \times d_1, \cdots \times d_k}$

defined by

$$P(T)[u_1, u_2, \cdots, u_{k+2}, \ldots] = T[u_2, u2, \ldots, u_{k+2}, \ldots] = \mathbf{D}^{(T)}[u_2, \ldots, u_{k+2}, \ldots]. \tag{74}$$

Thus a $(k+2)$-IGN can recover $\mathbf{D}^{(T)}$.

**Step 4: Computing $\mathbf{D}^{\mathbf{out}}$ from $\mathbf{D}^{(T)}$.** For most common MPNNs, this step decomposes into a linear aggregation (which can be handled exactly like step 2) followed by an MLP (which is equivalent to the update of step 3). Consequently, the derivative tensor $\mathbf{D}^{\mathbf{out}}$ can be computed from $\mathbf{D}^{(T)}$ using the same primitives described above.

$\square$

*Proof of corollary E.7.* Recall that the $(k+1)$-FWL has equivalent expressive power to the $(k+2)$-oblivious WL test (see Morris et al. (2023)). In addition, as shown in Geerts (2020); Azizian & Lelarge (2020); Maron et al. (2019), $(k+2)$-IGNs have the same expressive power as the $(K+2)$ oblivious WL test. This together with Theorem E.6 completes the proof. $\square$

**Theorem 4.3.** We now formally state and prove Theorem 4.3

**Theorem E.8** (HOD-GNN is strictly more expressive than RWSE+MPNN)**.** *For any MPNN $\mathcal{T}$ augmented with random walk structural encodings (see Definition E.3), there exists a 1-HOD-GNN model $\Phi$ that uses ReLU activations and only first-order derivatives such that, for every graph $\mathcal{G}$, it holds that*

$$\mathcal{T}(\mathcal{G}) = \Phi(\mathcal{G}). \tag{75}$$

*Moreover, there exist a pair of graphs $\mathcal{G}^1$ and $\mathcal{G}^2$ such that for every RWSE-augmented MPNN $\mathcal{T}$,*

$$\mathcal{T}(\mathcal{G}^1) = \mathcal{T}(\mathcal{G}^2), \tag{76}$$

*yet there exists a 1-HOD-GNN model $\Phi$, using ReLU activations and only first-order derivatives, such that*

$$\Phi(\mathcal{G}^1) \neq \Phi(\mathcal{G}^2). \tag{77}$$

*Proof.* To prove the first part of the theorem for RSWE with $L$ number of steps, we begin with a simple preprocessing step. For each input graph $\mathcal{G} = (\boldsymbol{A}, \boldsymbol{X})$ with node feature matrix

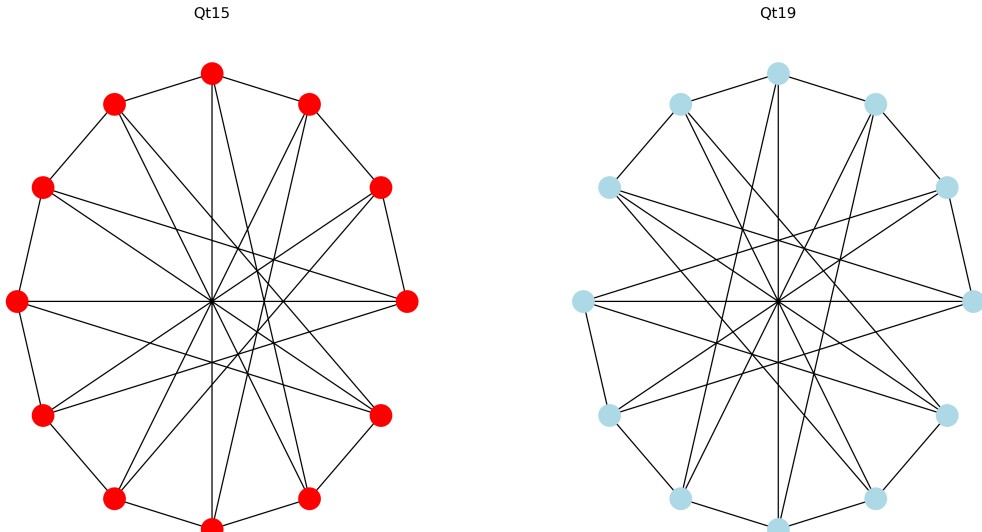

Figure 2: Two quartic vertex-transitive graphs that cannot be distinguished by MPNNs augmented with RWSE, but can be distinguished using HOD-GNN.

$\boldsymbol{X} \in \mathbb{R}^{n \times d}$, we define an extended feature matrix $\bar{\boldsymbol{X}} = \boldsymbol{X} \oplus \boldsymbol{1}_L \in \mathbb{R}^{n \times (d+L)}$ by padding $\boldsymbol{X}$ with a constant vecotr of lenght $L$ and value 1. We then pass $\bar{\boldsymbol{X}}$ to the HOD-GNN model instead of the original $\boldsymbol{X}$.

It is now sufficient to construct a base MPNN $\mathcal{M}$ with $T$ layers such that:

1. The first $d$ coordinates of each final node embedding satisfy $\boldsymbol{h}_{v,0:d-1}^{(T)} = \boldsymbol{X}_v$, i.e., the original features are preserved.

2. The first-order derivatives of the remaining $L$ coordinates (i.e., for indices $l = d, \ldots, d+L-1$) satisfy:

$$\frac{\partial \boldsymbol{h}_{v,l}^{(T)}}{\partial \bar{\boldsymbol{X}}_{v,l}} = \tilde{\boldsymbol{A}}_{v,v}^l, \tag{78}$$

where $\tilde{\boldsymbol{A}}$ is the row-normalized adjacency matrix (see Definition E.3).

If the above conditions hold, we can choose an MLP such that

$$\boldsymbol{h}_v^{\mathrm{der}} = \mathsf{MLP}\left(\boldsymbol{h}_v^{(T)} \oplus \mathrm{U}^{(T)}(\mathbf{D}^{(T)})_v \oplus \mathrm{U}^{\mathrm{out}}(\mathbf{D}^{\mathrm{out}})\right) = \boldsymbol{X}_v \oplus \boldsymbol{h}_v^{\mathrm{rwse}} \tag{79}$$

and choose our downstream model to be exactly $\mathcal{T}$ thus satisfying Equation 76.

We now construct an MPNN $\mathcal{M}$ that satisfies both of the conditions above.

For each layer $t = 0, \ldots, T-1$, we define the update rule of $\mathcal{M}$ to act separately on two parts of the node feature vector: the first $d+t$ coordinates and the remaining $T-t$ coordinates. Specifically:

$$\boldsymbol{h}_{v,0:d+t-1}^{(t+1)} = \boldsymbol{h}_{v,0:d+t-1}^{(t)}, \tag{80}$$

$$\boldsymbol{h}_{v,d+t:T-1}^{(t+1)} = \mathrm{ReLU}\left(\frac{1}{\deg(\mathrm{v})} \sum_{u \in \mathcal{N}(v)} \boldsymbol{h}_{u,d+t:T-1}^{(t)}\right). \tag{81}$$

.

We set the number of layers in $\mathcal{M}$ to be $T = L$, where $L$ is the number of random walk steps used in the original RWSE encoding.

First, the proposed update rule is straightforward to implement within the MPNN framework defined by Equation 1, as it follows a standard message-passing structure.

Second, Equation 80 guarantees that the first $d$ coordinates of each node's feature vector remain unchanged throughout the layers, thereby satisfying condition (1).

Finally, Equations 80 and 81 together imply that for all $t = 0, \ldots, T - 1$, it holds that

$$\boldsymbol{h}_{v,t}^{(T)} = \tilde{A}^t \cdot \mathbf{1}_T. \tag{82}$$

This implies that

$$\frac{\partial \boldsymbol{h}_{v,t}^{(T)}}{\partial \bar{\boldsymbol{X}}_{v,t}} = \tilde{\boldsymbol{A}}_{v,v}^t \tag{83}$$

and so condition (2) holds, completing the first part of the proof.

For the second part of the proof, following the notation of (Read & Wilson, 1998), let $\mathcal{G}_1$ and $\mathcal{G}_2$ denote the quartic vertex-transitive graphs Qt15 and Qt19, respectively (see Figure 2) with all initial node features equal to $\mathbf{1}_2 \in \mathbb{R}^2$. Here, "vertex-transitive" means that for any pair of nodes, there exists a graph automorphism mapping one to the other, and "quartic" indicates that the graphs are 4-regular. This pair of graphs was shown in (Southern et al., 2025) to be indistinguishable by MPNNs augmented with RWSE. Therefore, to conclude the proof, it suffices to construct an HOD-GNN model that can distinguish between them.

We define the base MPNN $\mathcal{M}$ to consist of two layers, specified as follows. The first layer performs standard neighbor aggregation:

$$\boldsymbol{h}_v^{(1)} = \mathrm{ReLU}\left(\sum_{u \in \mathcal{N}(v)} \boldsymbol{h}_u^{(0)}\right). \tag{84}$$

In the second layer, the first coordinate of each node feature is preserved, while the second coordinate is updated via another aggregation step. That is

$$\boldsymbol{h}_{v,0}^{(2)} = \boldsymbol{h}_{v,0}^{(1)}, \tag{85}$$

$$\boldsymbol{h}_{v,1}^{(2)} = \mathrm{ReLU}\left(\sum_{u \in \mathcal{N}(v)} \boldsymbol{h}_{u,1}^{(1)}\right). \tag{86}$$

.

from the same argument as the first part of the proof, we get that for $\mathcal{M}$, it holds that

$$\frac{\partial \boldsymbol{h}_{v,l}^{(2)}}{\partial \boldsymbol{X}_{u,l}} = \boldsymbol{A}_{v,u}^l, \tag{87}$$

Note that for a pair of nodes $u, v$, we have $\boldsymbol{A}_{u,v} = 1$ and $\boldsymbol{A}_{u,v}^2 = 0$ if and only if $u \in \mathcal{N}(v)$ and, for all $u' \in \mathcal{N}(v)$, it holds that $u \notin \mathcal{N}(u')$—that is, there is no path of length exactly 2 from $u$ to $v$. A straightforward computation shows that no such node pairs exist in $\mathcal{G}^1$, whereas several such pairs appear in $\mathcal{G}^2$. Thus, the set of off-diagonal derivative vectors $\{\mathbf{D}^{(2),2}[u, v, 1, :] \mid u \neq v \in V(\mathcal{G}^2)\}$ contains values that do not appear in the corresponding set $\{\mathbf{D}^{(2),1}[u, v, 1, :] \mid u \neq v \in V(\mathcal{G}^1)\}$.

We define the node derivative encoder $\mathrm{U}^{\text{node}}$ as a 2-IGN model that operates in two steps. First, it constructs a filtered tensor $\bar{\mathbf{D}}$ by zeroing out all entries of $\mathbf{D}^{(2)}$ that are not off-diagonal or are farther than $\epsilon = \frac{1}{4}$ from the vector $(1, 0)$. Then, it performs row-wise summation over $\bar{\mathbf{D}}$ to produce node features:

$$\mathrm{U}^{\text{node}}(\mathbf{D}^{(2)})_v = \sum_{u \in V(\mathcal{G})} \bar{\mathbf{D}}[v, u]. \tag{88}$$

It follows that $\mathrm{U}^{\text{node}}(\mathbf{D}^{(2),1}) = \mathbf{0}$, while $\mathrm{U}^{\text{node}}(\mathbf{D}^{(2),2}) \neq \mathbf{0}$. Therefore, when these vectors are passed to the downstream MPNN, it will be able to distinguish between the two graphs, completing the proof. $\qquad\square$

**Computational complexity.**

We now prove Propositions 4.4 and 4.5, which analyze the time and space complexity of HOD-GNN.

*Proof.* Recall that in Appendix D, we have defined the derivative sparsity of a node $v$ of an input graph $\mathcal{G}$ at layer $t$ of the base MPNN, as

$$s_{v,t} = \left| \left\{ (\boldsymbol{u}, i, \boldsymbol{\alpha}) \,\middle|\, \mathbf{D}^{(t)}[v, \boldsymbol{u}, i, \boldsymbol{\alpha}] \neq 0 \right\} \right|, \tag{89}$$

and the maximal derivative sparsity at layer $t$ is

$$s_t = \min_{v \in V(\mathcal{G})} s_{v,t}. \tag{90}$$

the tensor $\mathbf{D}^{(t)}$ has memory complexity of $O(n \cdot s_t)$ as it can be stored using sparse matrix representations. Algorithm 1 for efficient derivative computation iteratively computes $\mathbf{D}^{(t)}$ using $\mathbf{D}^{(t-1)}$ and finally computes $\mathbf{D}^{\text{out}}$ from $\mathbf{D}^{(T)}$. We now prove by induction that for an input graph $\mathcal{G}$ with maximal degree $d$[11], there exists a constant $C$ which depends on the maximal number of derivatives, the input node feature dimension, and the dimension of $\boldsymbol{h}^{(t)}$, all of which are hyperparameters, such that

$$s_t < C \cdot \min\{d^{k \cdot t}, n^k\}. \tag{91}$$

We additionally show that $\mathbf{D}^{(t)}$ can be constructed from $\mathbf{D}^{(t-1)}$ in time complexity of $O(d \cdot n \cdot \min\{d^{k \cdot (t-1)}, n^k\})$

To begin, as we saw in Appendix D,

$$\mathbf{D}^{(0)}[v, \boldsymbol{u}, i, \boldsymbol{j}, \boldsymbol{\alpha}] = \begin{cases} 1 & \text{if } \exists s \text{ s.t. } v = \boldsymbol{u}_s, i = \boldsymbol{j}_s, \boldsymbol{\alpha}_s = 1, \sum_{s' \neq s} \boldsymbol{\alpha}_{s'} = 0 \\ 0 & \text{otherwise,} \end{cases} \tag{92}$$

and so $s_{v,0} < C = m^k \cdot l^{k+1}$ where $m$ is the maximal derivative degree, and $l$ is the dimension of the input node features. Now assuming Equation 91 holds for $t - 1$, in Appendix D we saw that $\mathbf{D}^{(t)}$ can be computed from $\mathbf{D}^{(t-1)}$ in time complexity of $O(d \cdot n \cdot s_{t-1}) = O(d \cdot n \cdot \min\{d^{k \cdot (t-1)}, n^k\})$, secondly, we saw that $s_t = O\left(\min\{d \cdot s_{t-1}, n^k\}\right) = O\left(\min\{d^{k \cdot t}, n^k\}\right)$ completing the induction. As the memory complexity of storing $\mathbf{D}^{(t)}$ is $O(n \cdot s_t) = O\left(n \cdot \min\{d^{k \cdot t}, n^k\}\right)$ this proves Proposition 4.4. Recall now that we have seen in the proof of theorem E.4 that k-HOD-GNN is able to achieve the same expressivity as $k$-OSAN by disregarding $\mathbf{D}^{\text{out}}$ and choosing the encoder $\mathrm{U}^{\text{node}}$ to be

$$\mathrm{U}^{\text{ds-node}}(\mathbf{D}^{(T)})_v = \text{DeepSet}(\{\mathbf{D}^{(T)}[v, \boldsymbol{u}, i, \boldsymbol{\alpha}] \mid \boldsymbol{u} \in V^k(\mathcal{G}), i \in [d'], \in [d]^k, \boldsymbol{\alpha} \in [m]^k\}). \tag{93}$$

This encoder can leverage the sparsity of $\mathbf{D}^{(T)}$, having runtime complexity of $O(n \cdot s_T) = O(n \cdot \min\{n^k, d^{k \cdot T}\})$, completing the proof of proposition 4.5.

$\qquad\square$

---

[11]We slightly abuse notation by using $d$ to denote the maximal degree as it usually denotes the input feature dimension, we denote this quantity as $l$ for this discussion.

## F    ADDITIONAL PROPOSITIONS

In this section, we present three additional propositions which further explore the exact expressivity gains obtained from using derivative signals.

The first proposition shows that even when using only output-level derivatives, 1-HOD-GNN is still fairly expressive.

**Proposition F.1.** *Even when setting $U^{node} = 0$, 1-HOD-GNN is still as expressive as the DS-WL for node based policies test defined in Bevilacqua et al. (2021).*

*Proof.* We begin by recalling the definition of the DS-WL test. Consider two graphs $\mathcal{G}_1 = (\boldsymbol{A}_1, \boldsymbol{X}_1)$ and $\mathcal{G}_2 = (\boldsymbol{A}_2, \boldsymbol{X}_2)$. For each graph, the DS-WL test constructs a bag of node-marked subgraphs, where each subgraph is obtained by appending a unique mark to the feature of exactly one node. Formally, for $i \in \{1, 2\}$,

$$\mathcal{B}_i = \{ \mathcal{S}_{i,v} \mid v \in V(\mathcal{G}_i) \}, \tag{94}$$

where each marked subgraph is defined as

$$\mathcal{S}_{i,v} = (\boldsymbol{A}_i, \boldsymbol{X}_i \oplus \boldsymbol{e}_v), \tag{95}$$

with $e_v$ denoting a one-hot indicator applied to node $v$.

The DS-WL test then applies the WL refinement procedure independently to each marked subgraph $\mathcal{S}_{i,v}$ until convergence, and assigns it a final color

$$c_S = \mathrm{WL}(\mathcal{S}). \tag{96}$$

This produces, for each graph, a multiset of resulting WL colors:

$$C_i = \{ c_{\mathcal{S}} \mid \mathcal{S} \in \mathcal{B}_i \}, \qquad i \in \{1, 2\}. \tag{97}$$

Finally, DS-WL declares $\mathcal{G}_1$ and $\mathcal{G}_2$ to be **non-isomorphic** if and only if $C_1 \neq C_2$. If the two multisets coincide, the test does not distinguish the graphs.

As shown in Morris et al. (2019), for any finite collection of graphs there exists an MPNN $\mathcal{M}$ whose output is equivalent to the WL coloring. In particular,

$$\mathcal{M}(\mathcal{S}) = \mathcal{M}(\mathcal{S}') \iff c_{\mathcal{S}} = c_{\mathcal{S}'}, \tag{98}$$

for any marked subgraphs $\mathcal{S}$ and $\mathcal{S}'$.

Consequently, $\mathcal{G}_1$ and $\mathcal{G}_2$ are separated by the DS-WL test if and only if there exists an MPNN such that

$$\{ \mathcal{M}(\mathcal{S}) \mid \mathcal{S} \in \mathcal{B}_1 \} \neq \{ \mathcal{M}(\mathcal{S}) \mid \mathcal{S} \in \mathcal{B}_2 \}. \tag{99}$$

As previously discussed, we may assume without loss of generality that the activation function used by $\mathcal{M}$ is analytic. Indeed, if this is not the case, the MLP components of $\mathcal{M}$ can be approximated to arbitrary precision by MLPs employing an analytic activation function.

For every node $v \in V(\mathcal{G}_i)$, define the scalar function

$$f_v^i(x) = \mathcal{M}(\boldsymbol{A}_i, \boldsymbol{X}_i \oplus x\, \boldsymbol{e}_v), \tag{100}$$

where $x \in \mathbb{R}$.

We note the following properties:

1. $f_v^i(1) = \mathcal{M}(\mathcal{S}_{i,v})$ by definition of the marked subgraph $\mathcal{S}_{i,v}$.

2. For every $j \in \mathbb{N}$,

$$\left. \frac{\partial^j}{\partial x^j} f_v^i(x) \right|_{x=0} = \frac{\partial^j}{\partial \boldsymbol{X}_v^j} \mathcal{M}(\mathcal{S}_{i,v}), \tag{101}$$

   since the perturbation $x$ only modifies the feature at node $v$.

3. Each $f_v^i$ is analytic, as $\mathcal{M}$ is a composition of linear transformations and analytic activation functions.

Since analytic functions are uniquely determined by their derivatives at a point, we obtain that for any pair of nodes $v \in V(\mathcal{G}_1)$ and $u \in V(\mathcal{G}_2)$,

$$\left( \frac{\partial^j}{\partial \boldsymbol{X}_v^j} \mathcal{M}(\mathcal{S}_{1,v}) = \frac{\partial^j}{\partial \boldsymbol{X}_u^j} \mathcal{M}(\mathcal{S}_{2,u}) \text{ for all } j \in \mathbb{N} \right) \implies \mathcal{M}(\mathcal{S}_{1,v}) = \mathcal{M}(\mathcal{S}_{2,u}). \quad (102)$$

Therefore, if $\mathcal{M}(\mathcal{S}_{1,v}) \neq \mathcal{M}(\mathcal{S}_{2,u})$, analyticity implies that there exists some integer $m$ such that

$$\frac{\partial^m}{\partial \boldsymbol{X}_v^m} \mathcal{M}(\mathcal{S}_{1,v}) \neq \frac{\partial^m}{\partial \boldsymbol{X}_u^m} \mathcal{M}(\mathcal{S}_{2,u}), \quad (103)$$

and consequently

$$\mathbf{D}^{\text{out}}(\mathcal{S}_{1,v}) \neq \mathbf{D}^{\text{out}}(\mathcal{S}_{2,u}). \quad (104)$$

Hence,

$$\{ \mathcal{M}(\mathcal{S}) \mid \mathcal{S} \in \mathcal{B}_1 \} \neq \{ \mathcal{M}(\mathcal{S}) \mid \mathcal{S} \in \mathcal{B}_2 \} \quad (105)$$

can hold only if

$$\left\{ \mathbf{D}^{\text{out}}(\mathcal{S}) \mid \mathcal{S} \in \mathcal{B}_1 \right\} \neq \left\{ \mathbf{D}^{\text{out}}(\mathcal{S}) \mid \mathcal{S} \in \mathcal{B}_2 \right\}. \quad (106)$$

Finally, by choosing the encoder $\text{U}^{\text{out}}$ to be a fully expressive multiset function (e.g., DeepSets Zaheer et al. (2017)), 1-HOD-GNN is able to distinguish $\mathcal{G}_1$ from $\mathcal{G}_2$ using only the derivative tensor $\mathbf{D}^{\text{out}}$. This completes the proof.

$\square$

The following proposition analyses the expressive power of edge-HOD-GNN.

**Proposition F.2.** *edge-HOD-GNN is as expressive as the DS-WL for edge based policies test defined in Bevilacqua et al. (2021), and can thus separate pairs of 2-FWL indistinguishable graphs.*

*Proof.* The proof of this proposition follows the same argument as the previous one; we include it here for completeness. For brevity, for the rest of this prove we refer to the edge-marking WL test simply as the WL test, and the edge-derivative tensor simply as the derivative tensor, denoted by $\mathbf{D}^{\text{out}}$. We begin by recalling the definition of the DS-WL test with edge marking. Consider two graphs $\mathcal{G}_1 = (\boldsymbol{A}_1, \boldsymbol{X}_1, \boldsymbol{E}_1)$ and $\mathcal{G}_2 = (\boldsymbol{A}_2, \boldsymbol{X}_2, \boldsymbol{E}_2)$ where $\boldsymbol{E}_1, \boldsymbol{E}_2$ represent edge feature matrices. For each graph, the DS-WL test constructs a bag of edge-marked subgraphs, where each subgraph is obtained by appending a unique mark to the feature of exactly one edge. Formally, for $i \in \{1, 2\}$,

$$\mathcal{B}_i = \{ \mathcal{S}_{i,e} \mid e \in E(\mathcal{G}_i) \}, \quad (107)$$

where each marked subgraph is defined as

$$\mathcal{S}_{i,e} = (\boldsymbol{A}_i, \boldsymbol{X}_i, \boldsymbol{E}_i \oplus \boldsymbol{e}_e), \quad (108)$$

with $\boldsymbol{e}_e$ denoting a one-hot indicator applied to edge $e$.

The DS-WL test then applies the WL refinement procedure independently to each marked subgraph $\mathcal{S}_{i,e}$ until convergence, and assigns it a final color

$$c_S = \text{WL}(\mathcal{S}). \quad (109)$$

This produces, for each graph, a multi-set of resulting WL colors:

$$C_i = \{ c_\mathcal{S} \mid \mathcal{S} \in \mathcal{B}_i \}, \qquad i \in \{1, 2\}. \quad (110)$$

Finally, DS-WL declares $\mathcal{G}_1$ and $\mathcal{G}_2$ to be **non-isomorphic** if and only if $C_1 \neq C_2$. If the two multi-sets coincide, the test does not distinguish the graphs.

As shown in Morris et al. (2019), for any finite collection of graphs there exists an MPNN $\mathcal{M}$ whose output is equivalent to the WL coloring. In particular,

$$\mathcal{M}(\mathcal{S}) = \mathcal{M}(\mathcal{S}') \iff c_\mathcal{S} = c_{\mathcal{S}'}, \quad (111)$$

for any marked subgraphs $\mathcal{S}$ and $\mathcal{S}'$.

Consequently, $\mathcal{G}_1$ and $\mathcal{G}_2$ are separated by the DS-WL test if and only if there exists an MPNN such that

$$\{\, \mathcal{M}(\mathcal{S}) \mid \mathcal{S} \in \mathcal{B}_1 \,\} \;\neq\; \{\, \mathcal{M}(\mathcal{S}) \mid \mathcal{S} \in \mathcal{B}_2 \,\}. \tag{112}$$

As previously discussed, we may assume without loss of generality that the activation function used by $\mathcal{M}$ is analytic. Indeed, if this is not the case, the MLP components of $\mathcal{M}$ can be approximated to arbitrary precision by MLPs employing an analytic activation function.

For every edge $e \in E(\mathcal{G}_i)$, define the scalar function

$$f_e^{\,i}(x) \;=\; \mathcal{M}(\boldsymbol{A}_i,\, \boldsymbol{X}_i, \boldsymbol{E}_i \oplus x \cdot \boldsymbol{e}_v), \tag{113}$$

where $x \in \mathbb{R}$.

We note the following properties:

1. $f_e^{\,i}(1) = \mathcal{M}(\mathcal{S}_{i,e})$ by definition of the marked subgraph $\mathcal{S}_{i,e}$.

2. For every $j \in \mathbb{N}$,

$$\left. \frac{\partial^j}{\partial x^j} f_e^{\,i}(x) \right|_{x=0} \;=\; \frac{\partial^j}{\partial \boldsymbol{E}_e^j} \mathcal{M}(\mathcal{S}_{i,e}), \tag{114}$$

   since the perturbation $x$ only modifies the feature at edge $e$.

3. Each $f_e^{\,i}$ is analytic, as $\mathcal{M}$ is a composition of linear transformations and analytic activation functions.

Since analytic functions are uniquely determined by their derivatives at a point, we obtain that for any pair of edges $e_i \in E(\mathcal{G}_i)$ $i \in [2]$,

$$\left( \frac{\partial^j}{\partial \boldsymbol{E}_{e1}^j} \mathcal{M}(\mathcal{S}_{1,e_1}) \;=\; \frac{\partial^j}{\partial \boldsymbol{E}_{e_2}^j} \mathcal{M}(\mathcal{S}_{2,e_2}) \text{ for all } j \in \mathbb{N} \right) \;\implies\; \mathcal{M}(\mathcal{S}_{1,e_1}) = \mathcal{M}(\mathcal{S}_{2,e_2}). \tag{115}$$

Therefore, if $\mathcal{M}(\mathcal{S}_{1,e_1}) \neq \mathcal{M}(\mathcal{S}_{2,e_2})$, analyticity implies that there exists some integer $m$ such that

$$\frac{\partial^m}{\partial \boldsymbol{E}_{e_1}^m} \mathcal{M}(\mathcal{S}_{1,e_1}) \;\neq\; \frac{\partial^m}{\partial \boldsymbol{E}_{e_2}^m} \mathcal{M}(\mathcal{S}_{2,e_2}), \tag{116}$$

and consequently

$$\mathbf{D}^{\mathrm{out}}(\mathcal{S}_{1,e_1}) \;\neq\; \mathbf{D}^{\mathrm{out}}(\mathcal{S}_{2,e_2}). \tag{117}$$

Hence,

$$\{\, \mathcal{M}(\mathcal{S}) \mid \mathcal{S} \in \mathcal{B}_1 \,\} \;\neq\; \{\, \mathcal{M}(\mathcal{S}) \mid \mathcal{S} \in \mathcal{B}_2 \,\} \tag{118}$$

can hold only if

$$\left\{\, \mathbf{D}^{\mathrm{out}}(\mathcal{S}) \mid \mathcal{S} \in \mathcal{B}_1 \,\right\} \;\neq\; \left\{\, \mathbf{D}^{\mathrm{out}}(\mathcal{S}) \mid \mathcal{S} \in \mathcal{B}_2 \,\right\}. \tag{119}$$

Finally, by choosing the encoder $\mathrm{U}_{\mathrm{edge}}^{\mathrm{out}}$ to be a fully expressive multiset function (e.g., DeepSets Zaheer et al. (2017)), edge-HOD-GNN is able to distinguish $\mathcal{G}_1$ from $\mathcal{G}_2$ using only the edge derivative tensor $\mathbf{D}^{\mathrm{out}}$. the edge marking WL test was shown in Bevilacqua et al. (2021) to be able to separate 2-FWL indistinguishable graphs. This completes the proof. $\square$

Finally, the next proposition shows that for a fixed base MPNN $\mathcal{M}$, increasing the hyperparameter $m$ (Definition 3.1), which specifies the highest derivative order used by HOD-GNN, strictly improves expressivity.

**Proposition F.3.** *For any $m \in \mathbb{N}$, there exist choices of the base MPNN $\mathcal{M}$ such that a 1-HOD-GNN using $\mathcal{M}$ with hyper-parameter $m$ is unable to count the number of triangles of a given input graph, while a $k$-HOD-GNN using $\mathcal{M}$ with $m+1$ can.*

*Proof.* Let $m \in \mathbb{N}$. We define the base message-passing network $\mathcal{M}_m$ as follows.

Consider an input graph $\mathcal{G} = (\boldsymbol{A}, \boldsymbol{X})$, where $\boldsymbol{X} \in \mathbb{R}^n$ is a node-feature matrix with scalar features.

**Initialization.** We define the first hidden representation by a node-wise MLP update (which ignores all neighbors):

$$\boldsymbol{h}_v^{(1)} := \frac{\boldsymbol{X}_v^{m+1}}{(m+1)!}. \tag{120}$$

This update depends only on $\boldsymbol{X}_v$ and is therefore realizable by a local node transformation.

**Message passing.** For $t \in \{2, 3, 4\}$, we define

$$\boldsymbol{h}^{(t)} := \boldsymbol{A}\boldsymbol{h}^{(t-1)}, \qquad \text{i.e.,} \qquad \boldsymbol{h}_v^{(t)} = \sum_{u \in N(v)} \boldsymbol{h}_u^{(t-1)}. \tag{121}$$

Each update is realizable by a standard MPNN aggregation step without applying a post-aggregation MLP.

**Readout.** The global output of $\mathcal{M}_m$ is defined by summing the final node representations:

$$\boldsymbol{h}^{\text{out}} := \sum_{v \in V} \boldsymbol{h}_v^{(4)}. \tag{122}$$

Thus, $\mathcal{M}_m$ is a message-passing neural network consisting of one node-wise MLP layer followed by three pure aggregation layers, and a final sum readout.

As a consequence of the chain rule, all partial derivatives of both $\boldsymbol{h}^{(4)}$ and $\boldsymbol{h}^{\text{out}}$ of order at most $m$ vanish. Therefore, a 1-HOD-GNN model restricted to the hyperparameter $m$ has expressive power equivalent to a standard MPNN, and in particular it cannot count triangles.

Moreover, we have

$$\frac{\partial^{m+1} \boldsymbol{h}_v^{(4)}}{\partial \boldsymbol{X}_v^{m+1}} = \left(\boldsymbol{A}^3\right)_{v,v}. \tag{123}$$

Hence, a 1-HOD-GNN model with hyperparameter $m + 1$ can exploit this derivative to compute

$$\sum_v \frac{\left(\boldsymbol{A}^3\right)_{v,v}}{6}, \tag{124}$$

which equals the number of triangles in the input graph $\mathcal{G}$. This completes the proof.

$\square$

## G  EXPERIMENTAL DETAILS

In this section, we provide details on the experimental validation described and discussed in Section 5.

### G.1  ARCHITECTURE

In all of our experiments, we have used a 1-HOD-GNN architecture, We now describe its components in detail.

**Base MPNN.** In all experiments, we use a GIN (Xu et al., 2018) architecture as the base MPNN—described in Equation 9 (Section 3)—due to its simplicity and maximal expressivity. All MLPs used in the node updates are two layers deep and employ ReLU activations. We extract first-order derivatives from the base MPNN, which—by Theorem 4.3—yields greater expressivity than MPNNs augmented with RWSE. After computing the final node representations $\boldsymbol{h}^{(T)}$, we apply a residual connection by aggregating all intermediate layers:

$$\boldsymbol{h}^{(T)} \leftarrow \bigoplus_{t=1}^{T} \frac{\boldsymbol{h}^{(t)}}{t!}. \tag{125}$$

where normalizing in $t!$ helped stabalize training.

**Derivative Encoding.**

For simplicity, we ignore the output derivatives of the base MPNN and use only the final node-wise derivative tensor $\mathbf{D}^{(T)}$. The encoder $\mathrm{U}^{\mathrm{node}}$ is implemented as a lightweight, efficient module that applies a pointwise MLP to the diagonal entries of $\mathbf{D}^{(T)}$:

$$\mathrm{U}^{\mathrm{node}}(\mathbf{D}^{(T)})_v = \mathsf{MLP}\left(\mathbf{D}^{(T)}[v, v, \dots]\right). \tag{126}$$

**Downstream GNN.**

For simplicity and to isolate the effects of HOD-GNN, we restrict our experiments to MPNNs as downstream architectures—We use GIN for all experiments but peptieds func in which we used a GCN (Kipf & Welling, 2016). All MLPs used in the node updates are two layers deep with ReLU activations. The final MLP head also uses ReLU activations and consists of 1 to 3 layers, following the default settings from Southern et al. (2025), without tuning. In experiments with parameter budgets, we adjust only the hidden dimension to fit within the limit, selecting the largest value that satisfies the constraint.

**Initialization.** We initialize the base MPNN such that all categorical feature embeddings are set to the constant vector $\mathbf{1}$, all MLP weights are initialized to the identity, and the $\epsilon$ parameters in the GIN update (Equation 9) are set to $-1$. This initialization is motivated by the proof of Theorem 4.3; following a similar line of reasoning, it implies that the diagonal of the derivative tensor $\mathbf{D}^{(T)}[v, v, \dots]$ corresponds exactly to the centrality encoding proposed in Southern et al. (2025).

**Optimization.** We use separate learning rates for the base and downstream MPNNs. For the downstream MPNN, we adopt the learning rates used in Southern et al. (2025), while for the base MPNN, we fine-tune by selecting either the same rate or one-tenth of it.

## G.2 Experiments

We provide below the details of the datasets and hyperparameter configurations used in our experiments. Our method is implemented using PyTorch (Paszke et al., 2019) and PyTorch Geometric (Fey & Lenssen, 2019), and is based on code provided in Southern et al. (2025) and Rampášek et al. (2022). Test performance is evaluated at the epoch achieving the best validation score and is averaged over four runs with different random seeds. We optimize all models using AdamW (Loshchilov & Hutter, 2017), with a linear learning rate warm-up followed by cosine decay. We track experiments and perform hyperparameter optimization using the Weights and Biases platform. All experiments were conducted on a single NVIDIA A100-SXM4-40GB GPU.

**OGB datasets.** We evaluate on three molecular property prediction benchmarks from the OGB suite (Hu et al., 2020b): MOLHIV, MOLBACE, and MOLTOX21. These datasets share a standardized node and edge featurization capturing chemophysical properties. We adopt the challenging scaffold split proposed in (Hu et al., 2020a). To prevent memory issues, we use a batch size of 128 for MOLHIV and 32 for the remaining datasets. All Downstream models use a hidden dimension of 300, consistent with prior work (Hu et al., 2020a; Bevilacqua et al., 2024). We sweep over several architectural choices, including the hidden dimension of the Base MPNN $k = 8, 10, \dots, 30, 32$, the initial learining rate of the Base MPNN $k = 0.001, 0.0001$, and the dropout rate $k = 0.0, 0.1, 0.2, 0.3, 0.4, 0.5$ . The number of layers in the base MPNN was selected to match the positional encoding step used in the corresponding experiment from Southern et al. (2025). Hyperparameter tuning was performed on the validation set using four random seeds. Results are reported at the test epoch corresponding to the best validation performance. All models were trained for 100 epochs. The final parameters used for each experiment are reported in table 3

**Zinc** The ZINC dataset (Dwivedi et al., 2023) includes 12k molecular graphs of commercially available chemical compounds, with the task of predicting molecular solubility. We follow the predefined dataset splits and report the Mean Absolute Error (MAE) as both the loss and evaluation metric. Our downstream MPNN for this task includes 6 message-passing layers and 3 readout layers, with a hidden size of 120 and no dropout. We use a batch size of 32 and train for 2000 epochs. We performed a small sweep over the depth of the base MNPNN $k = 10, 12, 14, 16, 18, 20$ and the hidden dimension $k = 30, 35, \dots, 80$. The hidden dimension

of the downstream MPNN was chose as 120 to meet the $500k$ parameter constraint. The final hyperparameters are listed in Table 4.

**Peptides** Peptides-func and Peptides-struct, introduced by Dwivedi et al. (2022), consist of graphs representing atomic peptides. Peptides-func is a multi-label classification benchmark with 10 nonexclusive peptide function labels, while Peptides-struct is a regression task involving 11 different structural attributes derived from 3D conformations.

For both datasets, we adopt the hyperparameter setup proposed by Tönshoff et al. (2023) for the downstream GNN, which has a parameter budget under 500k and where they use 250 epochs. We set the number of message-passing layers in our base MPNN with the positional encoding steps to be 20, aligned with the number of steps used for the random-walk structural encoding. The only tuned component is the learning rate of the base MPNN. The final configurations are summarized in Table 5.

**Key empirical findings.** Across all benchmarks, HOD-GNN is highly competitive and *the only architecture that consistently ranks within the top two model tiers.* Additionally, the strong performance of HOD-GNN on the large-scale Peptides datasets—where full-bag Subgraph GNNs are generally unable to run—shows its ability to scale effectively. Notably, the base MPNNs in HOD-GNN are often significantly deeper and narrower than those in typical GNNs. For instance, on the OGB datasets, the base MPNNs use 17-20 layers with hidden dimensions as low as 16–32. Despite their compact size, these base MPNNs yield notable performance gains over standard GINE, suggesting potential robustness to oversquashing. Moreover, the fact that they are significantly deeper than typical MPNNs further suggests that HOD-GNN may help mitigate oversmoothing, a hypothesis we leave for future work.

Table 3: Best-performing hyperparameters for each OGB dataset.

| Hyperparameter | MOLHIV | MOLBACE | MOLTOX21 |
|---|---|---|---|
| **Downstream model** | | | |
| #Layers | 2 | 8 | 10 |
| #Readout Layers | 1 | 3 | 3 |
| Hidden Dimension | 300 | 300 | 300 |
| Dropout | 0.0 | 0.5 | 0.3 |
| Learning Rate | 0.0001 | 0.0001 | 0.001 |
| **Base MPNN** | | | |
| #Layers | 16 | 20 | 20 |
| Hidden Dimension | 16 | 16 | 16 |
| Dropout | 0.2 | 0.5 | 0.2 |
| Learning Rate | 0.0001 | 0.0001 | 0.0001 |
| #Parameters | 450,848 | 1,723,448 | 2,165,782 |

Table 4: Best-performing hyperparameters for the ZINC dataset.

| Hyperparameter | ZINC |
|---|---|
| **Downstream model** | |
| #Layers | 6 |
| #Readout Layers | 3 |
| Hidden Dimension | 120 |
| Dropout | 0.0 |
| Learning Rate | 0.001 |
| **Base MPNN** | |
| #Layers | 12 |
| Hidden Dimension | 75 |
| Dropout | 0.0 |
| Learning Rate | 0.0001 |
| #Parameters | 498,144 |

Table 5: Best-performing hyperparameters for the Peptides-func and Peptides-struct datasets.

| Hyperparameter | Peptides-func | Peptides-struct |
|---|---|---|
| **Downstream model** | | |
| #Layers | 6 | 10 |
| #Readout Layers | 3 | 3 |
| Hidden Dimension | 234 | 143 |
| Dropout | 0.1 | 0.2 |
| Learning Rate | 0.001 | 0.001 |
| **Base MPNN** | | |
| #Layers | 20 | 20 |
| Hidden Dimension | 8 | 8 |
| Dropout | 0.1 | 0.2 |
| Learning Rate | 0.0001 | 0.001 |
| #Parameters | 498,806 | 493,849 |

Table 6: Normalized MAE results on the counting subgraphs dataset. Cells below 0.01 are highlighted in yellow.

| Method | 3-Cycle | 4-Cycle | 5-Cycle | 6-Cycle | Tailed Tri. | Chordal Cycle | 4-Clique | 4-Path |
|---|---|---|---|---|---|---|---|---|
| MPNN | 0.3515 | 0.2742 | 0.2088 | 0.1555 | 0.3631 | 0.3114 | 0.1645 | 0.1592 |
| MPNN+RWSE | 0.0645 | 0.0264 | 0.0746 | 0.0578 | 0.0505 | 0.1008 | 0.0905 | 0.0217 |
| GPS+RWSE | 0.0185 | 0.0433 | 0.0472 | 0.0551 | 0.0446 | 0.0974 | 0.0836 | 0.0284 |
| HyMN | 0.0384 | 0.0933 | 0.1350 | 0.0936 | 0.0084 | 0.0746 | 0.0680 | 0.0120 |
| GNN-AK+ | 0.0004 | 0.0040 | 0.0133 | 0.0238 | 0.0043 | 0.0112 | 0.0049 | 0.0075 |
| HOD-GNN + ReLU | 0.0012 | 0.0046 | 0.0210 | 0.0380 | 0.0083 | 0.0510 | 0.0293 | 0.0081 |
| HOD-GNN + SiLU | 0.0008 | 0.0042 | 0.0068 | 0.0222 | 0.0066 | 0.0195 | 0.0055 | 0.0069 |

## H ADDITIONAL EXPERIMENTS

### H.1 SUBSTRUCTURE COUNTING

We adopt the synthetic node-level subgraph counting experiment used in Huang et al. (2022); Yan et al. (2024). The dataset consists of 5,000 graphs generated from a mixture of distributions (see Zhao et al. (2022) for more details), with a train/validation/test split of 0.3/0.2/0.5. The task is node-level regression: predicting the number of substructures such as 3-cycles, 4-cycles, 5-cycles, 6-cycles, tailed triangles, chordal cycles, 4-cliques and 4-paths, where continuous outputs approximate discrete counts. We report the normalized MAE for each baseline, and highlight cases where the error falls below 0.001, since in these instances rounding the predictions yields exact counts.

For training, we use the AdamW optimizer with an initial learning rate of 0.001, a cosine scheduler with warmup, and train for 5,00 epochs. The batch size is set to 128. We compare HOD-GNN against both positional/structural encoding methods (MPNN+RWSE (Dwivedi et al., 2021), GPS+RWSE (Rampášek et al., 2022), HyMN (Southern et al., 2025)) and node-based subgraph GNNs (GNN-AK+ (Zhao et al., 2022), Nested GNN (Zhang & Li, 2021), ID-GNN (You et al., 2021), HyMN (Southern et al., 2025)).

To examine the role of activation functions, we evaluate HOD-GNN with the non-analytic ReLU and the analytic SiLU. The results, summarized in Table 6, are consistent with our theory. In line with Theorem 4.1, HOD-GNN achieves performance comparable to or surpassing other subgraph GNNs. Furthermore, consistent with Theorem 4.3, we observe that analytic activations enhance expressivity, while even with non-analytic ReLU, HOD-GNN still outperforms encoding-based methods.

### H.2 GRAPH SEPARATION ABILITY OF $k$-HOD-GNN

To further evaluate the expressive power of $k$-HOD-GNN and to empirically validate Theorem 4.1, we experimented with both 1-HOD-GNN and 2-HOD-GNN on the family of regular graph pairs from the BREC benchmark (Wang & Zhang, 2024). This dataset contains 140 pairs of regular graphs: 50 pairs that are distinguishable by 3-WL but not by 2-WL, and

Table 7: Results on separation of pairs of regular graphs from the BREC dataset.

| Model | Regular Graphs Number | Regular Graphs Accuracy |
|---|---|---|
| 3-WL | 50 | 35.7% |
| NGNN | 48 | 34.3% |
| DE+NGNN | 50 | 35.7% |
| DS-GNN | 48 | 34.3% |
| DSS-GNN | 48 | 34.3% |
| SUN | 50 | 35.7% |
| SSWL_P | 50 | 35.7% |
| GNN-$\bar{\text{A}}$K | 50 | 35.7% |
| KP-GNN | 106 | 75.7% |
| $\text{I}^2$-GNN | 100 | 71.4% |
| OSAN | 8 | 5.7% |
| 1-HOD GNN | 47 | 33.5% |
| 2-HOD GNN | 84 | 60.0% |

90 pairs that remain indistinguishable even under 3-WL. We follow the exact training and evaluation procedures proposed in Wang & Zhang (2024).

Table 7 reports the separation performance of our models alongside several subgraph-based GNNs. As shown, 2-HOD-GNN successfully distinguishes 34/90 of the 3-WL-indistinguishable pairs, placing it among the top-performing models and providing strong empirical support for the theoretical advantage predicted in Theorem 4.1.

Furthermore, 1-HOD-GNN obtains accuracy comparable to DS-GNN on the same benchmark, aligning with Theorem 4.1 for the $k = 1$ case and reinforcing that, without accessing higher-order derivatives, the model behaves similarly to standard MPNNs.

### H.3 Ablation on Derivative Order

To assess the impact of higher-order derivatives on HOD-GNN's expressive power, we conduct an ablation study on the most challenging substructure counting task: 6-cycle prediction, which consistently yields the highest MAE. Specifically, we evaluate models restricted to derivatives of order at most $k$, for $k = 0, \ldots, 4$. The results, presented in Table 8, demonstrate that increasing the maximal derivative order consistently improves MAE, with performance saturating at $k = 4$. All experiments use the analytic SiLU activation function. We hypothesize that the observed saturation arises because higher-order derivatives of SiLU rapidly diminish toward zero, limiting the additional expressive gain.

Table 8: Effect of maximal derivative order on 6-Cycle MAE.

| Maximal derivative order | 6-Cycle MAE |
|---|---|
| 0 | 0.1555 |
| 1 | 0.0275 |
| 2 | 0.0223 |
| 3 | 0.0221 |
| 4 | 0.0231 |

### H.4 Stability Analysis

We evaluate the stability of HOD-GNN both in terms of training dynamics and the behavior of the derivative tensor norm. Across all tasks, we consistently observe smooth and stable loss curves, even when incorporating higher-order derivatives. To further assess stability, we conduct two dedicated experiments.

**Training Loss Dynamics.** We examine convergence behavior on the 6-cycle subgraph counting task by varying the maximum derivative order $d \in 1, 2, 3, 4$. Figure 3 shows the training loss curves, showing consistent and reliable convergence across all derivative orders.

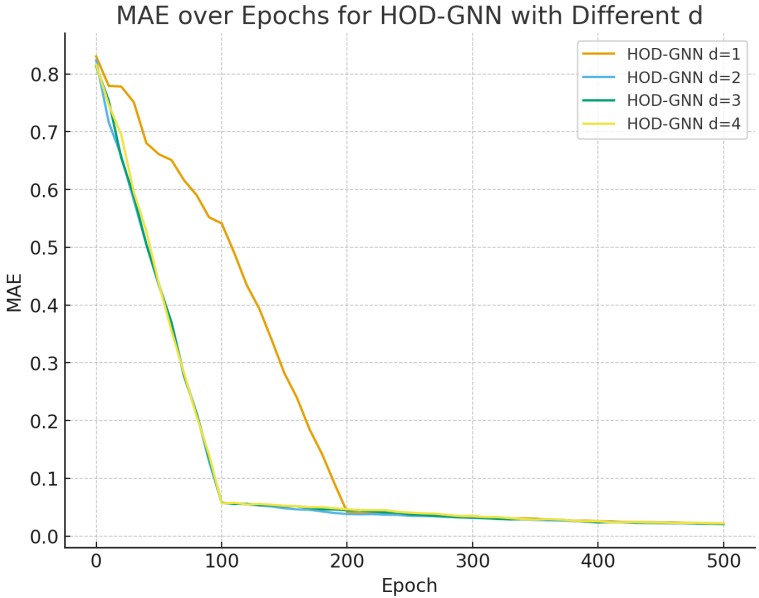

Figure 3: The training loss of HOD-GNN with maximal derivative $d \in \{1, 2, 3, 4\}$ on the 6-cycle counting task.

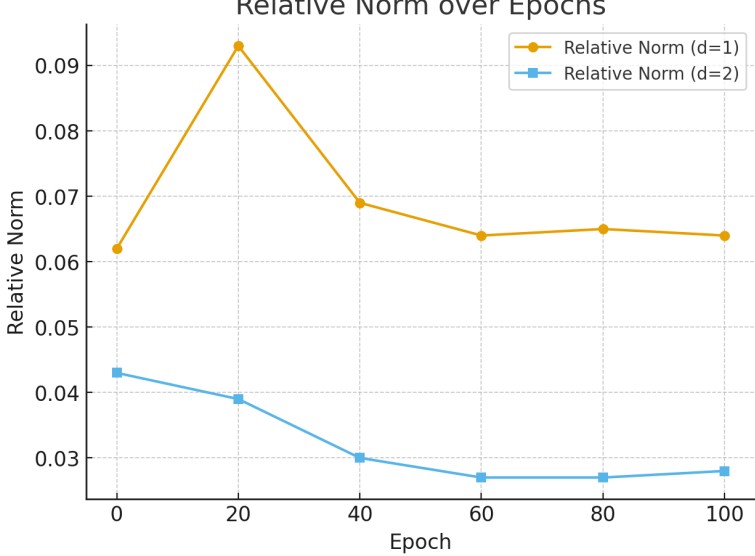

Figure 4: The relative norm of the derivative tensor with maximal derivative $d \in \{1, 2\}$ with respect to the norm of the final node feature matrix of the base MPNN on the MOLBACE dataset.

**Norm of the Derivative Tensor.** We additionally assess the stability of the derivative tensors used by HOD-GNN. We compute the norm of the derivative tensor during training on the MOLBACE dataset. Following standard practice (Higham, 2002), we report the relative norm: the ratio between the derivative tensor norm and the final node feature norm of the base MPNN. Figure 4 shows that the relative norm consistently remains significantly lower than that of the node features throughout training, confirming that HOD-GNN operates in a well-conditioned regime.

### H.5 ANALYSIS OF GENERALIZATION BEHAVIOR OF HOD-GNN

Recent studies (Franks et al., 2024; Maskey et al., 2025; Carrasco et al., 2025) have raised concerns about the generalization ability of highly expressive GNNs, showing that performance

Table 9: Comparison of model train-test performance gap on MOLHIV and MOLTOX21.

| Method | MOLHIV Train AUC | MOLHIV Test AUC | Gap-MOLHIV | MOLTOX21 Train AUC | MOLTOX21 Test AUC | Gap-MOLTOX21 |
|---|---|---|---|---|---|---|
| GCN | $88.65 \pm 2.19$ | $76.06 \pm 0.97$ | 12.59 | $92.06 \pm 1.81$ | $75.29 \pm 0.69$ | 16.77 |
| GIN | $88.64 \pm 2.54$ | $75.58 \pm 1.40$ | 13.06 | $93.06 \pm 0.88$ | $74.91 \pm 0.51$ | 18.15 |
| DSS-GNN (ED) | $91.71 \pm 3.50$ | $76.43 \pm 2.12$ | 15.28 | $92.38 \pm 1.57$ | $75.12 \pm 0.50$ | 17.26 |
| DSS-GNN (ND) | $89.70 \pm 3.20$ | $76.19 \pm 0.96$ | 13.51 | $91.23 \pm 2.15$ | $75.34 \pm 1.21$ | 15.89 |
| HOD-GNN | $88.15 \pm 0.51$ | $80.86 \pm 0.52$ | 7.29 | $93.73 \pm 1.09$ | $77.99 \pm 0.71$ | 15.74 |

can degrade when a model becomes "too expressive." These findings suggest that GNNs often generalize best when they have just the right capacity, a balance influenced both by architectural expressivity and by parameter scale (e.g., overly wide hidden dimensions can also harm generalization).

Motivated by these observations, we examine train-test gaps for HOD-GNN on the OGB datasets MOLHIV and MOLTOX21 (Table 9). HOD-GNN exhibits moderate gaps that are smaller than those of (i) less expressive architectures such as GCN and GIN, (ii) models of comparable expressivity such as DSS-GNN (ND), and (iii) more expressive variants such as DSS-GNN (ED). These findings indicate that HOD-GNN displays strong generalization. Baseline values are taken from Bevilacqua et al. (2021).

We propose two hypotheses as to why HOD-GNN generalizes well, informed by recent literature:

1. **Compact parameterization enabled by higher-order derivatives.** Across our experiments, the base MPNN within HOD-GNN consistently performs best with small hidden dimensions (8–32), resulting in models with relatively few parameters, thus aligning well with the "just the right capacity" approach mentioned above. We hypothesize that higher-order derivatives enrich these compact representations, effectively increasing expressive power without increasing model size, thereby supporting better generalization.

2. **Principled initialization that recovers RWSE.** The constructive proof following Theorem 4.2 gives an explicit initialization for HOD-GNN's derivative features that exactly matches Random Walk Structural Encodings (RWSE) - a widely used structural prior known to improve expressivity without overfitting. This suggests an alternative view of HOD-GNN:

   - At the beginning of training, the derivative-informed features remain close to RWSE, maintaining well known baseline behavior.
   - When the task benefits from additional capacity, the model can naturally move beyond RWSE and leverage higher-order derivative information.

   This yields a natural "just expressive enough" bias: HOD-GNN begins from a well-understood RWSE baseline and increases expressivity only when the data requires it.

We view further investigation of HOD-GNN's generalization properties as a promising direction for future work.

### H.6 COMPARISON OF RUNTIME AND MEMORY USAGE AGAINST SUBGRAPH GNNS

In section 5 we show that HOD-GNN is able to scale to the Peptides datasets, which are unreachable for full-bag subgraph GNNs on standard hardware, as stated in Southern et al. (2025); Bar-Shalom et al. (2023).To further demonstrate the scalability of HOD-GNN, we benchmark its runtime and memory usage against subgraph-based GNNs on the MOLHIV dataset. Since the choice of subgraph selection policy is the primary factor determining the asymptotic complexity of subgraph GNNs (Bevilacqua et al., 2021), we evaluate a range of policies using the default parameters from the original work. For subgraph GNNs, we adopt the hyperparameters reported in Bevilacqua et al. (2021) for MOLHIV. For HOD-GNN, we use the same hyperparameters as in Section 5, detailed in Appendix G.

**Results.** Table 10 reports GPU memory usage and per-epoch training and test runtimes. HOD-GNN achieves improvements in memory efficiency, requiring less than half the GPU memory compared to subgraph GNNs. In terms of runtime, HOD-GNN is faster than edge-deletion and node-deletion policies, while achieving comparable training time to ego-nets

Table 10: Runtime and memory comparison on the MOLHIV dataset. HOD-GNN demonstrates both improved memory efficiency and competitive runtime.

| GNN | GPU Memory (MiB) | Training Time / Epoch (s) | Test Time / Epoch (s) |
|---|---|---|---|
| edge-deletion | 32,944 | 62.13 | 5.70 |
| node-deletion | 29,826 | 58.80 | 4.01 |
| ego-nets | 25,104 | **53.16** | 3.07 |
| ego-nets+ | 25,211 | 54.19 | 3.20 |
| HOD-GNN | **12,964** | 53.34 | **2.28** |

Table 11: MAE comparison of different GNN architectures on ZINC-12K.

| GNN | MAE |
|---|---|
| GCN | $0.321 \pm 0.009$ |
| HOD-GNN + GCN | $0.080 \pm 0.006$ |
| GIN | $0.163 \pm 0.004$ |
| HOD-GNN + GIN | $0.066 \pm 0.003$ |
| GPS | $0.070 \pm 0.004$ |
| HOD-GNN + GPS | $0.064 \pm 0.002$ |

and ego-nets+. Notably, ego-net policies are known to be less expressive (Bevilacqua et al., 2021), highlighting that HOD-GNN achieves both efficiency and expressivity.

**Discussion.** The improvements primarily stem from the analytic computation of higher-order derivatives in HOD-GNN, which avoids the costly enumeration of subgraphs. While these advantages already translate to lower memory usage and faster runtimes in practice, we emphasize that HOD-GNN's scalability potential is not yet fully realized. In particular, it relies on efficient sparse matrix multiplications, which are currently suboptimally implemented in popular GNN libraries such as PyTorch Geometric. We therefore anticipate that further optimization of sparse kernels would amplify the scalability benefits of HOD-GNN.

### H.7 Ablation on Backbone GNN

In most experiments, we employ GIN (Xu et al., 2018) as the backbone of HOD-GNN. A key advantage of our approach, however, is its compatibility with any message-passing backbone. To assess the effect of backbone choice, we evaluate HOD-GNN on the ZINC dataset using GCN, GIN, and GPS as base architectures. The results, summarized in Table 11, demonstrate that HOD-GNN consistently improves upon its backbone across all settings. Among the tested architectures, GPS achieves the strongest performance, followed by GIN, with GCN ranking third—reflecting its lower expressivity relative to the other backbones.

## I Ethics Statement

This paper advances the theoretical and empirical study of graph neural networks through derivative-based architectures. Our experiments are conducted exclusively on widely used, publicly available benchmark datasets (ZINC, OGB, and Peptides molecular tasks), which contain no personally identifiable or sensitive information. We believe our work does not raise immediate ethical concerns. Nevertheless, we acknowledge that improvements in graph representation learning may be applied to sensitive domains (e.g., biological or social network data). We encourage responsible use of our methods in accordance with the ICLR Code of Ethics.

## J Reproducibility Statement

We have taken several steps to ensure reproducibility. All datasets used are standard benchmarks with clearly defined splits. A detailed description of architectures, hyperparameters, and training settings as well as full proofs of the theoretical results appear in the Appendix. Additionally, anonymized source code is provided as supplementary material and will be

released publicly upon publication. Together, these resources should allow researchers to fully reproduce our experimental and theoretical results.

