# OpenReview forum: "On The Expressive Power of GNN Derivatives"
_ICLR.cc/2026/Conference — ICLR 2026 Poster_

### Official Review · Reviewer_nnEj · 2025-10-16

**Soundness:** 2
**Presentation:** 3
**Contribution:** 3
**Rating:** 4
**Confidence:** 3

**Summary:**

This paper proposes HOD-GNN, a framework that enhances GNN expressiveness by computing high-order derivatives of a base MPNN with respect to node features and using them as structural encodings. Experiments on graph classification and regression benchmarks demonstrate competitive performance, with particular success on larger graphs where full-bag subgraph GNNs fail.

**Strengths:**

1. The paper introduces a novel approach to GNN expressivity. Unlike prior work that uses derivatives for analysis, this paper uses derivatives as input features to enhance expressiveness. The connection between Taylor expansion of marking and derivatives provides clear intuition.

2. The efficient derivative computation algorithm is a notable technical contribution. Exploiting graph sparsity through message-passing-like updates makes the approach practical, as demonstrated by favorable memory usage and successful scaling to Peptides datasets where subgraph GNNs fail.

3. The experimental evaluation is thorough for k=1 case, covering diverse benchmarks, multiple baseline families.

**Weaknesses:**

1. Despite the paper's title and emphasis on "k-HOD-GNN", all experiments use k=1. The core theoretical claim that `k-HOD-GNN matches k-OSAN expressivity for general k` is  unvalidated. No experiments test k=2 or higher, no comparison with k-OSAN for k>1, and no demonstration of distinguishing graph pairs that require k>1. This makes the paper's main contribution more claimed than demonstrated.

2. Theorem 4.2 only proves superiority over RWSE+MPNN for ReLU, not k-WL equivalence. The paper fails to adequately address this gap between theoretical claims and practical implementation. The k-WL distinguishability claim in the introduction is therefore misleading for the models actually tested.

**Questions:**

1. Can you provide experiments validating k-HOD-GNN for $k \ge 2$? Even a simple synthetic task distinguishing graph pairs requiring 2-node marking would substantially strengthen the main contribution. Without this, the paper's scope should be revised to focus on 1-HOD-GNN.

---

> ### Author Response · Authors · 2025-11-19
> **Response to reviewer nnEj**
>
> We thank the reviewer for their thoughtful and constructive feedback. We appreciate the positive remarks regarding our “novel approach”, the “clear intuition” behind the method, and the “thorough experimental evaluation” demonstrating “competitive performance, with particular success on larger graphs.” Below, we address the reviewer’s questions and concerns. We hope that you find our responses satisfactory, and that you will consider revising your score.
>
> - **Response to “Can you provide experiments validating k-HOD-GNN for $k \geq 2$? Even a simple synthetic task distinguishing graph pairs requiring 2-node marking would substantially strengthen the main contribution”** :
>
>   We thank you for raising this important point. We fully agree that extending the empirical study to $k > 1$ strengthens the paper. Following your suggestion, we conducted additional experiments evaluating both 1-HOD-GNN and 2-HOD-GNN on the family of regular graphs used in the BREC benchmark [4], which contains 140 pairs of regular graphs with 50 pairs that are 3-WL-distinguishable (but 2-WL-indistinguishable) and 90 pairs that are 3-WL-indistinguishable. The results, along with comparisons to subgraph-based GNNs, are presented in the Table below. The key observations are:
>
>   1. **2-HOD-GNN successfully distinguishes 34/90 of the 3-WL-indistinguishable pairs**, confirming the theoretical prediction the reviewer asked us to validate.
>
>   2. **1-HOD-GNN performs comparably to DS-GNN**, providing additional empirical support for Theorem 4.1 in the $k=1$ setting.
>
>   3.  Interestingly, the publicly available OSAN implementation, which follows a simplified instantiation of the full model described in the paper, does not distinguish any of the 3-WL-indistinguishable pairs on this benchmark, even though the theoretical results indicate that the full OSAN model is capable of doing so.
>
>   We appreciate your important suggestion; the extended experiments reinforce the theoretical claims and further clarify the capabilities of higher-order HOD-GNN variants. We will add it to the revised manuscript.
>
>   |Model|Regular Graphs Number | Regular Graphs Accuracy |
>   |-|-|-|
>   |3-WL| 50 | 35.7% |
>   |NGNN |48|34.3%|
>   |DE+NGNN| 50 |35.7%|
>   | DS-GNN | 48 | 34.3% |
>   | DSS-GNN | 48 | 34.3% |
>   | SUN | 50 | 35.7% |
>   | SSWL_P | 50 | 35.7% |
>   | GNN-AK | 50 | 35.7% |
>   | KP-GNN | 106 | 75.7% |
>   | I²-GNN | 100 | 71.4% |
>   |OSAN| 8 | 5.7% |
>   | 1-HOD GNN |47|33.5%|
>   | 2-HOD GNN |84| 60.0% |
>
> - **Response to “Theorem 4.2 only proves superiority over RWSE+MPNN for ReLU, not k-WL equivalence” and “gap between theoretical claims and practical implementation”:**
>
>   We thank the reviewer for raising this point and are happy to clarify the connection between our theoretical results and the practical models evaluated. Importantly, Theorem 4.1 with $k=1$ establishes that 1-HOD-GNN matches the distinguishing power of the subgraph-based WL refinement algorithm introduced in [3], thus offering stronger guarantees than those of Theorem 4.2. Since our empirical evaluations use the 1-HOD-GNN architecture, paired with either ReLU or **the analytic SiLU activation**, Theorem 4.1 applies directly to a non-trivial subset of our practical models. Consequently, our empirical instantiation is supported by the stronger expressivity guarantees of Theorem 4.1, and not only by the more ReLU-focused statement of Theorem 4.2.
>
>   Theorem 4.2 serves a different role: it shows that even with non-analytic activations like ReLU—whose higher-order derivatives vanish—the HOD framework still retains expressivity benefits. This complements, but does not replace, the stronger guarantees of Theorem 4.1, which apply whenever analytic activations are used. In fact, we present an ablation study in Table 6  in the appendix, confirming the gap in expressive power between SiLU and ReLU-based 1-HOD-GNNs, validating the theoretical gap between Theorems 4.1 and 4.2 . The confusion likely arose because we did not explicitly state that Theorem 4.1 also covers the $k = 1$ setting used in our experiments.  We believe our results are not misleading and will make our best effort to clarify this in the revision.
>
> We once again thank the reviewer for their careful reading and constructive feedback. We believe the additional experiments and clarifications provided here directly address the reviewer’s concerns. As the reviewer noted, adding such experiments would “substantially strengthen the main contribution”, and we believe the new results indeed achieve this. We would be grateful if the reviewer could consider raising their score in light of these improvements and clarifications.
>
>
> **References**:
>
> [1] Maron et al. “Provably Powerful Graph Networks”. NeurIPS 2019.
>
> [2] Qian at al.  “Ordered Subgraph Aggregation Networks”. NeurIPS 2022.
>
> [3] Bevilacqua et al. “Equivariant Subgraph Aggregation Networks”. ICLR 2022.
>
> [4] Wang et al. “An Empirical Study of Realized GNN Expressiveness” . ICML 2024.

---

> > ### Comment · Reviewer_nnEj · 2025-11-26
> >
> > I appreciate the reviewer for the additional experimental results and the clarification regarding Theorem 4.2, which address my concerns. I have updated my score accordingly.

---

### Official Review · Reviewer_986R · 2025-10-26

**Soundness:** 4
**Presentation:** 4
**Contribution:** 3
**Rating:** 6
**Confidence:** 4

**Summary:**

This paper introduces HOD-GNN, an architecture that enhances GNN expressivity by computing derivatives of a base MPNN with respect to input node features and using these derivatives as additional structural information to input to the second GNN.  The paper proves that k-HOD-GNN can approximate k-OSAN subgraph GNNs, which is strictly more expressive than MPNNs with random walk structural encodings, and achieves favorable computational complexity for sparse graphs with shallow base MPNNs by exploiting an efficient message-passing-like derivative computation algorithm. Empirically, HOD-GNN consistently ranks in the top two tiers across seven benchmarks.

**Strengths:**

1. The paper presents a fresh perspective on enhancing GNN expressivity through derivatives. The intuition is clearly explained through the connection to marking-based GNNs and Taylor expansion. I think the presented approach is conceptually elegant.

2. The theoretical analysis provides a complete expressivity characterization by proving k-HOD-GNN approximates k-OSAN subgraph GNNs while offering constructive separation examples that precisely identify when the method is efficient.

3. The three-component pipeline from base MPNN, intermediate derivative computation, and the downstream GNN is clear and easy to follow.

**Weaknesses:**

1. The claimed "inductive bias toward derivative-aware representations" is mentioned but never explained.

2. The triangle counting example in the Motivation section is thin as a driver of the whole method. This motivating toy showing first-order derivatives recover triangle counts uses a very special linear stack, i.e., identity activations, and does not show whether derivatives remain informative for typical non-linear, normalized, or regularized MPNN on real tasks. Thus, it’s illustrative but not compelling evidence that derivatives are broadly the right signal. Also, in the theoretical analysis, the separation beyond k-WL depends on analytic activations that  iss fine for softplus/tanh, but not ReLU.

3. Flattening $D^{out}$ could be large in practice.

4. For $U^{node}$, a 2-IGN is powerful but can be heavy. The paper claims scalability via sparsity, but constants can bite (derivative channels $\times$ feature dims $\times$ orders).

5. I didn’t see normalization for $D^{out}$ or $D^{(T)}$. Some stabilization may be necessary because  derivatives can vary wildly with feature scaling.

6. I’m confused about the rationale for computing two derivative tensors that target different functions. Specifically, $D^{(T)}$ computes derivatives of the final node embeddings, while $D^{out}$ computes that based on the graph-level output. Why is it necessary to encode both, rather than using only node-level derivatives with an invariant head, or using only graph-level derivatives if the target is graph classification?

**Questions:**

See weaknesses.

---

> ### Author Response · Authors · 2025-11-19
> **Response to reviewer 986R (part 1/2)**
>
> We thank the reviewer for their thoughtful and positive assessment of our work. We are especially grateful for the comments highlighting that the paper presents “a fresh perspective on enhancing GNN expressivity through derivatives”, that the core intuition is “clearly explained” and “conceptually elegant,” and that the overall exposition is “clear and easy to follow.” We also appreciate the reviewer’s recognition that our theoretical analysis provides a “complete expressivity characterization” and that the empirical results show HOD-GNN “consistently ranks in the top two tiers across seven benchmarks.” We are encouraged by this feedback and now address the reviewer’s questions and suggestions in detail below.
> - **Response to “inductive bias toward derivative-aware representations is mentioned but never explained”**:
>
>   We thank the reviewer for the opportunity to clarify this point. As discussed in the introduction and related works sections, derivatives of GNNs naturally arise in several lines of theoretical analysis (e.g. in studies of oversmoothing, oversquashing, and interpretability).  When we refer to an “inductive bias toward derivative-aware representations”, we mean that if derivatives consistently appear as useful and informative quantities in understanding GNN behavior, it is reasonable to bias the model toward utilizing this information directly. The idea is intentionally intuitive: representations that align with quantities highlighted by theoretical analysis may be especially informative for the model. This motivating intuition will be explained more clearly in the next revision.
>
>
>
> - **Response to “triangle counting example in the Motivation section is thin as a driver of the whole method”**:
>
>   We appreciate your insightful perspective on the triangle-counting example. We agree that this example is primarily illustrative rather than a complete justification that derivatives are broadly the right signal. This was intentional: in the Motivation section, our goal was to start with a simple, concrete scenario that helps build intuition for why derivative information can enhance expressivity. As a “warm-up” motivating example,it offers a simple and concrete problem which our approach solves , preparing  the reader for the more complex arguments developed in Section 4 of the paper. We also note that the motivation section discusses additional motivating arguments. Immediately after the triangle example, we provide a more general intuition using the Taylor expansion, to connect derivatives to marking-based GNNs, an explanation the reviewer described as “clearly explained” and “conceptually elegant.”
>
>   We strongly believe that in light of our theoretical results (Section 4) and empirical findings (Section 5), derivatives are a good input signal for graph learning. In particular, we demonstrate the benefit offered by derivatives throughout the paper, as follows:
>   - **Theoretical evidence:**
>       - Theorem 4.1 establishes stronger expressivity under analytic activations, as the reviewer notes.
>
>       - Theorem 4.2 shows that even with the non-analytic ReLU, HOD-GNN still yields strong expressivity improvements.
>
>
>   - **Empirical evidence:**
>       - Across seven benchmarks, HOD-GNN performs competitively with a range of expressive baselines.
>
>
>       - Table 6 in the Appendix provides a synthetic substructure-counting experiment showing that derivative features improve expressivity for both analytic and ReLU activations.
>       - Table 7 in the Appendix provides an ablation demonstrating that higher-order derivatives improve a model’s substructure-counting ability.
>   Together, these theoretical and empirical results give more robust evidence that derivatives provide a useful and general signal, beyond the initial illustrative triangle-counting example. We will clarify this narrative in the next revision. Thank you.

---

> > ### Author Response · Authors · 2025-11-19
> > **Response to reviewer 986R (part 2/2)**
> >
> > - **Response to “Flattening D_out  could be large in practice”**:
> >
> >   We appreciate the opportunity to discuss this. In practice, we find that the hidden dimensions and derivative orders used by HOD-GNN can be kept relatively small (typically derivative order ≤3 and hidden dimensions in the range of 8–32). This keeps the size of $D^{\text{out}}$  modest, making the flattening step inexpensive. To quantify this, we measured the actual runtime cost of the flattening operation during training on the MOLHIV dataset from Table 1. The flattening step took 1.86 milliseconds per batch, corresponding to approximately 14.5 microseconds per graph, which is negligible relative to the overall computation.
> >
> > - **Response to “For D_node, a 2-IGN is powerful but can be heavy”**:
> >
> >   We thank the reviewer for raising this important point. As noted, a standard 2-IGN can indeed be computationally heavy, and directly applying it could undermine the scalability benefits of HOD-GNN. To address this, it is important to note that we do **not** use a full 2-IGN in our implementation. Instead (as mentioned in Footnote 4 on page 5) we employ a **modified variant of 2-IGN** that is specifically designed to **preserve the sparsity structure of the derivative tensors, while preserving the expressiveness guarantees of Theorem 4.1**. The full details of this sparse-friendly k-IGN variant are provided in Appendix C, and results on its effect on computational complexity and expressivity are provided in Appendix E. Together, these results demonstrate that using the sparse 2-IGN variant greatly reduces the computational overhead and makes the approach compatible with the efficient derivative computation pipeline. We will highlight this design choice in the next revision. Thank you for the important comment.
> >
> > -  **On normalizing the derivative features and the stability of HOD-GNN:**
> >   We thank the reviewer for highlighting this important point. As suggested, normalization is indeed crucial when working with derivative features. Hence, in our paper, we have dedicated thought and space for this matter. In particular, the normalization strategy we employ is described in Equation (73) in Appendix F.1: we divide the node-wise derivative tensor computed at layer $t$ (i.e. $D^{(t)}$) of the base MPNN by $t!$. This approach is inspired by the normalization used for centralized positional encodings in [1] and helps maintain numerical stability. We analyze the stability of HOD-GNN in Appendix G.3. There, we show that for derivative orders $\alpha = 1, \dots , 4$, the training curves remain stable and the norms of the derivative tensors stay well-behaved relative to the final node-feature norms produced by the base MPNN. We will make clearer references to both the normalization method and the stability analysis in the revised manuscript.
> >
> > - **Response to "I'm confused about the rationale for computing two derivative tensors that target different functions”**:
> >
> >   In the full description of HOD-GNN, we distinguish between the output-level derivative tensor $D^{\text{out}}$ and the node-level derivative tensor $D^{\text{node}}$. The primary reason for presenting both is generality: depending on the downstream task, derivatives of either the final node embeddings or the graph-level output may, in principle, provide useful structural information. Since both forms of sensitivity can be informative, we chose to give the most general formulation of the framework. That said, for practical implementation and simplicity, we rely only on the node-level derivative tensor, as detailed in Appendix F.1. Empirically, this choice performed very well across all datasets we evaluated, and we did not observe clear benefits from incorporating $D^{\text{out}}$ in the tasks considered. We will clarify this in the revised manuscript.
> > We thank the reviewer again for the valuable feedback. The clarifications above enhance the clarity and scope of our work, and we believe they address the concerns raised.  We will integrate all of these improvements into the revised manuscript.  If the clarifications and additions in our rebuttal adequately address your concerns, we would be grateful if you would consider adjusting your score accordingly.
> >
> > **References**:
> >
> > [1] Southern et al. “Balancing Efficiency and Expressiveness: Subgraph GNNs with Walk-Based Centrality‏”. ICML 2025.

---

> > > ### Comment · Reviewer_986R · 2025-11-25
> > >
> > > Thanks for the author response. The response is detailed and has largely addressed my concerns. I wish to keep my positive score.

---

### Official Review · Reviewer_S9Dt · 2025-10-29

**Soundness:** 3
**Presentation:** 3
**Contribution:** 3
**Rating:** 6
**Confidence:** 4

**Summary:**

The authors consider the expressive power of GNN from a derivative perspective. Combining high-order derivatives, the authors enhance the expressivity of GNN.

**Strengths:**

In this paper, the authors consider the expressive power for the derivatives of GNN, an innovative and interesting perspective to analyze expressive ability. By leveraging to high-order node derivatives, the authors enhance expressive power of GNN. The analysis seems reasonable and the conclusion seems reliable.

**Weaknesses:**

While this is an interesting work, this finding is not surprising. Normally, the node feature vectors can be viewed as the zero-order information, i.e., the function value $h_v$. High-order derivatives can be seen as the high-order information for the function value. Combining much more high-order information can distinguish many graphs doesn't seem surprising to me. Instead, I am more interested in uncovering deeper insights from the derivative perspective, as such insights could provide a more fundamental understanding of the expressive power of GNNs.

**Questions:**

1.I am interested in a more detailed characterization of the relationship between the k-WL test and GNN derivatives. For example, on a high level, can the k-WL test be equivalently expressed using derivatives of order less than k? Such a connection could directly elucidate the source of enhanced expressive power.

2.I would like to clarify the impact of different choices for $\alpha$ on expression ability in k-HOD-GNN? Theorem 4.1 only states existence, but it remains unclear: whether any possible choice of $\alpha$ in k-HOD-GNN has stronger expression power than the k-WL test, when k and m are fixed. Then, it would be valuable to clarify the conditions on $\alpha$ that guarantee stronger expression power. In addition, a systematic method for selecting the optimal $\alpha$ is also an important theoretical justification.

---

> ### Author Response · Authors · 2025-11-19
> **Response to reviewer S9Dt (part (1/2)**
>
> We thank the reviewer for their positive review, stating our work is innovative,  interesting, and reliable. Below, we address the reviewer’s questions.
>
> - **Response to “can the k-WL test be equivalently expressed using derivatives of order less than k?”**:
>
>   We thank the reviewer for raising this insightful question. As shown in Theorem 4.1, $k$-HOD-GNN can distinguish graph pairs that are indistinguishable by the $k$-Folklore WL (FWL) test (and therefore also by $(k+1)$-oblivious WL). Motivated by the reviewer’s question, we expanded our analysis and derived three additional propositions (detailed below) that further clarify how the order of derivatives relates to the expressive power of HOD-GNN. These propositions and their formal proofs will be added to the revised manuscript promptly.
>   Together, these propositions help clarify how derivative order relates to WL-type expressivity. They show that derivatives do not simply act as generic higher-order information, but can provide efficient access to higher-order distinctions, sometimes allowing one (or even two) levels of WL tuple complexity to be recovered from lower-order derivative tensors.
>   -  **Proposition 1**: A 1-HOD-GNN that uses only the output-level derivative tensor $D^{\text{out}}$ can already distinguish graphs that 1-FWL cannot, and is more expressive than MPNN+RWSE.
>
>        Intuitively, this result follows naturally from Equation (3) and the extended motivation discussion in Appendix B. Here, $D^{\text{out}}$ is only a first-order tensor (indexed by a single node and feature channels). Since distinguishing 1-FWL-indistinguishable graphs using 2-FWL would normally require operating on 2-tuples of nodes, this result shows that derivatives allow us to “save” one level of tuple complexity.
>   - **Proposition 2**: The expressive power of k-HOD-GNN is upper-bounded by the $(k+1)$-FWL test.
>
>       The proof of this proposition proceeds by carefully simulating the computations of k-HOD-GNN with a $(k+2)$-IGN, which is known to have the exact same expressive power as the $(k+1)$-FWL test [2]. This result shows that the one-level savings described above is maximal: derivatives cannot collapse multiple levels of the WL hierarchy.
>   - **Proposition 3**: When first-order derivatives with respect to edge features are incorporated (a natural extension of 1-HOD-GNN that will be formally defined in the revised manuscript), a 1-HOD-GNN can distinguish certain 2-FWL indistinguishable graphs.
>
>       The intuition mirrors the reasoning in Equation (3) to simulate DS-GNN with edge-deletion operations, which is known to separate pairs of graphs distinguishable by the $k$-FWL test only for $k\geq3$. Under the natural sparsity assumption $|E| = O(n)$, this means that by only processing an $O(n)$-sized tensor, the model achieves distinguishing power that normally requires operating on 3-tuples of nodes, saving two levels of tuple complexity in this setting. The key difference from the previous proposition is the use of edge-wise derivatives, analogous to the distinction in subgraph GNNs, where node-based variants are bounded by 2-FWL expressivity [3] , while those that incorporate edge-deletion policies are not [4].
>
> - **Response to “High-order derivatives can be seen as the high-order information for the function value. Combining much more high-order information can distinguish many graphs doesn't seem surprising to me”**:
>
>   While we agree that interpreting derivatives as higher-order information is a natural intuition, we believe that our specific formulation of modeling higher-order information through derivative tensors is indeed novel and, as the reviewer noted, innovative. Beyond the strong empirical performance demonstrated in Section 5, the main theoretical advantage of HOD-GNN over existing higher-order methods lies in its ability to achieve significantly higher computational efficiency while maintaining comparable expressive power.  In particular, the expressive power of k-HOD-GNN matches that of k-OSAN [1], and both are incomparable to the $(k+1)$-FWL test (that is, each can distinguish certain graph pairs that the other cannot). At the same time, HOD-GNN leverages the sparsity of natural graphs, yielding substantially improved asymptotic computational complexity. The table below summarizes these differences (n is the number of nodes, d is the maximal degree and T is the number of layers used):
>
>   |Method|Space Complexity|Runtime Complexity|
>   |-|-|-|
>   |$(k+1)$-FWL|$O(n^{k+1})$|$O(n^{k+2})$|
>   |k-OSAN|$O(n^{k+1})$|$O(d \cdot n^{k+1})$|
>   | HOD-GNN|$O( n \cdot \min (n^k , d^k\cdot T) )$|$O( d \cdot n \cdot \min (n^k , d^k\cdot T) )$|
>
>   This computational advantage is also reflected empirically, as 1-HOD-GNN can train on larger graph benchmarks such as the Peptides datasets (see Table 2), whereas subgraph-based GNNs typically fail due to out-of-memory errors.

---

> > ### Author Response · Authors · 2025-11-19
> > **Response to reviewer S9Dt (part (2/2)**
> >
> > - **Response to “it would be valuable to clarify the conditions on $\alpha$ that guarantee stronger expression power. In addition, a systematic method for selecting the optimal  $\alpha$  is also an important theoretical justification”**:
> >
> >   We thank the reviewer for raising this subtle and interesting question. In our current formulation, $\alpha$ is treated as a tunable hyperparameter. Just as the expressive power of standard GNNs is constrained by limiting depth or hidden dimension [5], the expressive power of k-HOD-GNN can be restricted when $\alpha$ is bounded. Larger $\alpha$ values allow the model to access higher-order derivative information and thus potentially distinguish more complex structures albeit at increased computational cost. For this reason, we aim to use the smallest $\alpha$ that already yields strong performance.
> >
> >   To address the reviewer’s request for “conditions on $\alpha$ that guarantee stronger expressive power” , we would like to highlight that **Theorem 4.2 provides such a guarantee**. As the theorem states, even in the simplest setting of $\alpha = 1$ with ReLU activation, k-HOD-GNN is provably more expressive than MPNN + RWSE, which is an architecture already known to exceed the power of the 1-WL test [6]. This theoretical insight is further supported by the empirical results in Table 6 in the Appendix, where we evaluate substructure-counting ability for $\alpha =1$ + ReLU. Notably, even this minimal configuration of HOD-GNN achieves strong substructure‐counting performance: it **substantially outperforms baselines that rely on RWSE** (GPS+RWSE, MPNN+RWSE), as well as sampling-based subgraph GNNs like HyMN.  This provides concrete empirical evidence that derivative information is already highly informative even at $\alpha = 1$, supporting the theoretical guarantees of Theorem 4.2.
> >
> >   To address the reviewer’s request for a “method for selecting the optimal  alpha”, we highlight the ablation study reported in Table 7 in the Appendix, where we  evaluate substructure-counting performance for $\alpha = 0,1,...,4$. **The results  show that increasing $\alpha$ consistently strengthens expressivity**, until the performance gains eventually plateau. Practically, this suggests that a small sweep over $\alpha \in$ {2,3,4} is typically sufficient.
> >
> >   Finally, motivated by your question, we added a new proposition showing that using higher values of $\alpha$ can help get more useful information from the same base MPNN, without adding parameters to the overall model. Specifically, we show:
> >
> >   - **Proposition**: For any $\alpha \in N$, there exist choices of the base MPNN $M$ such that a $k$-HOD-GNN using $M$ with order $\alpha$ cannot distinguish certain graph pairs, while the same architecture using order $\alpha+1$ can.
> >
> >    We will include this new proposition, along with clearer references to our empirical ablation, in the revised manuscript. We believe this helps strengthen both the theoretical and empirical discussion of $\alpha$’s role.
> > We hope our responses fully address the reviewer’s questions, and we sincerely appreciate the insightful feedback that helped us strengthen the paper.  Thank you.  If our clarifications are helpful, we would sincerely appreciate your consideration of a higher score.
> >
> > **References**:
> >
> > [1] Qian at al.  “Ordered Subgraph Aggregation Networks”. NeurIPS 2022.
> >
> > [2] Maron et al. “Provably Powerful Graph Networks”. NeurIPS 2019.
> >
> > [3]  Frasca et al. “Understanding and Extending Subgraph GNNs by Rethinking Their Symmetries”.
> >
> > [4] Bevilacqua et al. “Equivariant Subgraph Aggregation Networks”. ICLR 2022.
> >
> > [5] Morris et al. “Weisfeiler and Leman Go Neural: Higher-order Graph Neural Networks”.  AAAI 2019.
> >
> > [6] Dwivedi et al. “Graph neural networks with learnable structural and positional representations”. ICLR 2022.

---

> > > ### Comment · Reviewer_S9Dt · 2025-11-26
> > >
> > > Thank you for your rebuttal and detailed responses. My concerns have been addressed. I will maintain my positive score.

---

### Official Review · Reviewer_WWrV · 2025-10-29

**Soundness:** 3
**Presentation:** 3
**Contribution:** 3
**Rating:** 6
**Confidence:** 4

**Summary:**

This paper introduces new class of Graph Neural Networks that aims to enhance expressivity by leveraging derivatives of MPNNs with respect to node features. The authors argue that derivatives naturally encode structural information and can thus extend the expressive power of base GNNs, drawing parallels with Weisfeiler–Lehman hierarchies, subgraph GNNs, and positional encodings. The paper includes theoretical expressivity proofs, an efficient message-passing algorithm for derivative computation, and experiments.

**Strengths:**

1) The paper is well written and easy to follow. The idea is clear, simple but not trivial, and the conceptual move of translating the use of derivatives, which are widely explored in other contexts, into the domain of expressivity is interesting.

2) It is particularly interesting that k-HOD-GNN can approximate any k-OSAN model to arbitrary precision while remaining computationally more efficient. This means HOD-GNN advances the field in a desirable direction, i.e., toward architectures that are highly expressive yet computationally tractable.

3) Generally, the theoretical exposition is coherent and convincing, and it is always clear what is being done and why. Overall, the contribution feels clean: the narrative is consistent, and the motivation for using derivatives is well justified.

**Weaknesses:**

**Minor issues:**

1) There are a few small notational inconsistencies:

-Line 150: $\mathbf{X}$ should be bold, and h should carry a subscript (v for node-level, out for global as the authors define), since it’s unclear otherwise which representation is being referred to.

-Line 158: when mentioning the final node representation $h_v$, it would be more consistent to add the superscript ($T$) as done earlier.

-Line 125: you use $V^k$, which is understandable but never formally defined. Similarly, what exactly a “model” $\mathcal{M}$ is could be defined more formally, though the meaning is still clear.

2) The overview paragraph (lines 191–199) could be made clearer. In particular, point (4) says that $\mathcal{T}$ is applied to “derivative-informed features”, but since $\mathcal{T}$ is itself a GNN, the phrasing might mislead the reader to think that it acts only on those features, rather than on the full graph. Moreover, earlier in the paragraph $\mathcal{T}$ is only called “downstream network,” which increases potential confusion. For such a crucial overview section, absolute clarity would help.

**Conceptual concerns:**

I think the paper could engage more critically with the current stage of the field. The literature on GNN expressivity went through a strong wave of works proposing models with expressive power going beyond 1-WL. However, in recent times the community has started to question whether such additional expressivity is truly useful in practice [1]—especially on feature-rich datasets, where node attributes already break most of the symmetries that 1-WL cannot distinguish. One could argue that even if 1-WL failures are rare in real-world datasets, developing models that go beyond the WL limit is still valuable, as it guarantees the ability to distinguish as many input as possible, regardless of whether such cases are frequent or not. However, several recent works have shown that higher expressivity often comes at the cost of worse generalization [2,3,4]. In this sense, proposing a model that only aims to surpass the WL boundary, without discussing why this added expressivity is beneficial or how it impacts generalization, feels a bit outdated relative to how the field has evolved. It would strengthen the paper to address questions like: 1) Do you have any hints about the generalization behavior of your model? 2) Why might HOD-GNN preserve generalization compared to other expressive models? 3) Can you provide an intuition for how derivative-based features may induce better inductive bias? A discussion about this would make the contribution feel more in tune with the current research conversation.


**Experimental completeness:**

If results for certain baselines were not reported in prior papers, they were not rerun here. This leads to some missing entries, particularly for two datasets. It’s not necessarily a major issue, but it would be useful to explain the reason for the omission explicitly. For instance, were those models prohibitively expensive to run?


[1] Jogl, Fabian, Pascal Welke, and Thomas Gärtner. "Is Expressivity Essential for the Predictive Performance of Graph Neural Networks?." NeurIPS 2024 Workshop on Scientific Methods for Understanding Deep Learning. 2024.

[2] Franks, Billy Joe, et al. "Weisfeiler-Leman at the margin: When more expressivity matters." Forty-first International Conference on Machine Learning.

[3] Maskey, Sohir, et al. "Graph Representational Learning: When Does More Expressivity Hurt Generalization?." arXiv preprint arXiv:2505.11298 (2025).

[4] Carrasco, Martin, et al. "Rademacher Meets Colors: More Expressivity, but at What Cost?." arXiv preprint arXiv:2510.10101 (2025).

**Questions:**

See weaknesses.

---

> ### Author Response · Authors · 2025-11-19
> **Response to reviewer  WWrV (part (1/2)**
>
> We thank the reviewer for their positive assessment, noting that the core ideas are “interesting,” “simple but not trivial,” and “advance the field in a desirable direction,” and for highlighting that “the theoretical exposition is coherent and convincing” and that “the contribution feels clean.” We now address your questions and concerns.
> - **Response to “Do you have any hints about the generalization behavior of your model?” and “Why might HOD-GNN preserve generalization compared to other expressive models?”**:
>
>   We thank the reviewer for raising this thoughtful point, and appreciate the opportunity to elaborate on how HOD-GNN behaves in terms of generalization and inductive bias.
>
>   Empirically, HOD-GNN’s test performance is consistently competitive with a wide range of baselines, indicating **strong generalization behavior**. These benchmarks rely on expertly curated, chemically meaningful splits (e.g., scaffold splits), which provide a meaningful and challenging test of a model’s ability to generalize.
>   Next, inspired by your question, and to better understand the generalization behavior of HOD-GNN, we provide train–test performance and gaps for the OGB datasets MOLHIV and MOLTOX21. The performance gaps observed in HOD-GNN are moderate, and smaller (i.e., better) than those observed in both less expressive architectures such as GIN and GCN, models of similar expressivity such as DSS-GNN with node deletion, as well as more expressive variants such as DSS-GNN with edge deletion, indicating good generalization behavior overall. Baseline values are taken from [1].
>
>   | Method|MOLHIV Train AUC|MOLHIV Test AUC|Gap-MOLHIV|MOLTOX21 Train AUC| MOLTOX21 Test AUC|Gap-MOLTOX21|
>   |-|-|-|-|-|-|-|
>   |GCN| 88.65±2.19 | 76.06±0.97 | 12.59 | 92.06±1.81 | 75.29±0.69 | 16.77 |
>   |GIN| 88.64±2.54 | 75.58±1.40 | 13.06 | 93.06±0.88 | 74.91±0.51 | 18.15 |
>   |DSS-GNN (ED)| 91.71±3.50 | 76.43±2.12 | 15.28 | 92.38±1.57 | 75.12±0.50 | 17.26 |
>   |DSS-GNN (ND)| 89.70±3.20 | 76.19±0.96 | 13.51 | 91.23±2.15 | 75.34±1.21 | 15.89 |
>   |HOD-GNN| 88.15 ± 0.51| 80.86±0.52 |  7.29 | 93.73±1.09 |77.99±0.71|15.74 |
>
>
>
>   Regarding the question  “Why might HOD-GNN preserve generalization compared to other expressive models?” We offer two intuitive hypotheses that we believe align well with the recent literature on expressivity and generalization:
>
>
>   - **Effective use of small hidden dimensions**:
>
>       As the reviewer mentions, recent work [3] shows that GNNs tend to generalize best when they have “just the right capacity.” This refers both to the expressive power of the architecture and to its parameter scale: generalization bounds in Theorem 4.1 in the aforementioned [3], highlight that having too many parameters, or an overly wide hidden dimension, can also harm generalization. Interestingly, across our experiments, the base MPNN inside HOD-GNN achieves its best performance with small hidden dimensions (8–32). This results in models with relatively few parameters, which we believe contributes to their strong generalization behavior.
>
>      We hypothesize that HOD-GNN performs well with such compact architectures because higher-order derivatives allow it to extract richer information from a small base representation, effectively enhancing representational capacity without increasing the parameter count.
>
>   -  **Principled initialization via RWSE**
>
>       As discussed after Theorem 4.2 in our paper, the proof of the theorem is fully constructive and provides an explicit initialization for the derivative-informed features that exactly matches Random Walk Structural Encodings (RWSE). RWSE is a widely used technique for enhancing expressivity and introducing meaningful inductive bias, aligning well with the “just expressive enough” principle highlighted in [3].
>       Thus, an alternative perspective on HOD-GNN, beyond its placement within the WL hierarchy as presented in the paper, is to view it as a task-adaptive refinement of RWSE:
>       - At the beginning of training, the derivative-informed features remain close to RWSE, maintaining well known baseline behavior.
>       - When the task benefits from additional capacity, the model can naturally move beyond RWSE and leverage higher-order derivative information.
>
>
>       This offers a natural inductive bias: the model begins with a well-understood, robust structural encoding and becomes more expressive only when the data requires it.
>   We hope this discussion provides helpful intuition into the generalization behavior of HOD-GNN, and we will incorporate an expanded version of it into the revised manuscript. We view further investigation of HOD-GNN’s generalization properties as a promising direction for future work.

---

> > ### Author Response · Authors · 2025-11-19
> > **Response to reviewer WWrV (part (2/2)**
> >
> > - **Response to “it would be useful to explain the reason for the omission explicitly”**:
> >
> >   We thank the reviewer for raising this point. As noted, the missing entries in Tables 1–2 arise because, to the best of our knowledge, these specific baseline–dataset combinations have not been reported in any prior published work. Because our goal is to provide as broad and fair of a comparison as possible, we include all baselines that offer substantial reported results, even if some entries are unavailable. This is a common practice in graph ML benchmarking [2,4] and is explicitly mentioned in the description of Table 2. We will make it more visible in the revised version for enhanced clarity. Thank you.
> >
> > -  **Response to Minor issues**:
> >
> >     We appreciate the reviewer for these careful and constructive suggestions regarding notation and presentation. We will incorporate all of these refinements in the next revision, as we believe that these will enhance readability. Thank you.
> > We are grateful for  your careful reading and insightful comments. We believe the points addressed above enhance the clarity and impact of the work, and we will include them in the revised manuscript. Thank you.  If our responses help address the issues you raised, we would sincerely appreciate your consideration of a higher score.
> >
> > **References**:
> >
> >
> > [1] Bevilacqua et al. “Equivariant Subgraph Aggregation Networks”. ICLR 2022.
> >
> > [2] Bevilacqua et al. “Efficient Subgraph GNNs by Learning Effective Selection Policies”. ICLR 2024.
> >
> > [3] Maskey, Sohir, et al. "Graph Representational Learning: When Does More Expressivity Hurt Generalization?." arXiv preprint arXiv:2505.11298 (2025).
> >
> > [4] Ma et al. “Graph Inductive Biases in Transformers without Message Passing”. ICML 2023.

---

> > > ### Comment · Reviewer_WWrV · 2025-11-24
> > >
> > > I thank the authors for the detailed responses which have addressed most of my concerns. I have one remaining concern regarding the completeness and comparability of the experimental results.
> > >
> > > Since the baseline results were taken from prior papers rather than rerun in a unified setup, could the authors clarify whether the experimental settings across these sources are fully aligned? For instance, do the reported means and standard deviations come from the same number of runs, and were the train/validation/test splits and stopping criteria consistent across works?

---

> > > > ### Author Response · Authors · 2025-11-25
> > > > **Response to final remaining concern**
> > > >
> > > > We thank the reviewer for noting that most concerns have been addressed, and we are happy to clarify the remaining point regarding comparability of experimental setups.
> > > >
> > > > We agree that transparency and fairness of reported baseline results are important, which is why Tables 1–2 explicitly indicate that these results come from prior papers. Across all experiments, our setup exactly follows that of [1-3]. As is common in GNN benchmarking (see, e.g., [1–7], spanning both highly cited and more recent works), we follow the standard practice of reporting baseline results as originally published when they rely on widely used, standardized protocols. Across the benchmarks used in our paper, these protocols are aligned, making the reported results a natural and fair basis of comparison—both among the baselines and with our own method. As this is an important topic, we will better clarify this in the revised manuscript.
> > > >
> > > > With best regards,
> > > >
> > > > Authors.
> > > >
> > > > References:
> > > >
> > > >
> > > >
> > > > [1] Rampasek at al. “Recipe for a General, Powerful, Scalable Graph Transformer”. NeurIPS 2022.
> > > >
> > > > [2]  Southern et al. “Balancing Efficiency and Expressiveness: Subgraph GNNs with Walk-Based Centrality‏”. ICML 2025.
> > > >
> > > > [3] Tonshoff et al. “Where did the gap go? reassessing the long-range graph benchmark”. LOG 2023.
> > > >
> > > >
> > > > [4] Ma et al. “Graph Inductive Biases in Transformers without Message Passing”. ICML 2023.
> > > >
> > > > [5] Bao et al. “Homomorphism Counts as Structural Encodings for Graph Learning”. ICLR 2025.
> > > >
> > > > [6] Lim et al. “Sign and Basis Invariant Networks for Spectral Graph Representation Learning”. ICLR 2023
> > > >
> > > > [7]  Bevilacqua et al. “Equivariant Subgraph Aggregation Networks”. ICLR 2022.

---

### Author Response · Authors · 2025-11-19
**General comment**

We thank all the reviewers for their positive and constructive feedback. We underscore 4 key contributions of our paper mentioned by the reviewers:




1. **Strong theoretical contribution:** Reviewers consistently highlighted the strength of our theoretical work, noting that the “theoretical exposition is coherent and convincing,” that our “efficient derivative computation algorithm is a notable technical contribution,” and that our analysis provides a “complete expressivity characterization” along with “constructive separation examples that precisely identify when the method is efficient” (**nnEj**, **WWrV**, **986R**).




2. **Novel and valuable perspective:** Reviewers described our approach as “novel”, “innovative,” “interesting,” and offering a “fresh perspective on GNN expressivity”, further noting that it "advances the field in a desirable direction" (**nnEj**, **S9Dt**, **WWrV**).




3. **Thorough and reliable experiments:** Reviewers noted that our experimental section is “thorough” and “reliable,” demonstrating “competitive performance” on graph learning benchmarks, with “particular success on larger graphs” (**nnEj**, **S9Dt**, **986R**).




4. **Clarity and quality of exposition:** Several reviewers highlighted that the paper is “clear”, “intuitive”, “well-written”, and “easy to follow”, and that the contribution “feels clean” (**nnEj**, **WWrV**, **986R**).


During the rebuttal period, we added several strong improvements to our paper:


1. In response to the reviewers’ inquiries regarding experimental validation of k-HOD-GNN for $k > 1$, we conducted new synthetic experiments demonstrating that **2-HOD-GNN can separate 34 pairs of 3-WL–indistinguishable graphs**, thereby directly validating our theoretical results (as requested).




2. In response to questions about the generalization ability of HOD-GNN, we compared the train–test gap of HOD-GNN on MOLHIV and MOLTOX21 to baselines that are both more expressive and less expressive. The results show that **HOD-GNN exhibits stronger generalization** than these baselines. We additionally include an intuition explaining why we believe HOD-GNN generalizes well.




3. Finally, to address inquiries about the exact expressivity gains obtained from using derivative signals, we provide **several complementary theoretical results** that clarify these effects. First, we show that even a 1-HOD-GNN with only output level derivatives can separate graphs that 1-FWL cannot, allowing us to “save” one level of tuple complexity. Second, $k$-HOD-GNN is bounded by $(k+1)$-FWL, showing that this one-level gain is maximal when using node/output level derivatives. Finally, adding edge-wise derivatives lets 1-HOD-GNN distinguish some 2-FWL-indistinguishable graphs, giving a two-level gain under standard sparsity assumptions.

---

### Author Response · Authors · 2025-11-21
**Revised manuscript upload**

Dear reviewers, we thank you for your helpful comments and suggestions, and have updated the manuscript accordingly. All changes have been marked in blue for ease of reference. Please note that the inclusion of new material required renumbering several items in the paper. In particular:
- Theorem 4.2 is now Theorem 4.3


- Propositions 4.3 and 4.4 are now Propositions 4.4 and 4.5


- Table 7 in the appendix is now Table 8

We believe these revisions address your feedback and improve the clarity and completeness of the work.

---

### Author Response · Authors · 2025-12-02
**Summary message for re-assigned AC**

To assist the newly assigned AC, we summarise below the reviews and subsequent rebuttal discussion. We are grateful to the original AC, and all reviewers for their constructive engagement. During the rebuttal, we addressed all comments in detail and ran several additional experiments, further strengthening the paper.

**General assessment**:

The initial scores were 6 (WWrV), 6 (S9Dt), 6 (986R), and 4 (nnEj). All reviewers highlighted the strengths of our work and raised thoughtful questions,  which we addressed with detailed responses and new experiments where appropriate. After the rebuttal, reviewer nnEj increased their score to 6, yielding final scores of:
 - 6 (WWrV)
- 6 (S9Dt)
- 6 (986R)
- 6 (nnEj)

Following the score increase, the paper had an **overall positive score with high confidence, reflecting a clear consensus among the reviewers**. Several reviewers also noted that their remaining concerns had been fully resolved.

**Strengths**

Reviewers positively highlighted:

1. **Strong theoretical contribution:** Reviewers consistently highlighted the strength of our theoretical work, noting that the “theoretical exposition is coherent and convincing” (WWrV), that our “efficient derivative computation algorithm is a notable technical contribution” (nnEj), and that our analysis provides a “complete expressivity characterization” (986R) along with “constructive separation examples that precisely identify when the method is efficient” (986R).

2. **Novel and valuable perspective:** Our approach was described as “novel” (nnEj), “innovative and interesting” (S9Dt),  and offering a “fresh perspective on GNN expressivity” (986R), further noting that it "advances the field in a desirable direction" (WWrV).

3. **Thorough and reliable experiments:** The experiments are described as “thorough” (nnEj) and “reliable” (S9Dt), demonstrating “competitive performance” (nnEj) on graph learning benchmarks, with “particular success on larger graphs” (nnEj).

4. **Clarity and quality of exposition:** Reviewers praised the paper as “clear” (nnEj), “intuitive” (986R), “well-written and easy to follow” (WWrV), and that the contribution “feels clean” (WWrV).

**Key Points Raised in Review and how we addressed them**:

- **Experimental validation of k-HOD-GNN for $k > 1$**:  Reviewer nnEj requested empirical validation of the theoretical results in Theorem 4.1 regarding the expressive power of k-HOD-GNN for $k>1$. In response, we ran new synthetic experiments (Appendix H.2 in the revised manuscript) showing that 2-HOD-GNN separates 34 pairs of 3-WL–indistinguishable graphs, directly confirming the theory. This evidence led the reviewer to raise their score.

- **Generalization ability of HOD-GNN** Reviewer WWrV asked about the generalization properties of HOD-GNN. In response, we compared its train–test gap on MOLHIV and MOLTOX21 against baselines that are both more and less expressive. The results (Appendix H.5 in the revised manuscript) show that HOD-GNN demonstrates stronger generalization than all these baselines. We additionally provided intuition for why HOD-GNN is expected to generalize well.

- **Fine-grained theoretical analysis of the expressivity  gains obtained from derivative signals**: Reviewer S9Dt requested a more detailed analysis quantifying “how much expressivity can be gained” by using derivatives of different orders. In response, we added several complementary theoretical results, presented in Sections E and F of the appendix in the revised manuscript:

   - **Proposition F.1** 1-HOD-GNN using only output-level derivatives can distinguish some graphs that 1-FWL cannot, effectively saving one level of tuple complexity.
    - **Theorem E.6** k-HOD-GNN is upper-bounded by (k+1)-FWL, showing that this one-level gain is maximal when using node/output derivatives.
    - **Proposition F.2** Adding edge-wise derivatives enables 1-HOD-GNN to separate some 2-FWL–indistinguishable graphs, yielding a two-level gain under standard sparsity assumptions.

- **Other comments**: The remaining comments raised by reviewers were minor, and follow-up replies from the reviewers indicated that these had been satisfactorily resolved.

**Post rebuttal phase**:

We believe our responses provided strong evidence addressing the reviewers’ concerns. Reviewer nnEj appreciated the additional experiments and raised their score to 6.
Reviewers 986R and S9Dt both described our rebuttal as detailed and satisfactory, noting that it addressed their concerns and confirming that they would keep their positive scores.
Reviewer WWrV noted that most of his concerns had been resolved and posed  one final clarification regarding the experimental setups.  We explained that baseline results were taken directly from their original papers and that our own experiments followed the same widely adopted protocols used in those works. This approach ensures transparency and fairness in comparison and is consistent with standard practice in the field.

---

### Meta-Review · Area_Chair_5TQa · 2026-01-04

**Summary:**

This paper shows that GNN derivatives improve the network's expressive power. The reviewers were mostly positive: three scores slightly above the threshold and one score below. The reviewer who gave a score below the threshold requested extra experiments that were provided by the authors and was satisfied with the authors' response. My recommendation is to accept (poster).

**Reviewer Concerns:**

Some reviewers were concerned with whether the GNN derivatives actually improve the models in practice and not just in theory. They requested extra experiments to support the claims. The authors provided that.

One reviewer asked if it was possible to give a more precise mathematical statement about the increase in expressivity. The authors responded to that request by adding new theoretical results.

**Reviewer Scores:**

I believe that after the discussion, all reviews would have been above the acceptance threshold.

---

### Decision · Program_Chairs · 2026-01-26

Accept (Poster)